# Riparian Area Changes in Greenness and Water Use on the Lower Colorado River in the USA from 2000 to 2020

Pamela L. Nagler [1,*], Armando Barreto-Muñoz [2], Sattar Chavoshi Borujeni [3,4], Hamideh Nouri [5], Christopher J. Jarchow [2] and Kamel Didan [2]

1    U.S. Geological Survey, Southwest Biological Science Center, Tucson, AZ 85721, USA
2    Biosystems Engineering, University of Arizona, Tucson, AZ 85721, USA; abarreto@email.arizona.edu (A.B.-M.); christoj@email.arizona.edu (C.J.J.); didan@email.arizona.edu (K.D.)
3    School of Life Sciences, Faculty of Science, University of Technology Sydney, Sydney, NSW 2007, Australia; chsatar@gmail.com
4    Soil Conservation and Watershed Management Research Department, Isfahan Agricultural and Natural Resources Research and Education Centre, AREEO, Isfahan 8174835117, Iran
5    Division of Agronomy, University of Göttingen, Von-Siebold-Strasse 8, 37075 Göttingen, Germany; hamideh.nouri@uni-goettingen.de
*    Correspondence: pnagler@usgs.gov; Tel.: +1-520-670-3357

**Abstract:** Declines in riparian ecosystem greenness and water use have been observed in the delta of the Lower Colorado River (LCR) since 2000. The purpose of our case study was to measure these metrics on the U.S. side of the border between Hoover and Morelos Dams to see if declining greenness was unique to the portion of the river in Mexico. In this case study, five riparian reaches of the LCR from Hoover to Morelos Dam since 2000 were studied to evaluate trends in riparian ecosystem health. We measure these riparian woodlands using remotely sensed measurements of the two-band Enhanced Vegetation Index (EVI2; a proxy for greenness); daily evapotranspiration (ET; mmd$^{-1}$) using EVI2 (ET(EVI2)); and an annualized ET based on EVI2, the Phenology Assessment Metric (PAM ET), an annualized ET using Landsat time-series. A key finding is that riparian health and its water use has been in decline since 2000 on the U.S. portion of the LCR, depicting a loss of green vegetation over the last two decades. EVI2 results show a decline of −13.83%, while average daily ET(EVI2) between the first and last decade had a decrease of over 1 mmd$^{-1}$ (−27.30%) and the respective average PAM ET losses were 170.91 mmyr$^{-1}$ (−17.95%). The difference between the first and last five-year periods, 2000–2005 and 2016–2020, showed the largest decrease in daily ET(EVI) of 1.24 mmd$^{-1}$ (−32.61%). These declines come from a loss in healthy, green, riparian plant-cover, not a change in plant water use efficiency nor efficient use of managed water resources. Our results suggest further deterioration of biodiversity, wildlife habitat and other key ecosystem services on the U.S. portion of the LCR.

**Keywords:** Lower Colorado River; riparian vegetation; EVI2; daily ET; PAM ET; water use; Landsat; remote sensing

## 1. Introduction

Accurate estimates of evapotranspiration (ET; water evaporated by bare soil and transpired by riparian plants) for riparian, urban green spaces, and cultivated lands are needed for management tasks at all scales. These tasks include scheduling for irrigation; sustaining agricultural production; securing foods and safe water quality and quantity for human uses; managing watersheds; allocating water; determining water rights; forecasting weather; and monitoring, managing, and projecting the long-term effects of land use change and global climate change on water resources [1–5]. Remotely sensed ET maps are useful for negotiating interstate and international water agreements, determining allocations for mining, urban use or natural resources managed by the U.S. Department of Interior, tribes

and citizens, and for estimating water use by natural vegetation which creates habitat and requires protections for native and by invasive species [6–10]. Remotely sensed ET studies are often focused on agricultural water needs in drylands and importantly assess the impact of drought on consumptive water use [11–13]. Thus, there has been an explicit need to increase the accuracy of riparian ET to better manage ecosystems for wildlife, particularly to monitor native and non-native riparian plants and their changing compositions and respective water use [14–16]. Some studies have focused on the relationship between groundwater and ET [17–21], while many riparian water use studies have focused explicitly on non-native species [22–24]. Accurately measured and validated riparian ET [25–30] and urban ET [31] have rarely been included in water budgets because of their small land cover relative to vast agricultural plots and, because of this, the majority of remotely sensed ET work is focused on agricultural water balance methods and models (i.e., crop coefficients, energy balance, and thermal methods) [32–37].

Many techniques to measure ET are used both on the ground and remotely. Improvements in ground methods have increased our ability to measure ET in drylands using only optical wavelengths and ground meteorological information, and these ET estimates are sufficient for land manager decision-making [38,39]. Vegetation index-based crop coefficients have been used to estimate ET by remote sensing in both agricultural and natural ecosystems [40]. Remote sensing from satellites is perhaps the only feasible means for spatially capturing and scaling ET over large landscape units [41]. Transpiration and ET at the plot scale can be scaled over larger regions such as the river reach [42]. Early research in the southwestern U.S. was conducted using remote sensing-based methods in hydrology [43–46], and these methods for mapping ET have marked advantages over conventional field methods and have proven to be useful for providing near-instantaneous measurements for large areas and continuous spatial coverage [47–50].

### 1.1. The Lower Colorado River: Water Accounting and Evapotranspiration

The Bureau of Reclamation stated in 2012 that "the Colorado River and its tributaries provide water to nearly 40 million people for municipal use and supply water to irrigate nearly 5.5 million acres of land. It is the lifeblood for at least 22 federally recognized tribes, seven National Wildlife Refuges, four National Recreation Areas, and 11 National Parks" [51]. Reclamation's 2012 study also predicted that the gap in water supply and demand will increase to nearly 4 million acre-feet (maf) by 2060, with significant shortages starting back in 2017 [51]. Water allocations and deliveries are for the following resources: municipal, industrial, and agricultural use; hydroelectric power generation; recreation; fish, wildlife, and their habitats (including candidate, threatened, and endangered species); water quality including salinity; flow- and water-dependent ecological systems; and flood control [51]. Furthermore, a U.S. Geological Survey (USGS) study describes water use data four ways: (1) water that is withdrawn from a source (groundwater or surface water, fresh or saline), (2) water that is delivered (domestic homes), (3) water that is unavailable (consumptive use, ET), and (4) water that is returned to a water resource (via wastewater returns) [52].

Water shortage in the Colorado River Basin is one of the major threats to riparian ecosystems. In drylands, more than 90% of continental precipitation returns to the atmosphere in the form of ET [53,54]. A sound understanding of riparian plant transpiration and quantification of water use in arid land ecosystems is integral to predicting water balance changes that can be due to drought and increasing maximum temperatures; changing temporal patterns of precipitation and timing of monsoon rains [55–58]; and increases in salinity, fire, land clearing, and land use activities [59–62]. Alterations to streamflow by water diversions and groundwater pumping [63,64] can affect phreatophytes by lowering base flows and water tables [65–67] and affect woody riparian species by reducing available water [68,69]. Physiological competition may affect the extent and type of riparian vegetation and its usefulness as wildlife habitat [70–72]. These pressures lead to changes in plant community composition [73]; thus, it is especially important to quantify spatiotemporal

variability or changing land use and land cover in riparian area greenness and its actual ET [74–76].

Riparian vegetation water use is dependent on hydrological processes that include groundwater/surface (blue water) availability, timing, and duration of large streamflow events, and whose dynamics are derived from climatic controls on precipitation and mountain snowpack [77], changes in carbon balance [78,79], carbon uptake [80], and canopy temperature [81]. Future climate projections indicate that dryland regions face generalized temperature increases with largest warming in the summer months, and a likely decrease in precipitation, particularly in the cool season [82–84]. Recent studies have demonstrated that the effects of climate variability and hot drought cause stress and mortality on woodlands [85–87], which could be the reason, in part, for the dramatic losses that were reported in the LCR delta, with up to a 51% decrease in riparian vegetation in some reaches over the last 20 years [14].

### 1.2. Tamarix, Tamarix Biocontrol, and Dryland Riparian Health

Major land cover changes in the riparian corridor of the LCR include invasive species colonization, particularly by salt cedar (*Tamarix* spp.), and more recently by *Diorhabda* spp., the salt cedar leaf beetles or tamarisk defoliation beetles, whose presence since 2014 in the LCR is causing defoliation of salt cedar plants. *Diorhabda carinulata* defoliates salt cedar in riparian zones at latitudes above 38°N because it has a photoperiod requirement for diapause (dormancy) and beetles would experience premature dormancy and impaired reproductive success [88]. *D. carinulata* was released in the Upper Colorado River Basin (UCRB) about 20 years ago starting in 2001 [89,90]. The beetles feed exclusively on salt cedar leaves, resulting in episodic defoliation events with some mortality of shrubs after several cycles of defoliation [91,92]. Defoliation events typically result in a sharp decrease in VIs measured for salt cedar stands in mid-summer, followed in many cases by a rapid regrowth of new leaves after 3–4 weeks when beetles become dormant or move on [93], but recently multiple cycles of defoliation have occurred in one growing season [94]. The repeated defoliation of green leaves denudes the riparian corridor and has allowed for secondary invasive species to thrive in its absence [95]. Over a period of years of repeated defoliation, salt cedar shrubs can suffer mortality resulting in a permanent reduction in ET [96]. The beetle biological restrictions would prevent them from entering the LCR where the endangered southwestern willow flycatcher (*Empidonax traillii extimus*) nests in salt cedar in the absence of native willows [97]. Unfortunately, this beetle species adapted to the new environment and entered the LCR a decade ago and has been defoliating in southwestern willow flycatcher habitat [98–100]. Several studies have evaluated beetle–salt cedar interactions with satellite imagery on various river systems with vegetation indices and other optical band methods used as indicators of the extent of defoliation [101,102] and ET [103–105]. The beetles exhibit a pattern of colonize–defoliate–emigrate, so that riparian zones contain a mosaic of completely defoliated, partially defoliated, and refoliated salt cedar stands [106,107]. As of 2019, *D. carinulata* was observed as far south as Yuma, Arizona [97], for the first time since it was released along the rivers and streams of the Colorado Plateau. The subtropical beetle, *D. sublineata*, was released in the Rio Grande region in 2009 [108–111] and is expected to eventually move west into the LCR and might eventually be nearly as widespread as salt cedar shrubs [112]. Several research papers have documented that defoliation from *Diorhabda spp.* is now proceeding at a rate of approximately 40–100 km yr$^{-1}$ along river systems [93,105], much faster than initially projected by DeLoach et al. (2004) [112].

Our goal of quantifying not only riparian woodlands water use but also discriminating non-native plant area dominated by salt cedar and quantifying its wide-area, reach-level water use, is of particular importance to land managers in the southwestern U.S. Salt cedar was once deemed a high-water user until it was shown that its water use is lower than previously assumed [113] and ranges widely (i.e., 0.42 to 1.18 m yr$^{-1}$) [114] depending on stand density, and thus its water use may be similar or lower than native riparian

species [115,116]. The presence of beetles has a short-term effect on ET and water salvaged for instream uses, but regrowth of salt cedar leaves is just one example of how the riparian vegetation community changes rapidly in response to disturbances [117]. Recent research on the upper basin rivers measured over a decade after biocontrol and until 2017 showed that salt cedar cover across sites initially declined an average of ~50% in response to the beetle, but then recovered, confirming remotely sensed data trends and demonstrating that riparian plant communities remain stable in response to a second cycle of biocontrol defoliation [118].

Our study examines the riparian ecosystem over a 21-year period since the year 2000 using remotely sensed imagery from Landsat in 16-day time-steps, allowing for near-real-time monitoring and timely management intervention. Validation of riparian species coverage and water use measured on the ground has been critical for scaling to the river wide area. For example, riparian plants have a range of leaf shapes and coatings (e.g., narrow needles versus planar) and their transpiration rates vary depending on the species, light interception, leaf area index (LAI), and canopy characteristics which do not scale to the landscape-level easily without species-specific allometric measurements [119]. VI-based measurements are an accurate proxy for the leaf and canopy traits of these riparian plant species, and thus VI-based estimates of ET have been validated with flux tower and sap flow measurements [25–27] and perform well in these riparian systems of the LCR where many previous ground-based water use estimates at the local measurements of scale have been made [28,29]. Dynamic estimates of ET are challenging to quantify because they vary both within and among years in response to fluctuations in its driving factors. We use a Phenology Assessment Metric (PAM), which is ET over the full-year or PAM ET, to explain phenological shifts that could be attributed to environmental factors such as changes in the monsoon or other precipitation patterns, changes in temperature, land cover impacts due to fire, clearing, and beetle damage [14]. We provide long-term trends to track not only changes in vegetation greenness in the riparian corridor, but also to monitor these areas with the ability to observe data variations and anomalies. Changes in the regular phenology trends are expected to be due in part to biocontrol impacts, as well as excessive heat, reoccurrence of droughts, fire, secondary plant invasions, and other vegetation community changes, such as die-off that is correlated with rainfall delivery and temperature fluctuations.

*1.3. Objectives*

Our hypothesis is that declines in the overall riparian health and water use would be observed on the U.S. side of the border between Hoover and Morelos Dams over the past two decades similar to what has happened in the delta of the Colorado River south of Morelos Dam and into Mexico [14]. Our objectives were to (i) delineate the riparian corridor into reaches from Hoover Dam in the north to Morelos Dam in the south (the first two reaches were in canyons with little riparian vegetation and exempted from this study) and (ii) track long-term greenness and water use over 21-years (2000–2020) using remotely sensed, time-series data for the riparian corridor between these two dams using the entire USGS/NASA collection of 30 m Landsat since 2000. Our specific goals are to apply recently developed analyses tools [14] to the U.S. riparian zone LCR to assess land cover changes as follows: (a) in five reaches, (b) in three time sets including annually, periods of 5–6 years, and periods of decades, and (c) using three metrics: (a) vegetation index (VI), (b) daily peak growing season ET, and (c) full-year PAM ET. We characterize riparian plant health and its water use overall, by reach, without separating native and non-native plant species or removing any of the small restoration plots from our analyses. Our aim is to assess system pressures at certain time points relative to increases and decreases in riparian health. These pressures include increasingly limited water and longer, maximum air temperatures as well as salt cedar biocontrol.

*1.4. Research Questions*

In this study, we assess how an environmental asset such as healthy riparian vegetation and habitat is threatened by changes in natural- and human-induced pressures on the plant communities along the LCR. We provide responses to the following research questions, the first set about trends and the second set about causes.

Trend Questions: (a) What are the riparian ecosystem VI and ET values and their trends over the last 21 years? (b) Using year-to-year and groups of years to determine changes, has the riparian ecosystem health, as measured by VI and ET, been increasing or decreasing when comparing year-to-year, and periods of five years (first and last), and periods of ten years (first and last decades)? (c) How much has the riparian vegetation changed by reach? (d) Has there been any observed shifts in green-up start dates or onset of senescence? (e) Are reaches that are dominated by salt cedar showing different outcomes in VI and ET than other reaches? (f) Can the onset of biocontrol be detected from the remotely sensed data for this riparian corridor?

Cause and Effect Questions: (a) Are there relationships between precipitation, temperature, and/or biocontrol and riparian health? (b) Are there reach–level relationships between native and non-native plant communities and the trends in greenness and water use? (c) How do greenness and water use in riparian reaches compare pre-beetle and post-beetle and to other areas such as the delta where the beetle has not yet arrived?

## 2. Materials and Methods

*2.1. Study Area*

The Colorado River traverses the Colorado Plateau and abruptly turns south near Las Vegas, Nevada, forming the California/Arizona border in the U.S.A. before continuing to the Sea of Cortez in Mexico. For this study, the narrow riparian vegetation of the LCR between Hoover and Morelos Dams is the focus of this study. We selected and digitized the riparian zone study areas after viewing the entire USGS/NASA imagery dataset over the 21-years to select areas of natural vegetation; however, note that within the LCR bottomlands there are urban areas with parks and golf courses, and agricultural lands that have been reverted to natural areas passively or used for active riparian restoration such as by the U.S. Bureau of Reclamation Multi-Species Conservation Plan (MSCP) [120]. These important restoration plots were considered within the boundary of our riparian reaches in this research, but agriculture and urban land cover was removed. We divided the river into seven reaches (Reaches 1, 2, 3, 4, 5, 6, and 7, labeled R1–R7) with Reach 1 (R1) beginning at Hoover Dam in the north and Reach 7 (R7) ending at Morelos Dam in the south. Reaches 1 and 2 (R1, R2) were excluded from this study because of canyons which had very little visible green vegetation; our study only evaluates Reaches 3-7 (R3 to R7) (Figure 1). Additionally, R4 has very little riparian vegetation as it is comprised of a narrow corridor with canyons on both sides. Herein we refer to the total riparian area as "All". All is computed using a weighted average to take into account the relative size of reaches. The total acreage of 63,628 ha varies by reach, with R3 = 16,430 ha, R4 = 159 ha, R5 = 22,645 ha, R6 = 13,371 ha, and R7 = 11,024 ha. The weighted value is used in our assessment and evaluation as the sum All (R3 to R7) in our figures and tables.

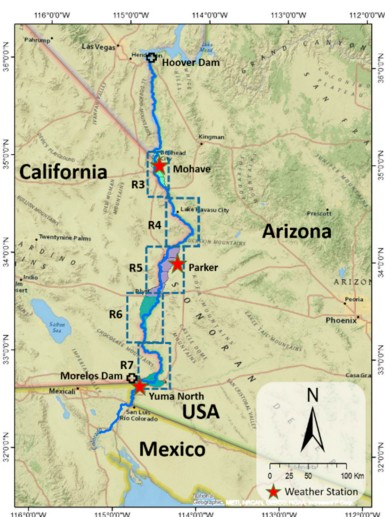

**Figure 1.** Study area divided into five riparian zones (R3 to R7) outlined in dashed boxes on the Lower Colorado River between Hoover Dam and Morelos Dam on the border of the U.S. and Mexico, with the two dams shown (black circle symbol) and three Arizona Meteorological ground stations (AZMET) used for the weather variables collected over the 21 year study (red stars).

### 2.2. Data Acquisition and Analyses

We acquired Landsat time-series data to perform these analyses: (1) assess vegetation greenness using Normalized Difference Vegetation Index (NDVI) and EVI2 for the study period 2000–2020; (2) estimate daily, peak growing-season ET; (3) estimate annual PAM ET; and (4) provide change maps of scaled NDVI (i.e., NDVI*) and ET for the LCR riparian corridor and its five reaches individually.

### 2.3. Remote Sensing Data: Satellites and Sensors Used

Our study follows the data acquisition and calculation of the VI, daily ET, and annual PAM ET methods used recently in the riparian ecosystem study focused on the delta of the Colorado River [14]. To cover the full period 2000–2020, we combined data from different Landsat missions: Landsat 5 (TM5), Landsat 7 (ETM+), and Landsat 8 (OLI). All the Landsat scenes for the area—038037, 038036, 039037, and 039036—were acquired using the USGS Earth Resources Observation and Science Center Science Processing Architecture (http://earthexplorer.usgs.gov/, accessed on 24 February 2021). While atmosphere correction is now a standard remote sensing tool for the removal of most atmosphere contaminants [121], the resulting data still exhibit poor quality and erroneous outliers resulting from subpixel clouds, haze, shadows, and thick hard-to-correct-for aerosols. We filtered out these observations prior to any change detection analysis to minimize spurious results. We have developed and implemented a rigorous per-pixel, QA-based, filtering approach aimed at capturing these poor-quality observations [122]. We applied an additional 8-pixel-wide buffer around all cloudy pixels aimed to remove shadow and residual clouds in addition to following protocols in published methods [123].

We note that ETM+ developed a Scan Line Corrector failure in 2003, resulting in scenes with major spatial gaps and making the data of limited use; nonetheless, we wanted to have a complete 21-year time-series and needed to use ETM+ in the temporal gap in 2012 between the decommission of TM5 and OLI coming online, as was recently applied [14]. We bridged this temporal gap in the time series (2000–2011, 2013–2020) with data from ETM+ using only 32 scenes between the end of 2011, all of 2012, and the beginning of 2013. The number of observations in this study was 249 from 2000 to 2011 from TM5 (decommissioned in 2012), 32 observations using ETM+, and 148 observations from 2013 to 2020 from OLI. A simple gap-filling algorithm based on neighboring temporal observation of the before-and-after data was used to fill temporal gaps. These transfer functions were generated and used to transfer one sensor data to the other; in this case, OLI was the

reference and TM5 and ETM+ were converted to OLI-like data in a process [122]. Thus, all data were then standardized to OLI and all remaining spatial gaps filled with long-term average continuity corrected data.

OLI data showed an apparent overall amplitude increase in the VI, due to the dynamic range and thus required additional continuity adjustment to address the TM5, ETM+, and OLI series range differences [124]. Specifically, TM5 and ETM+ require processing to match the OLI imagery dynamic range, which is consistently higher. Cross-sensor continuity was applied to ensure a consistent and comparable dynamic range.

### 2.4. Mathematical Components

2.4.1. Vegetation Indices

In the optical wavelengths, VI at the Landsat spatial resolution shows that green vegetation in the field-of-view is practically the same no matter what the plant species on the ground is. Therefore, the two-band Normalized Difference Vegetation Index (NDVI) works well over non-dense vegetation, as it optimizes and stretches the dynamic range [122]; however, it also remains very sensitive to the background signal like soil color and wetness. For this reason, we also decided to use EVI2 for capturing green riparian vegetation extent and density in the pixel area. However, NDVI is not the best VI to use because the signal does not capture all the information from the ground in the wavelengths measured [125,126]. We include NDVI in our research because NDVI is the most common, longest-used VI in remote sensing research, and is used in this study for comparison to a more improved Enhanced Vegetation Index (EVI2), the two-band version of EVI which uses three bands (see https://lpdaac.usgs.gov/documents/178/VIP_User_Guide_ATBD_V4.pdf, accessed on 24 February 2021) [14,125–128]. Our data produced an $r^2$ = 0.99 between EVI and EVI2 (not shown) which shows that EVI and EVI2 are practically identical in these riparian reaches [14]. The two-band version, EVI2, is a simpler VI because it does not require the blue band. We selected EVI2 to parameterize the ET algorithm.

NDVI and EVI were available as part of the standard Landsat data acquired from Collection 1 Level-2 (Tier-1 only). Atmospherically-corrected surface reflectance in the red, blue, and near-infrared (NIR) (https://www.usgs.gov/centers/eros, accessed on 24 February 2021) were also acquired to derive the EVI2 time-series data, which is computed with following [127] Equation (1):

$$\text{EVI2} = 2.5 \times (\text{NIR} - \text{red})/(\text{NIR} + 2.4 \times \text{red} + 1.0) \tag{1}$$

where NIR and red are the corrected surface reflectance.

As mentioned previously, there are differences in the dynamic range of VIs derived from the three Landsat sensors (TM5, ETM+, and OLI) which complicate time-series and change analyses [122]. To mitigate this issue, we followed original methods aimed at minimizing inter-scene biases and inter-sensor differences which we have applied in recent research [14,19,123]. The NDVI scaling algorithm denoted as NDVI* Equation (2) normalizes NDVI values by stretching them between $\text{NDVI}_0$ and $\text{NDVI}_s$ [129].

$$\text{NDVI*} = (\text{NDVI} - \text{NDVI}_0)/(\text{NDVI}_s - \text{NDVI}_0) \tag{2}$$

where NDVIs is the maximum saturation of the NDVI signal and NDVI0 is the absolute NDVI minimum corresponding to bare soil [129].

We produced NDVI* from 2000 to 2020 only to be used in our change maps. NDVI* minimizes scene-to-scene illumination differences and is used with Landsat imagery [74,75]. NDVI* is an additional, processing correction to the NDVI to account for the illumination conditions and to improve the usefulness of NDVI.

2.4.2. Vegetation Index (VI) Based Evapotranspiration (ET) Algorithms

Two types of methods have been developed to estimate ET from remote sensing data [26]. In this study, we focus on the empirical/statistical relationships that we previ-

ously developed using ET measured or estimated on the ground to project to larger scales through remotely sensed VIs. Note that in both methods there are errors in predicting ET because of inaccuracies in the ground information, which needs to be less than 10% due to propagation when scaling ET [41]. If ground methods continue to improve, remotely sensed ET will improve as well. These algorithms are either empirical approaches or physical models that derive the inputs to the Penman–Monteith [130,131] and Priestley–Taylor [132] algorithms using both ground and remote sensing information ET from moisture flux towers [26,27], which provide continuous measurements of actual and potential ET with an accuracy or uncertainty of 10–30% [38,39,41]. EVI2 was used as input to the ET algorithms developed from empirical relationships over a variety of land cover types [29,42] and often was used in dryland riparian areas [93,105]. The ET algorithm used in this study was first developed with data from Bowen ratio and eddy covariance moisture flux towers in riparian areas in the southwestern U.S., including the LCR [25], and was then parameterized [26] and applied in other riparian zones [29] taking the form in Equation (3).

$$\text{ET} = \text{ET}_0 \times [1.65\,(1 - e^{-2.25 \times \text{EVI (or EVI2)}}) - 0.169] \qquad (3)$$

This ET formula was initially designed for use with MODIS EVI (or EVI2) but has been modified for use with Landsat imagery at the 30 m resolution scale [14], and thus has been adopted and applied to finer scales, demonstrating an important improvement and beneficial tool for assessing narrow corridors of riparian vegetation [14,116]. Therefore, EVI and EVI2 are competent predictors of ET for natural vegetation in drylands [14,29,70,71]. Using EVI2 and nearest ground-based meteorological data, we calculated daily ET (mmd$^{-1}$) for the peak growing season (May 1 to Oct 30) and the full year with ~23 scenes per year and showed this as a time series. ET$_o$ was acquired from three of the Arizona Meteorological Network (AZMET) stations, one from Yuma North Gila, Parker, and Mohave (https://cals.arizona.edu/azmet/, accessed on 24 February 2021), the closest operating meteorological stations to the region of interest. R3 is within 20 km to the Mohave station and R4 is 85 km to the Mohave station. R4 is 25 km to the Parker station, R5 is 15 km to Parker, and R6 is 80 km to Parker. R6 is 60 km to Yuma North Gila station and R7 is 10 km to Yuma North Gila station. These stations are marked on Figure 1 with red stars. For assisting our interpretation of the EVI2 and ET trend data we provide plots of the weather data variables, maximum air temperature, precipitation and ET$_o$ from each of the three AZMET stations: Yuma North Gila, Parker, and Mohave (Figure 2).

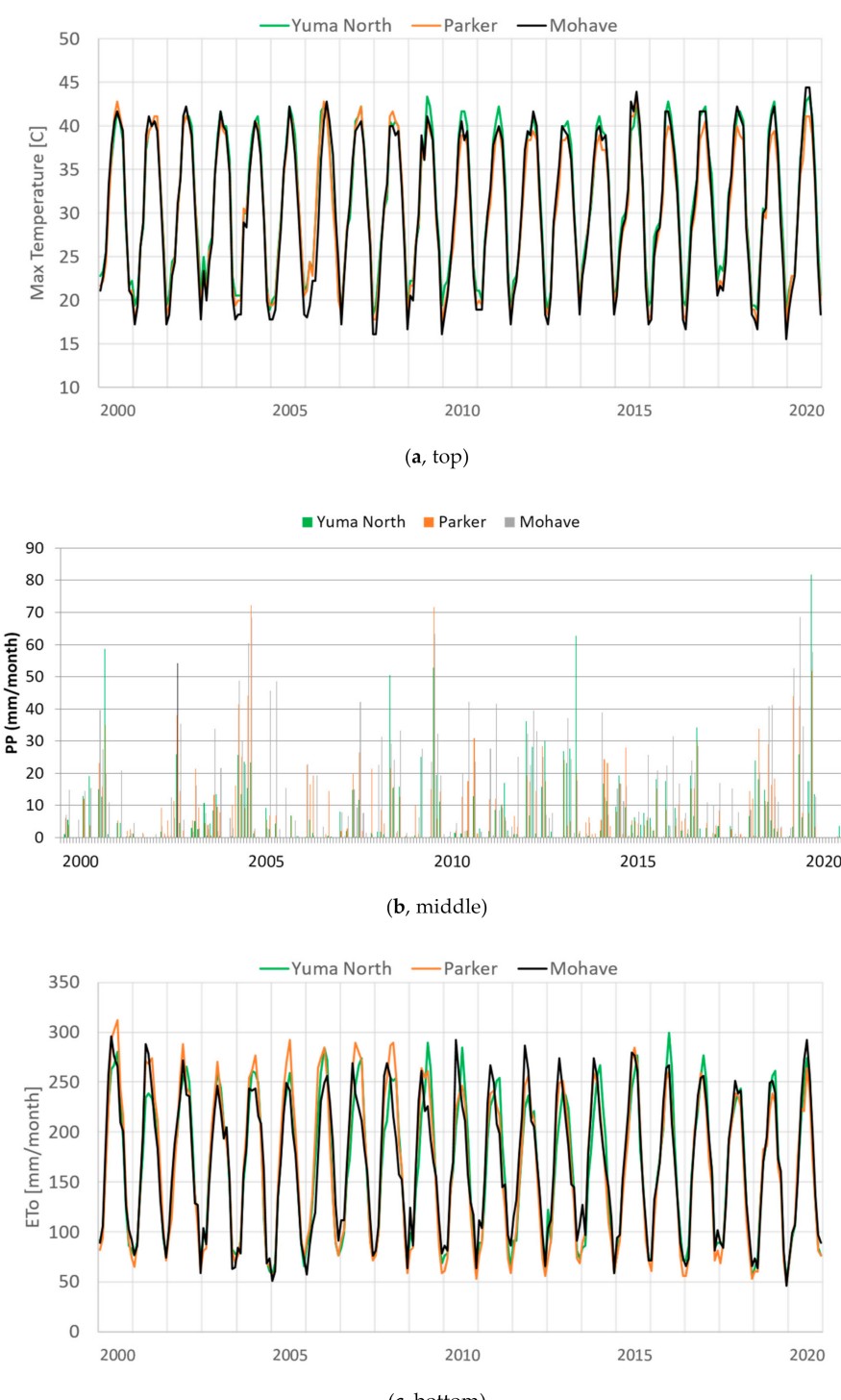

**Figure 2.** Weather variables from three Arizona Meteorological Network (AZMET) stations (https://cals.arizona.edu/azmet/, accessed on 24 February 2021) located from north to south along the Lower Colorado River (Mohave, Parker, Yuma North Gila) showing 21 years of data plotted for maximum air temperature (**a**, **top**), precipitation (**b**, **middle**), and $ET_o$ (**c**, **bottom**).

### 2.4.3. Phenology Assessment Metric (PAM) from Annual ET

PAM ET ($mmyr^{-1}$) was recently created for detecting overall plant-related phenology shifts and applied to measure changes in the greenness of riparian areas with cycles of plant stress, leaf drop, defoliation, and/or re-greening, such as those with disturbance from the salt cedar biocontrol beetles, drought, heat, fire, or land clearing [14]. We previously

formulated this annualized ET metric to understand and better capture changes in the full-year ET by summing the area under the VI curve [14]. PAM integrates VI over the whole year because natural vegetation in this region is often growing the entire year, not just the growing season (1 May to 30 October); furthermore, the growing season is primarily used with agricultural crops where the interest is in quantifying peak water use. In agriculture, daily ET is estimated, summed to estimate monthly ET, and then further summed to estimate annual ET. For some research methods, remotely sensed values of ET are calculated only over the peak growing season months, usually between May and October using the 16-day satellite periods. We report a daily ET value using May to October, but we also want to account for the non-growing season months, November to April, where growth in this region can be quite high, and certainly not zero. We found that there are overestimates of annual ET when simply summing the daily ET values for the full year [14]. For those studies which use only the growing season, PAM ET is more accurate.

The advantage of applying our new PAM ET method is that a full year assessment of riparian woodland annual ET is produced using the full-year 16-day periods in time series using EVI2 (or EVI). The area under the curve then captures the oscillations in the satellite data that could be due to changes in the landscape. The PAM ET determined from the full year thus has advantages over the daily ET from the peak growing season because it captures the year-round growth that these riparian communities experience. Using ET for the full year, we produce annual totals using Equation (4):

$$\text{PAM} = \sum_{i=1}^{n-1} \left( \frac{\text{ET}_{i+1} + \text{ET}_i}{2} \right) (\text{DOY}_{i+1} - \text{DOY}_i) \tag{4}$$

where $\text{ET}_i$ = ET for period i and $\text{DOY}_i$ = Day of Year for period i and n= number of satellite images acquired for the year.

PAM ET captures the magnitude of water use over a year despite interferences to the normal greenness phenology cycle from events such as fire, land clearing or, in the LCR, the salt cedar biocontrol beetle defoliation events, regardless of the changing onset and senescence of overall vegetation greenness [14]. PAM ET is a simple tool that provides a fast way to evaluate long-term trends in total annual ET, particularly when applied to disturbed riparian ecosystems. The beetle can defoliate salt cedar multiple times in one growing season and these shrubs re-sprout quickly after fire; therefore, PAM ET is useful for quantifying annual water use and depicting changes in total ET year to year.

Our results are structured to understand the land cover changes within the LCR riparian corridor over the past 21 years. We have three metrics (EVI2, ET (daily, mmd$^{-1}$), and PAM ET (annually, mmyr$^{-1}$)), five reaches, and seven sets of grouped years in addition to the full set of 21 individual years. Our results are organized to first report the absolute values, then the difference in these three metrics, followed by reporting the percent change over time. First, we report EVI2, daily ET, and the annualized PAM ET for the total area of vegetation community change for five reaches (R3 to R7) and for the total area (the total of the five reaches, "All"). The reach composition varies by (i) gallery forest, (ii) shrubs and shrub cover dominated by salt cedar, and (iii) salt cedar cover impacted by biocontrol. Next, we quantified the difference year-to-year. We then did this for groups of years averaged across periods of 5 and 10 years. For example, for each of our three metrics, we averaged the absolute value, difference, and percent change data from these seven periods: 2000–2020, 2000–2010, 2011–2020, 2000–2005, 2005–2010, 2011–2015, and 2016–2020. Last, we provide the percent change of these differences for the three metrics, five reaches, and seven periods in addition to quantifying percent change for the year-to-year periods. Appendix A, Figure A4 shows a flow chart of the methodology.

## 3. Results

We show NDVI in time-series because this index is a standard product, yet we provide change maps using NDVI* as scene-to-scene comparisons of change require the scaled version (hence NDVI* and not NDVI for change maps). We assessed NDVI and NDVI*, but our research results are primarily confined to data derived from EVI2: EVI2 for the peak growing season, EVI2 for the full year, ET with EVI2 as the input VI, and PAM ET using EVI2. We first report the absolute values, then the difference and percent change data from these seven periods: 2000–2020, 2000–2010, 2011–2020, 2000–2005, 2005–2010, 2011–2015, and 2016–2020.

### 3.1. Vegetation Index Time-Series from 2000 to 2020, EVI2

We produced time-series data for three VIs including NDVI, EVI, and EVI2, but because we found that EVI and EVI2 had an $r^2 = 0.99$, we proceeded with using only NDVI and EVI2 (Figure 3). NDVI (Figure 3a) and EVI2 (Figure 3b) were produced following our methods which account for amplitude shifts across three sensors, Landsat5 (TM5), Landsat7 (ETM+), and Landsat8 (OLI).

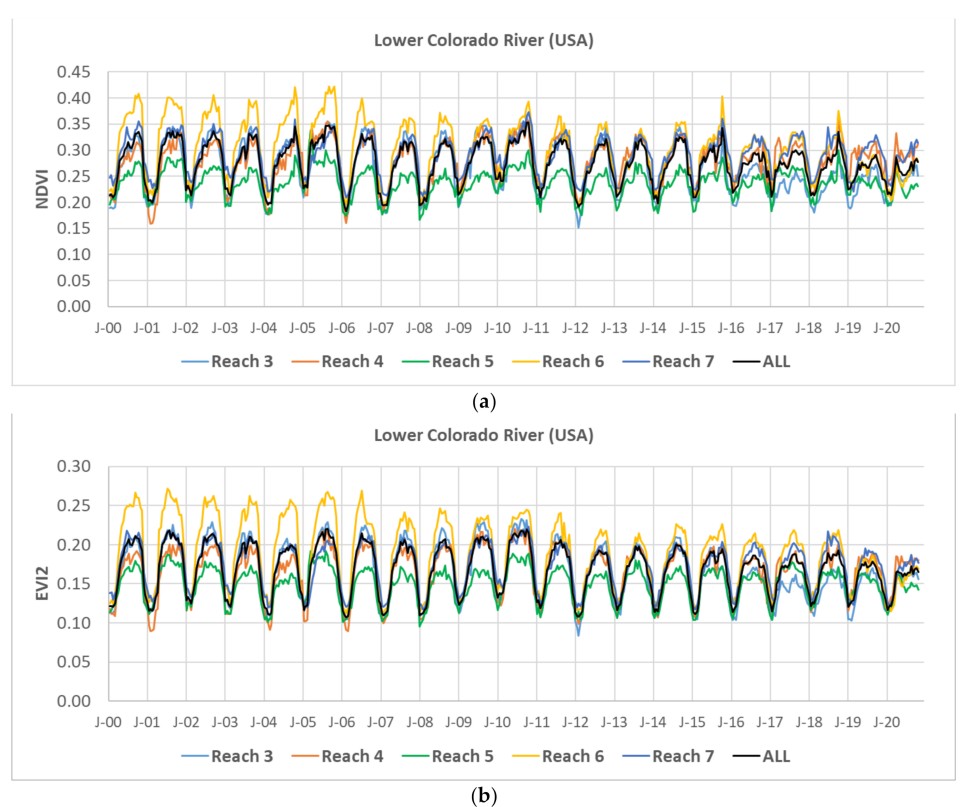

**Figure 3.** Time-series data from Landsat 5, 7, and 8 showing the riparian corridor of the Lower Colorado River from Hoover to Morelos dams in five reaches and the average weighted full riparian zone, All, for Normalized Difference Vegetation Index (NDVI) (**a**), and EVI2 (**b**).

Time series trend data, both NDVI and EVI2, since 2000are shown for each reach (R3 to R7) and the weighted average for the sum of the reaches (All) (Figure 3). This beginning date was selected because, in our previous study, we used both Landsat and MODIS to compare how closely they provided results, and we determined that they are essentially measuring the same information [14]. For that reason, we chose not to show MODIS data in this research, yet we begin the acquisition of Landsat time-series with the date that corresponds with the start of the MODIS collection. NDVI (Figure 3a) and EVI2 (Figure 3b) have some similarities and differences. The differences are that NDVI has a higher amplitude and nearly all of the results range between 0.2 and 0.4, whereas EVI2

ranges between 0.10 and 0.25, much lower. Furthermore, NDVI shows a deterioration of the signal beginning in 2015 through 2020, whereas EVI2 depicts these seasonal trends as having kept their shapes during this same period. NDVI does not capture the full spectrum the way EVI2 does; NDVI saturates at the high end and increases the signal amplitude, but EVI2 performs throughout the full red to near-infrared dynamic range [122,125,126]. The similarities are that generally Reach 6 (R6) has the highest VI, followed by Reach 7 (R7), Reach 3 (R3), and Reach 4 (R4), while Reach 5 (R5) has the lowest VI. In both trends, we see similar decreases in VI, such as a drop of 0.34 to 0.275 for NDVI and 0.22 to 0.175 for EVI2 between 2000 and 2020.

Because we use EVI2 and not NDVI as input to the ET algorithm, we focus our discussion on EVI2. The EVI2 is used to compute the peak growing season average for each reach (R3 to R7) and the weighted average (All) for the 21 years in the study. These EVI2 data are in the Supplemental Information, Appendix A, Table A1 (Appendix A, Table A1). In Appendix A, Table A2, the EVI2 difference between years for the peak growing season, averaged between May 1 and October 30, is provided for each reach (R3 to R7) and the weighted average as All Reaches (Appendix A, Table A2). We provide the percent change of EVI2 between individual years for the five reaches and the full All reach (Appendix A, Table A3). Last, we also provide the percent change of EVI2 between groups of years for the peak growing season, averaged between May 1 and Oct. 30. We provide the percent change between 2000–2020, 2000–2010, 2011–2020, 2000–2005, 2006–2010, 2011–2015, and 2016–2020 in Appendix A, Table A4. This shows EVI2 for the peak growing season averaged per reach and All for each of the seven periods using to compare changes (Appendix A, Table A4).

EVI2, when averaged only over the growing season between May 1 and October 30 each year, decreases from 0.1993 to 0.1650 over the 21 years (Appendix A, Table A1). The decrease in EVI2 for R5 is smallest, 0.1689 to 0.1494, and R6 has the largest increase, from 0.2484 to 0.1604, over the same time. The EVI2 difference between each 1-year period is 12 decreasing and eight increasing years. We see that the largest increase in EVI2, a jump of 0.0181, occurred between 2004 and 2005, and the largest decline, a drop of 0.0196, occurred in the period 2010 to 2011 (Appendix A, Table A2). Knowing in which year the most or least greening occurred is a major contribution to our research findings. Another new contribution is the calculated percent change between years (Appendix A, Table A3). We broke the average peak growing season EVI2 into two ten-year periods, 2000–2010 and 2011–2020, and four five-year periods, 2000–2005 (six years), 2006–2010 (five years), 2011–2015 (five years), and 2016–2020 (five years), for further evaluation of the increases and decreases over these periods so we could state which period depicted the most greening and the most loss in vegetation health (Appendix A, Table A4). Similarly, we can determine the differences and percent changes for these periods within the five different river reaches. The data show the five-year data had a peak greenness, 0.1995, occurring during 2000–2005, and the maximum loss of green vegetation, 0.1719, happened between 2016 and 2020. A decline of −0.01 in EVI2 was observed between the first and last ten-year periods. R6 showed the maximum greenness, 0.2466, in the 2000–2005 period and the least green vegetation, 0.1599, in R5 during 2016–2020.

To visualize these changes, Figure 4 shows five important statistics which include the minimum, first quantile (Q1, 25th percentile), median (Q2, 50th Percentile), third quantile (Q3, 75th percentile), and maximum, values that allow for a comparison of the reaches according to the EVI2 percent change (4a, top) and demonstrate the rate of change from year-to-year in all reaches (4b, bottom). In the top diagram, the darker the box, the larger the range of change. As an example, R4 shows the largest range of change in EVI2 (−8.33% to 17.49%), followed by R5 (−9.09% to 16.35%).

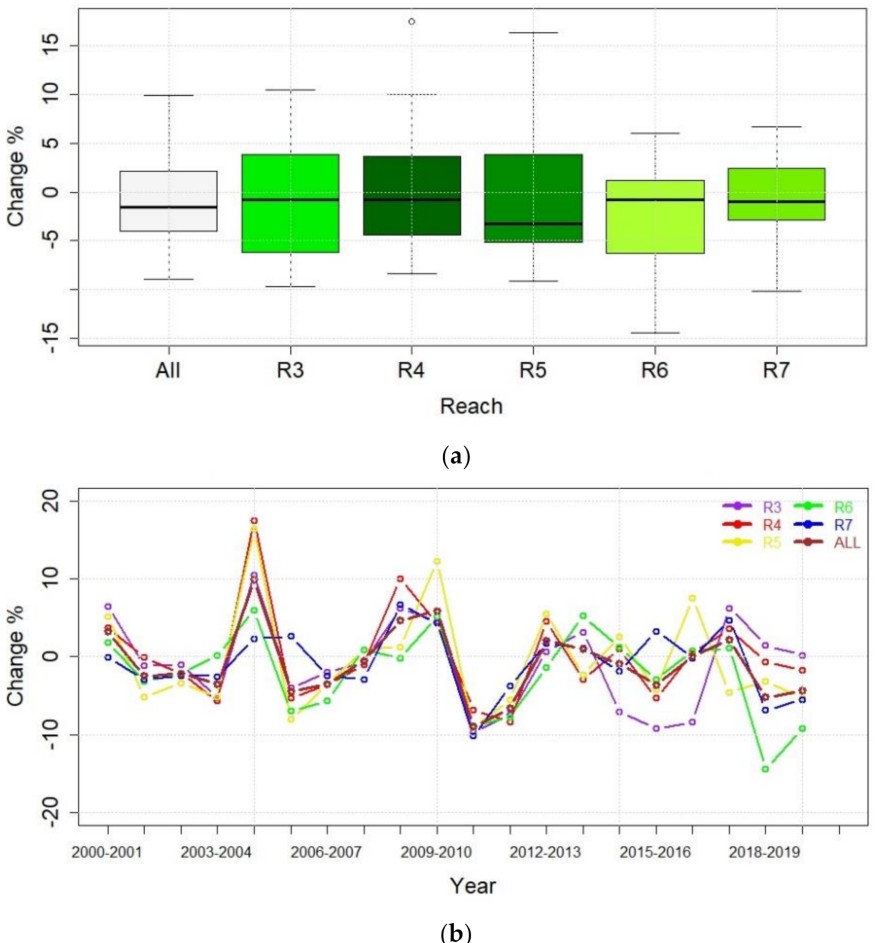

(**a**)

(**b**)

**Figure 4.** The percent change in EVI2 for the peak growing season for each reach and the entire LCR riparian corridor (All) is shown using plots with a horizontal line, box, and whisker ends that indicate the median, 25th and 75th percentiles, and the 10th and 90th percentiles, respectively, and the data points outside this range are shown by dots (**a**); the rate of change in EVI2 is shown from year-to-year during 2000–2020 and for each reach (**b**).

Because it can be difficult to ascertain the importance of these absolute differences in terms of their ups and downs over the growing seasons of these two decades, we provide the percent change of the EVI2 growing season VI data for these same time steps, year-to-year over 20 years, the two ten-year periods, and the four five-year periods. Figure 4 shows the direction (positive or negative (-)) of the percent change in box plots EVI2 for the weighted average of five reaches (All) and the individual reaches for each year-to-year period (Figure 4a) and the rate of change (Figure 4b). In Appendix A, Table A3 the values corresponding to Figure 4 are provided. Using these, the period between 2004 and 2005 shows large greening took place across all reaches, and the highest percent increases of all 20 years took place in this one one-year period where greening increased by 10.48% (R3), 17.49% (R4), 16.35% (R5), 6.00% (R6), and 2.29% (R7), an average of 9.57% increase (All).

This year 2004–2005 was also the period of most greening in the delta [14] due to nearly three years of rainfall after nearly zero precipitation in 2002 as determined by data from the AZMET Yuma station. In 2009, precipitation was only 20 mmyr$^{-1}$, but was followed by the highest rainfall, 150 mmyr$^{-1}$, of the past 21 years we studied [14]. The period between 2010 and 2011 shows large losses in greenness for the entire year and across reaches, where greening decreased by 9.65% (R3), 6.85% (R4), 9.09% (R5), 8.88% (R6), and 10.18% (R7), an average 9.38% decrease (All). This year was also the period of greatest decline in vegetation health as measured by VI for the delta [14]. The preceding year, 2009, had very low rainfall followed by peak rainfall, then low rainfall again in 2011

($\sim$45 mmyr$^{-1}$), so it is not surprising that vegetation declined not only over 2010–2011, but also in the following year, 2011–2012. The period 2011–2012 showed additional losses in vegetation greenness, i.e., R4 decreased $-6.85\%$ followed by $-8.33\%$, compounding the declining trend. For R6, the period from 2018–2019 showed decreases of $-14.43\%$, followed by a decrease of $-9.18\%$ the following year, 2019–2020. This was the most drastic percent change in the negative direction and is likely due to biocontrol of the vast salt cedar cover in the reach.

After the year-to-year comparisons were documented, we grouped the years into periods to do a comparison of percent change in EVI2 for the average peak growing season relative to other periods, for all reaches, and the full area (All) (Table 1). In this last analysis step, we provide the percent change of the EVI2 between groups of years and compare them with one another as shown in the first columns, Periods 1 and 2 (Table 1). Once again, the data are provided for each reach (R3 to R7) and the weighted average as All Reaches. The groups include the ten-year periods from 2000 to 2010 and 2011 to 2020 and four shorter periods of nearly five years: 2000–2005 (six years), 2006–2010, 2011–2015, and 2016–2020. We compare the changes in these time periods and determine percent change in EVI2 for each comparison made. A percent change in small amounts exist for R3 (2.06%), R4 (4.87%), and R7 (0.34%) when comparing 2000–2005 to 2006–2010; however, the greatest percent change was a $-13.83\%$ decrease in greenness which occurred for the five-year periods between 2000–2005 and 2016–2020. Over the 10-year comparisons, a decrease in green vegetation is as much as $-10.46\%$ between the two decades of 2000–2010 and 2011–2020.

**Table 1.** The percent change in EVI2 for the peak growing season (averaged between 1 May and 30 October) between two periods with both positive and negative ($-$) directions of change provided for each reach and the full area, All reaches, in the riparian corridor of the Lower Colorado River.

| EVI2 | | | | | | | |
|---|---|---|---|---|---|---|---|
| Period 1 | Period 2 | R3 | R4 | R5 | R6 | R7 | ALL |
| 2000–2005 | 2006–2010 | 2.06 | 4.87 | −1.53 | −7.02 | 0.34 | −1.67 |
| 2006–2010 | 2011–2015 | −9.18 | −4.34 | −1.75 | −10.61 | −6.97 | −6.89 |
| 2011–2015 | 2016–2020 | −16.24 | −4.51 | −1.93 | −7.31 | 1.02 | −5.88 |
| 2000–2005 | 2016–2020 | −22.36 | −4.21 | −5.12 | −22.96 | −5.70 | −13.83 |
| 2000–2010 | 2011–2020 | −15.63 | −4.07 | −3.51 | −17.29 | −6.32 | −10.46 |

The EVI2 is plotted for the full year, January 1 to December 31, for each of the 21 years, with 2000–2010 (top) and 2011–2020 (bottom) to demonstrate spatiotemporal changes across the full LCR riparian zone, All reaches (Figure 5). The same data are averaged across the full period, 2000–2020 (black solid line), shown for comparison. The same plots for 2000–2010 (top) and 2010–2020 (bottom) follow for the five individual reaches, R3 to R7 and are shown in Appendix A (Figure 1a–e). The data can be interpreted as the closer together the lines (each year), the less change over time and the more intact the phenology curve. Any change in the phenology curve can be due to either a shift in green-up and/or senescence or a disturbance to the growing cycle. Lines representing years below the average (bold black line) show stressed, defoliated, or dead/dying plant communities.

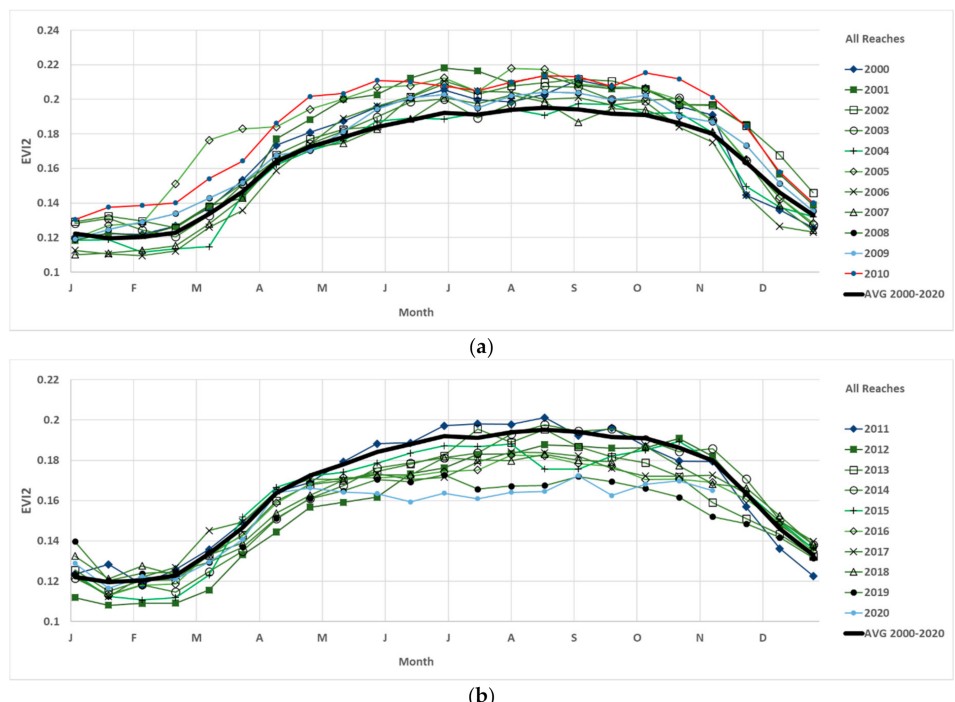

**Figure 5.** EVI2 time-series information for All reaches for the full year with months on the *x*-axis (J-D), for individual years 2000–2010 (**a**) and 2011–2020 (**b**), and for each decade. The bold line shows the full 21-year period, 2000–2020.

Seeing the anomalies overlaid with one another and in comparison with the 21 year mean (black line, Figure 5), the EVI2 curves with each annual dataset plotted as lines for all 21 years, is a good way to determine in which months disturbances to the riparian plant system may have occurred. The important information conveyed in the plots show that the first decade has relatively normal riparian greenness and health and water use in All reaches (and for each reach) relative to the average for the study. The second decade, aside from the year 2011, shows the opposite, where all the curves fall under the mean for the study.

EVI2 is an indicator of riparian corridor vegetation health. We demonstrate the annual trends in green vegetation by showing the weighted average EVI2 for All reaches over the last ten years by plotting the individual years 2011–2020, as well as the average 2000–2010 period (red dashes), the average 2011–2020 period (orange short dashes), and the full period 2000–2020 (black solid line) for comparison (Figure 5). Compared with the first decade (red dashes), all the individual years are less than this, indicating losses in green vegetation year after year. Furthermore, all but 2011 are less EVI2 than the average 21-year full period (black solid line). The collective loss of healthy riparian vegetation over the many months between April and December is indicated by the area under the curves. For example, decreases in magnitude of the curve in summer may be indicative of defoliation caused by the salt cedar biocontrol beetle, and the impact of not having any green canopy affects the overall annual riparian water use. It is the full-year measurement of the area under the ET curve, importantly derived in part from the EVI2 curve, that we use to determine the full-year amount of water use, which is determined by the phenology assessment metric.

The last VI figure shows average EVI2 for seven time-periods using time-series information for the full year for the average weighted total riparian area, All reaches (Figure 6). Individual reaches are shown in Appendix A, Figure 1a–e. The periods shown are the average 2000–2020 period (black solid line), the average EVI2 over the first decade 2000–2010 (red solid line), the average EVI2 over the last decade 2011–2020 (red dotted line), plus four additional approximately five-year periods: 2000–2005 (blue line), 2006–2010 (green line), 2010–2015 (red dashed line), and 2016–2020 (green dashed line). The first decade has the highest EVI2, indicating it was also the greenest in terms of riparian vegetation health.

The last decade, 2011–2020 has lower EVI2 with the most recent last five years having the very lowest EVI2, indicating tremendous loss in terms of the ecosystem greenness over the last five-year and ten-year periods. Despite recent on-the-ground research in the upper basin rivers, which found salt cedar-invaded systems to be stable after an 8-year study ending in 2018 [118], these losses in the LCR are likely not only in part due to biocontrol, which undergoes periods of defoliation that may reduce EVI2, but also is more likely the result of a decrease in greenness in other species, due to recent very hot maximum temperatures and lack of rainfall. Our data support the stability of these ecosystems up to the recent few years, where we observe large declines across the riparian community. The average data over 21 years are nearly the same as the five-year period between 2010 and 2015.

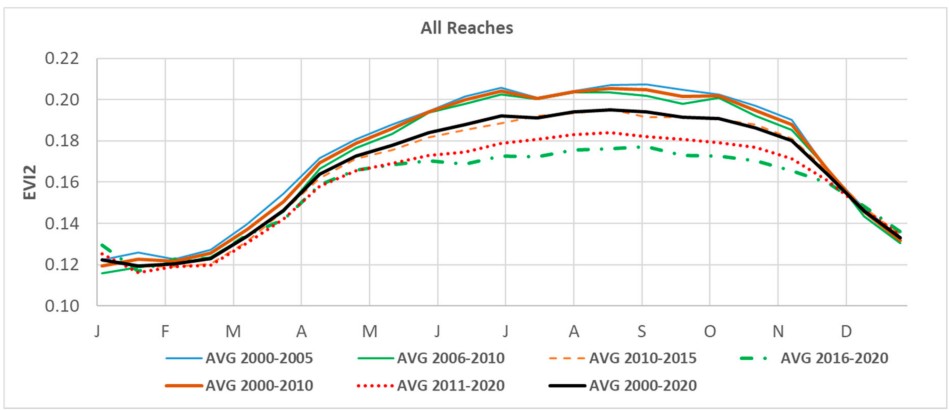

**Figure 6.** EVI2 time-series information for riparian vegetation on the Lower Colorado River for All reaches for the full year with months on the *x*-axis (J-D) for seven groups of years between 2000 and 2020 to demonstrate spatiotemporal changes across the first decade, 2000–2010 (red solid line) and the second decade, 2011–2020, (red dashed line), as well as the average for the full period, 2000–2020 (black solid line), plus four additional approximately five-year periods, 2000–2005 (blue line), 2006–2010 (green line), 2010–2015 (red dashed line), and 2016–2020 (green dashed line).

The EVI2 is plotted for the full year, January 1 to December 31, for seven groups of years between 2000 and 2020 to demonstrate spatiotemporal changes in All reaches (Figure 6). Shown for comparison, the data are averaged across the first decade, 2000–2010 (red solid line), and the second decade, 2011–2020 (red dashed line), as well as the average for the full period, 2000–2020 (black solid line), plus four additional approximately five-year periods: 2000–2005 (blue line), 2006–2010 (green line), 2010–2015 (red dashed line), and 2016–2020 (green dashed line).

These same plots for the five reaches, R3 to R7, are provided in Appendix A (Figure 2a–e). In Appendix A, Figure 2a for R3, the last five-year and ten-year periods are much lower than the full 21-year period and other period averages. The average data over 21 years is practically the same as the five-year period between 2010 and 2015. In Appendix A, Figure 2b, R4 and Figure 2c, R5, the wide range of EVI2 values by groups of years is not present; the data are very tight indicating similar EVI2 data and time-series trends across all months and across all groups of years. In Appendix A, Figure 2d, R6, the first decade has very high EVI2, depicting much green foliage in this part of the riparian corridor. As with other reaches, the last decade indicates a significant loss of green foliage in R6. The last two five-year periods also show these losses, with the last five years, 2016–2020, showing drastic losses in riparian health. In Appendix A, Figure 2e, R7, the data are very tight, indicating similar EVI2 data and time-series trends across all months and across all groups of years, but the last decade and the last two five-year periods are lower EVI2 than the averages for the first decade and the full 21-year period.

### 3.2. ET Daily from VI-Based Time-Series from 2000 to 2020

ET (mmd$^{-1}$) was calculated following Equation 4, which used EVI2 and potential ET (ETo) from three nearby weather stations. We show the riparian ET for the full year in five reaches, R3 to R7, as time-series data using Landsat imagery collected ~16 days from 2000 to 2020 for the riparian zone (Figure 7).

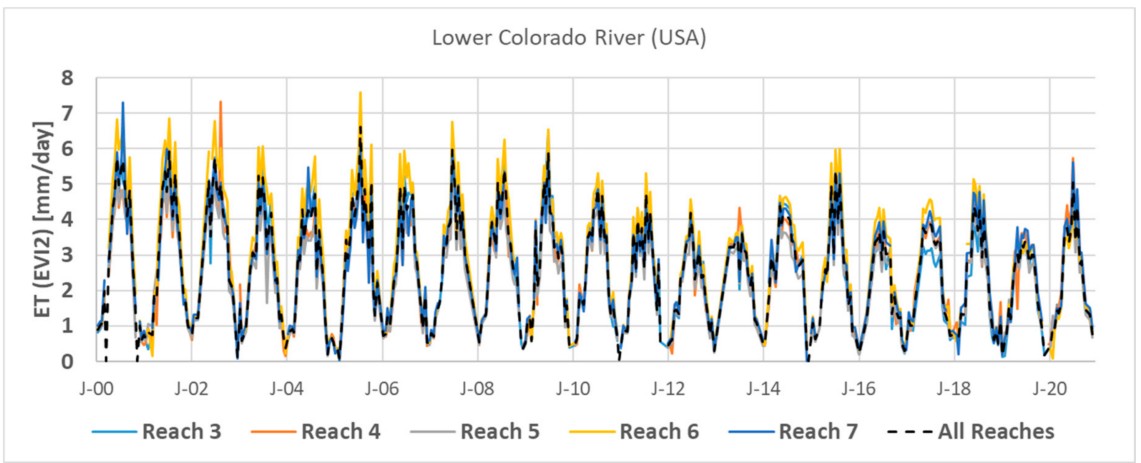

**Figure 7.** Daily water use (mmd$^{-1}$) on the Lower Colorado River for 21 years as full-year, time-series data from 2000 to 2020 for the riparian zone five reaches, R3 to R7, shown by colored lines (R3 = light blue, R4 = red, R5 = grey, R6 = yellow, R7 = dark blue), and the average daily ET for All reaches is shown as the black dashed line.

We provide values that correspond with Figure 7 for ET (mmd$^{-1}$) in the peak growing season from 2000–2020 for each reach and the total weighted average in All reaches (Appendix A, Table A5). In Appendix A, Table A5, the absolute value of ET over all reaches (All) dropped over 1 mmd$^{-1}$ from 4.34 mmd$^{-1}$ in 2000 to 3.25 mmd$^{-1}$ in 2020. Appendix A, Table A6 shows the difference in daily ET values, demonstrated as positive or negative (−) change, as measured from one year to the next. The greatest differences across all reaches were observed between 2015 and 2016, possibly due to the initial wave of biocontrol defoliation, where negative change persisted with average daily ET ranging between a decrease of 1.44 mmd$^{-1}$ and 1.86 mmd$^{-1}$, and All reaches was defined by a decrease in average daily ET of 1.62 mmd$^{-1}$.

The percent change from year-to-year using the average daily ET (mmd$^{-1}$) for the peak growing season (May 1 to October 30) for each of the 21 years in the study is provided (Figure 8). We also provide corresponding values of the percent change between years in Appendix A Table A7. ET change is either positive or negative (−) and the data are shown for each reach and the weighted average in All reaches (Appendix A, Table A7).

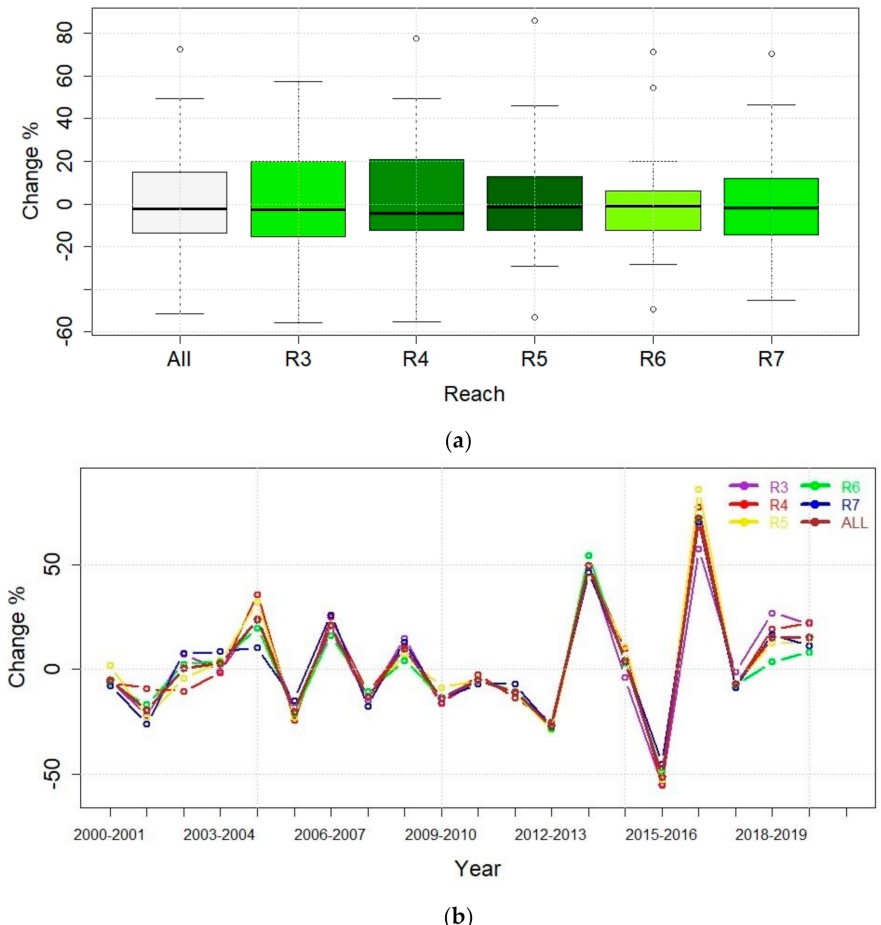

**Figure 8.** The percent change in daily water use for the peak growing season for five reaches and the entire riparian corridor (All) is shown using plots with a horizontal line, box, and whisker ends that indicate the median, 25th and 75th percentiles, and the 10th and 90th percentiles, respectively, and the data points outside this range are shown by dots (**a**). The rate of change in ET from year-to-year from 2000 to 2020 for each reach (**b**).

A boxplot shows the range of daily ET changes across reaches individually and together as "All" with the darker the box, the larger the range of change (Figure 8a). For example, R5 shows the largest range of change, followed by R4. Following this the rate of change over years is provided (Figure 8b). Using the values in Appendix A, Table A7, as an example, R5 shows the largest range of change in daily ET ($-53$% to 86%, followed by R4 ($-55$% to 77%). Figure 8 shows that the largest increase in ET was 2004–2005 with 160.8 mmyr$^{-1}$, followed by 2007–2008 with 152.2 mmyr$^{-1}$. The largest loss in ET was 189.5 mmyr$^{-1}$ for the year 2002–2003, followed by the second largest loss in ET, 134 mmyr$^{-1}$ for the year 2015–2016. The year-to-year percent change shows a decrease of as much as $-51.12$% between 2015 and 2016; however, the following year, 2016–2017, shows as much as a 69.53% increase in average daily ET.

Too much variance in the year-to-year change data makes it difficult to quantify the overall ecosystem change trends. Therefore, we employed seven groups of years for evaluation of the ET metric. We then averaged the daily water use for seven groups of years in the study (2000–2020, 2000–2010, 2011–2020, 2000–2005, 2006–2010, 2011–2015, and 2016–2020) and reported these averages for each reach and the weighted average for All reaches (Table 2).

**Table 2.** ET (mmd$^{-1}$) averaged across seven groups of years, 2000–2020, 2000–2010, 2011–2020, 2000–2005, 2006–2010, 2011–2015, and 2016–2020, for the riparian vegetation on the Lower Colorado River.

| ET (mmd$^{-1}$) | R3 | R4 | R5 | R6 | R7 | All |
|---|---|---|---|---|---|---|
| 2000–2020 | 3.34 | 3.21 | 2.94 | 3.71 | 3.29 | 3.24 |
| 2000–2010 | 3.94 | 3.61 | 3.33 | 4.35 | 3.69 | 3.72 |
| 2011–2020 | 2.68 | 2.77 | 2.52 | 3.00 | 2.85 | 2.71 |
| 2000–2005 | 4.00 | 3.65 | 3.40 | 4.51 | 3.72 | 3.78 |
| 2006–2010 | 3.87 | 3.57 | 3.25 | 4.14 | 3.65 | 3.66 |
| 2011–2015 | 2.99 | 2.89 | 2.60 | 3.21 | 2.88 | 2.87 |
| 2016–2020 | 2.37 | 2.65 | 2.43 | 2.80 | 2.83 | 2.54 |

These average daily ET values for each reach and the entire riparian corridor (All) (Table 2) are as follows: 3.24 mmd$^{-1}$ for the full study period, 2000–2020; 3.72 mmd$^{-1}$ for the first decade, a full 1 mmd$^{-1}$ less; 2.71 mmd$^{-1}$ for the most recent decade; and a steady decrease for the past four approximately five-year periods (see Figure 9). These periods dropped from 3.78 mmd$^{-1}$ (2000–2005) to 3.66 mmd$^{-1}$ (2006–2010) to 2.87 mmd$^{-1}$ (2011–2015) to 2.54 mmd$^{-1}$ (2016–2020). Importantly, for all the riparian vegetation over these four periods of time since 2000, the average daily ET values were all under 4.5 mmd$^{-1}$. We plotted daily ET for the four groups of approximately five-year periods in a bar graph for individual reaches and All reaches (Figure 9).

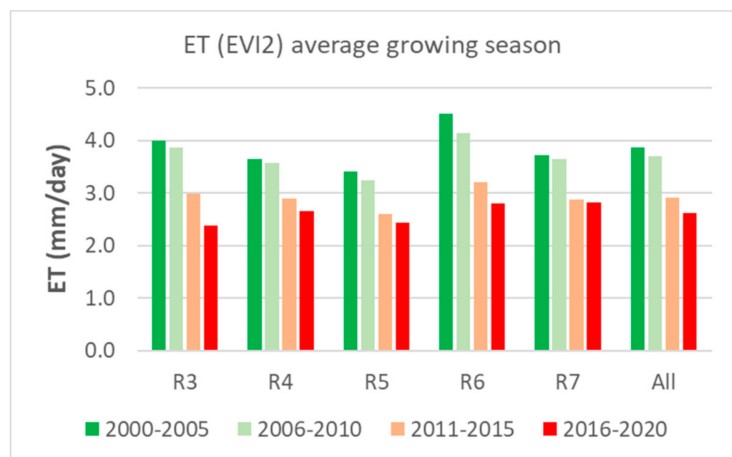

**Figure 9.** ET (mmd$^{-1}$) averaged across four groups of years in ~5-year periods (2000–2005 = green, 2006–2010 = light green, 2011–2015 = peach, 2016–2020 = red) for five reaches and the weighted average of All reaches along the Lower Colorado River.

Finally, we quantified the difference in the average daily water use using ET for the peak growing season for grouped years to be compared in two periods. We used the following temporal groups: 2000–2010, 2011–2020, 2000–2005, 2006–2010, 2011–2015, and 2016–2020. We then subtracted one group of years from another group as demonstrated in the first and second columns, Periods 1 and 2, to determine percent change between these groups (Table 3). The ET percent change between periods is shown for each reach and the weighted average in All reaches.

**Table 3.** ET (mmd$^{-1}$) percentage change by period, 5-year or 10-year, for each reach and All reaches with all riparian vegetation water use along the Lower Colorado River being negative (−) change and demonstrating decreases in ET between the first and second periods.

| ET (mmd$^{-1}$) | | | | | | | |
|---|---|---|---|---|---|---|---|
| Period 1 | Period 2 | R3 | R4 | R5 | R6 | R7 | ALL |
| 2000–2005 | 2006–2010 | −3.23 | −2.00 | −4.48 | −8.26 | −1.96 | −2.95 |
| 2006–2010 | 2011–2015 | −22.77 | −19.07 | −20.11 | −22.58 | −21.17 | −21.63 |
| 2011–2015 | 2016–2020 | −20.55 | −8.38 | −6.34 | −12.74 | −1.68 | −11.39 |
| 2000–2005 | 2016–2020 | −40.63 | −27.34 | −28.53 | −38.03 | −24.01 | −32.61 |
| 2000–2010 | 2011–2020 | −31.95 | −23.32 | −24.57 | −30.91 | −22.67 | −27.30 |

Table 3 shows the percent change in daily ET between these six periods, the two ten-year periods, and the four approximately five-year periods, with all change being negative (−). When comparing 2000–2005 to 2006–2010, the least amount of decrease in water use, 0.12 mmd$^{-1}$, occurred in the daily ET metric, a decrease of −2.95% for All reaches. The comparison between the first and last ca. five-year periods, 2000–2005 and 2016–2020, showed the largest decrease in average daily water use, a decrease of 1.24 mmd$^{-1}$, or −32.61% as measure by the daily ET metric. Another loss in the average peak growing season daily ET was 0.79 mmd$^{-1}$ when the years 2006–2010 were compared with 2011–2015, a decrease of −21.63%. After this large percent change decrease, the years 2011–2015 compared with 2016–2020, another but smaller decrease in water use was estimated to be 0.33 mmd$^{-1}$, or a change of −11.39%. The comparison between the two decades, first and last, also yielded large declines in daily water use, an average decrease of over 1 mmd$^{-1}$, or a change of −27.30%, as measured by the daily ET metric.

*3.3. ET Annually, PAM ET, from VI-Based Time-Series from 2000 to 2020*

We provide annual water use data as PAM ET (mmyr$^{-1}$) for the LCR riparian area. The PAM ET provides ET on an annualized basis because it uses the full-year EVI plotted as time-series data and sums the area under the phenology curve. The annualized ET captures growth of the natural vegetation that happens outside of the peak growing season. Average peak growing season ET is mainly used when estimating water use of cultivated crops and has also been applied to riparian woodlands.

PAM ET (mmyr$^{-1}$) values are plotted as a bar graph for the last two decades, 2000–2020 for the five reaches, R3 to R7, and the entire area, All (Figure 10). Each reach is also plotted in Appendix A, Figure 2a–e, and the corresponding values are in Appendix A, Table A7. Because PAM ET is based on the area under the EVI2 curves (see Figure 5), the decrease in annual curve magnitude has a significant impact on the annual water use totals (Figure 10). These data also show that the very low years of 2019 and 2020 in terms of EVI2 greenness do not impact annualized water use when the ETo and temperatures are higher than normal. We can use 2005 as an example of early green-up compared with the other years, and this may be due to high precipitation in 2004 (Figure 2); but we note that the early green-up trends appear in R3, R4, and R5, but not in R6 and R7, which could be due to the presence of MSCP and/or other restoration activities.

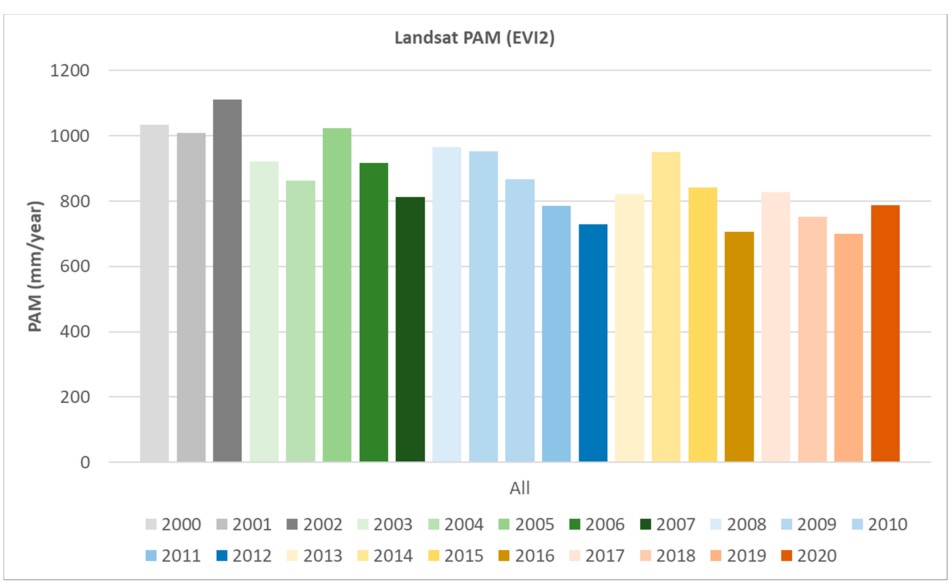

**Figure 10.** PAM ET (mmyr$^{-1}$) is shown for each of the years 2000–2020 for five reaches and the weighted average of All reaches along the Lower Colorado River.

The annual ET cycles show oscillations that occur in declines of three years and one of five years between 2008 and 2012 (Figure 10). In recent years, we see that the data supports the data in the delta [14] and 2020 is increasing rather than continuing the declining trend observed between 2000 and 2019. The ET algorithm relies not only on vegetation greenness but also on weather station data (see Figure 2). Vegetation greenness using EVI2 was the lowest ever recorded in 2020 (Figure 5); however, ETo was high in 2020, having increased to 275 mm from 250 mm in 2019, as temperature was higher by 1–3 degrees C, and there was no rainfall. This is likely the reason for the increase in PAM ET in 2020 (Figure 10).

Several things are important to note: (i) the 2013 data are a mix of Landsat 7 (ETM+) and Landsat 8 (OLI); (ii) the salt cedar biocontrol beetle was first observed on the LCR in 2014 but reach-wide impacts from defoliation events take several years to make a measurable difference in ET; (iii) PAM ET measured over the full year, not just the peak growing season, shows 2016 is consistently the lowest ET across reaches; and (iv) the full year PAM ET is on average a little less than 800 mmyr$^{-1}$ with declining trends in four 3-year periods, then followed by an increase in ET, i.e., 2002–2004, 2005–2007, 2010–2013, 2014–2016, and 2017–2019. There is one 5-year period from 2008–2012 before the increase in ET the following year, 2013.

The absolute values of the full-year water use, PAM ET (mmyr$^{-1}$), are shown for each of the 21 years from 2000 to 2020 for five reaches and the weighted average of All reaches along the LCR (Table 4). The initial 10-year period was in fact greener with higher values of PAM ET than the last decade as shown in Table 4. The All column in Table 4 shows values that began at 1034 mmyr$^{-1}$ and end below 700 mmyr$^{-1}$ until 2020, when an increase occurred and is due to higher ETo and temperatures. Because of the different composition of species and cover reach by reach, our reach level results are also shown. A plot of PAM ET (mmyr$^{-1}$) is shown for each river reach in Supplemental Information (Appendix A). The PAM ET for R3 is in Appendix A, Figure 2a, R4 is A2b, R5 is A2c, R6 is A2d, and R7 is A2e.

**Table 4.** PAM ET (mmyr$^{-1}$), total annualized water use, for each of the 21 years from 2000 to 2020 for five reaches and the weighted average of All reaches along the Lower Colorado River.

| PAM ET (mmyr$^{-1}$) | | | | | | |
|---|---|---|---|---|---|---|
| Year | Reach 3 | Reach 4 | Reach 5 | Reach 6 | Reach 7 | All |
| 2000 | 1047.86 | 947.36 | 966.71 | 1267.67 | 1134.15 | 1034.20 |
| 2001 | 962.29 | 874.21 | 923.11 | 1150.08 | 956.01 | 1008.94 |
| 2002 | 1098.06 | 1101.25 | 1009.92 | 1225.50 | 1122.35 | 1111.37 |
| 2003 | 969.99 | 880.14 | 805.11 | 1084.52 | 893.20 | 921.85 |
| 2004 | 873.71 | 775.97 | 747.23 | 1026.19 | 882.01 | 861.93 |
| 2005 | 1053.12 | 1025.40 | 954.06 | 1179.62 | 927.96 | 1022.69 |
| 2006 | 952.14 | 843.90 | 811.53 | 1058.09 | 909.83 | 916.76 |
| 2007 | 837.06 | 705.19 | 706.20 | 875.76 | 809.78 | 813.10 |
| 2008 | 980.38 | 880.10 | 861.66 | 1089.72 | 829.19 | 965.29 |
| 2009 | 1018.61 | 891.10 | 846.86 | 988.29 | 906.94 | 951.61 |
| 2010 | 888.79 | 827.70 | 806.11 | 917.30 | 865.07 | 865.82 |
| 2011 | 814.59 | 782.00 | 714.74 | 859.83 | 770.47 | 785.95 |
| 2012 | 736.66 | 701.00 | 665.41 | 809.38 | 751.10 | 729.00 |
| 2013 | 829.52 | 853.91 | 758.27 | 854.13 | 810.30 | 821.23 |
| 2014 | 1000.37 | 924.24 | 842.62 | 1052.28 | 958.79 | 949.79 |
| 2015 | 806.47 | 802.45 | 739.64 | 931.37 | 827.11 | 840.60 |
| 2016 | 648.78 | 697.14 | 642.47 | 827.73 | 747.84 | 706.56 |
| 2017 | 706.63 | 856.94 | 821.99 | 958.52 | 861.63 | 827.85 |
| 2018 | 656.23 | 760.86 | 699.48 | 841.92 | 826.52 | 751.67 |
| 2019 | 656.37 | 721.95 | 677.70 | 731.24 | 765.48 | 699.83 |
| 2020 | 839.29 | 906.75 | 768.14 | 796.10 | 855.06 | 787.94 |

　　For year-to-year changes over 21 years, the percent change in PAM ET is demonstrated with positive and negative (−) box and line plots for each of the five reaches and the weighted average of All reaches along the LCR (Figure 11). The percent change in PAM ET is shown by box plots with the range of change highlighted by the darker the box (11a, top), and the lines indicate the rate of change between years 2000 and 2019 (11b, bottom). For example, R4 shows the largest range of change, followed by R5. Using the difference values in Appendix A, Table A8, as an example, R4 shows the largest range of change in PAM ET, from −221.1 mmyr$^{-1}$ to 249.4 mmyr$^{-1}$ a range between −220% and 250%, followed by R5, from −207.8 mmyr$^{-1}$ to 206.8 mmyr$^{-1}$, a range between −200% and 200%. That the percent change in PAM ET year-to-year is as much as 200% in the positive direction and 200% in the negative direction is a key finding. This large amount of change was somewhat steady between 2007 and 2013 and corresponds to the period of lower ETo, although the three stations of weather data do not seem to have significant variation. Difference in PAM ET (mmyr$^{-1}$) between years for 21 years is provided in Appendix A Table A8. PAM ET (mmyr$^{-1}$) change is either positive or negative and the data are shown for each reach and the weighted average in All reaches (Appendix A, Table A8).

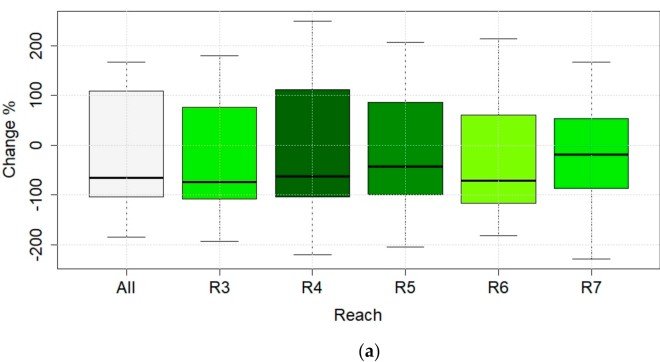

(**a**)

**Figure 11.** *Cont.*

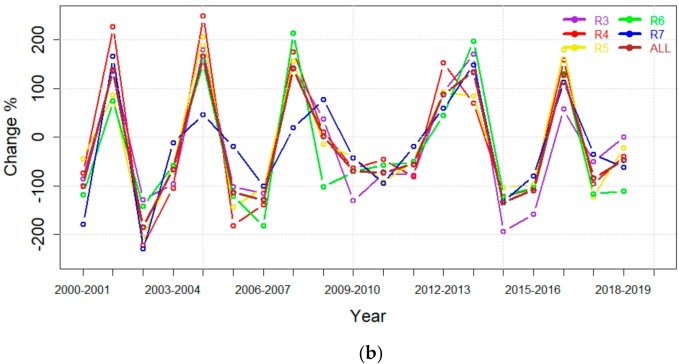

(**b**)

**Figure 11.** The percent change in PAM ET (mmyr$^{-1}$) between reaches and from year-to-year on the LCR. The horizontal line, box, and whisker ends indicate the median, 25th and 75th percentiles, and 10th and 90th percentiles, respectively, and data points outside this range are shown by dots (**a**); rate of change in PAM ET is shown with lines for each reach annually during 2000–2019 (**b**).

The PAM ET (mmyr$^{-1}$) is provided for seven groups of years is shown for the periods 2000–2020, 2000–2010, 2011–2020, 2000–2005, 2006–2010, 2011–2015, and 2016–2020 for five reaches and the weighted average of All reaches along the LCR (Table 5). Overall, the PAM ET was 870.76 mmyr$^{-1}$ over the past two decades. The first decade, 2000–2010, the PAM ET averaged 952.14 mmyr$^{-1}$, and the first years, 2000–2005, PAM ET averaged 993.50 mmyr$^{-1}$. The water use estimates dropped in the second, more recent decade to 781.23 mmyr$^{-1}$, and in the last years, 2016–2020, PAM ET averaged 737.15 mmyr$^{-1}$. A comparison of the average changes between the first three five-year periods shows decreases in PAM ET of 90.99 mmyr$^{-1}$, 77.20 mmyr$^{-1}$, and 88.16 mmyr$^{-1}$.

**Table 5.** The average annual water use, PAM ET (mmyr$^{-1}$), for seven groups of years from 2000–2020 for five reaches and the weighted average of All reaches along the Lower Colorado River.

| PAM ET (mmyr$^{-1}$) | R3 | R4 | R5 | R6 | R7 | All |
|---|---|---|---|---|---|---|
| 2000–2020 | 866.38 | 836.89 | 794.22 | 974.30 | 872.44 | 870.76 |
| 2000–2010 | 971.09 | 886.57 | 858.04 | 1078.43 | 930.59 | 952.14 |
| 2011–2020 | 751.20 | 782.24 | 724.00 | 859.76 | 808.47 | 781.23 |
| 2000–2005 | 1000.84 | 934.06 | 901.02 | 1155.60 | 985.95 | 993.50 |
| 2006–2010 | 935.40 | 829.60 | 806.47 | 985.83 | 864.16 | 902.51 |
| 2011–2015 | 837.52 | 812.72 | 744.14 | 901.40 | 823.55 | 825.31 |
| 2016–2020 | 664.87 | 751.77 | 703.87 | 818.13 | 793.39 | 737.15 |

Percent change in PAM ET (mmyr$^{-1}$) is shown between two periods, Periods 1 and 2, with comparisons made for groups of years 2000–2010, 2011–2020, 2000–2005, 2006–2010, 2011–2015, and 2016–2020, and for five reaches and the weighted average of All reaches along the LCR (Table 6). The percent change between these groups of years for PAM ET is all negative change (-) and shows large, significant decreases in the estimated annualized water use (Table 6).

**Table 6.** The percent change in the total annualized water use, PAM ET (mmyr$^{-1}$) between two periods, Period 1 compared to Period 2, which are comprised of groups of years 2000–2010, 2011–2020, 2000–2005, 2006–2010, 2011–2015, and 2016–2020, and for five reaches and the weighted average of All reaches along the Lower Colorado River. Note: all the calculated percent change is negative (−) and indicate decreases in PAM ET.

| PAM ET (mmyr$^{-1}$) | PAM ET (mmyr$^{-1}$) | | | | | | |
|---|---|---|---|---|---|---|---|
| Period 1 | Period 2 | R3 | R4 | R5 | R6 | R7 | ALL |
| 2000–2005 | 2006–2010 | −6.54 | −11.18 | −10.49 | −14.69 | −12.35 | −9.16 |
| 2006–2010 | 2011–2015 | −10.46 | −2.03 | −7.73 | −8.56 | −4.70 | −8.55 |
| 2011–2015 | 2016–2020 | −20.61 | −7.50 | −5.41 | −9.24 | −3.66 | −10.68 |
| 2000–2005 | 2016–2020 | −33.57 | −19.52 | −21.88 | −29.20 | −19.53 | −25.80 |
| 2000–2010 | 2011–2020 | −22.64 | −11.77 | −15.62 | −20.28 | −13.12 | −17.95 |

Table 6 shows the percent change in annual PAM ET between six periods, the two ten-year periods, and the four approximately five-year periods. Percent changes between periods compared and across reaches are denoted with a negative sign (−) to indicate that all were decreases in PAM ET (Table 6). Importantly, all reaches for all compared time periods have negative change. The percent change in PAM ET between the first and last 5-year periods, 2000–2005 and 2006–2010, is equal to a −9.16% decline; between 2006–2010 and 2011–2015 PAM ET is equal to a −8.55% decline; and between 2011–2015 and 2016–2020 PAM ET is equal to a −10.68% decline. The comparison between the first and last periods, 2000–2005 and 2016–2020, shows a loss in PAM ET of 256.35 mmyr$^{-1}$ or a decrease of −25.80%. A comparison between the first and last decade also reflects these large losses. Results show a loss in PAM ET of −170.91 mmyr$^{-1}$ or a decrease of −17.95% and indicate a shift toward an unhealthy riparian corridor with less green vegetation and corresponding losses in reach level riparian ET.

*3.4. Change Maps from Two Periods Showing Two Metrics, Scaled NDVI, and ET*

Two types of map that represent spatiotemporal change (positive and negative) over two periods of time are provided. We produce change maps using the metrics of a scaled NDVI (referred to as NDVI*) and ET that was parameterized with EVI2. The primary reason for using the NDVI* scaling method is that it takes into account changing illumination conditions scene-to-scene. NDVI* is an additional correction to the NDVI, used here because it is widely known, to account for illumination conditions. In other words, it is a processing correction step to improve the usefulness of NDVI. NDVI* is not used to parameterize the ET model. The first set of maps compares the first to the last decade (Figure 12) and is of the NDVI* (Figure 12a, left) and ET using EVI2 (Figure 12b, right). The second set of maps represent change between the first 5-year period and the last 5-year period (Figure 13) with both NDVI* (Figure 13a, left) and ET using EVI2 (Figure 13b, right) change indicating areas of increasing and decreasing greenness. Each map has a histogram of the number of pixels in the map, and the distribution of positive and negative changes is displayed. Due to the long, narrow nature of the riparian corridor region, we show enlarged maps for the five reaches we studied, R3 to R7. Areas such as R4 had very few pixels representing riparian plants due to the physical geography of the reach.

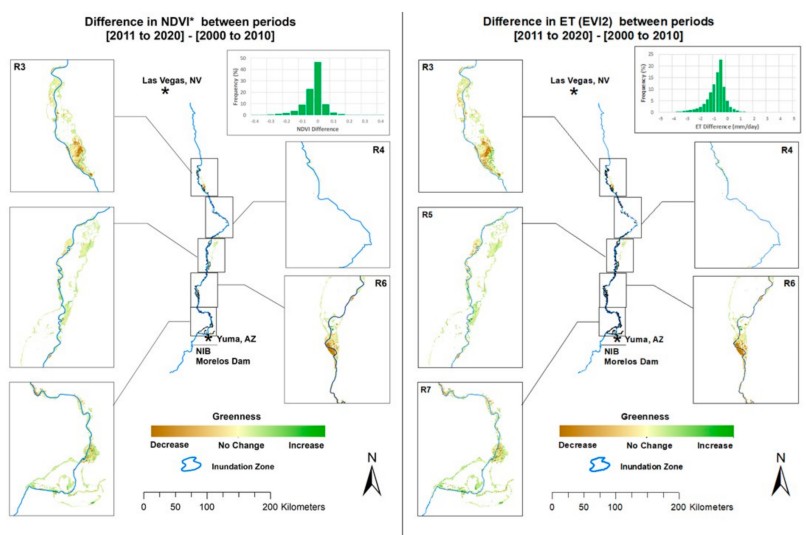

**Figure 12.** Maps of five reaches along the Lower Colorado River showing change in greenness using scaled NDVI (NDVI*) (12a, left) and ET (12b, right) for the compared periods of the first (2000 to 2010) and last decade (2011 to 2020). The histogram (upper right) shows the frequency of pixels. Values less than zero indicate a loss of vegetation greenness and water use. Also, each map figure contains five enlarged inset areas, boxes from north to south showing riparian vegetation in R3, R4, R5, R6, and R7. The legend shows decreases (brown) and increases (green) depicting how these two metrics, NDVI* and ET, have changed over the respective time periods.

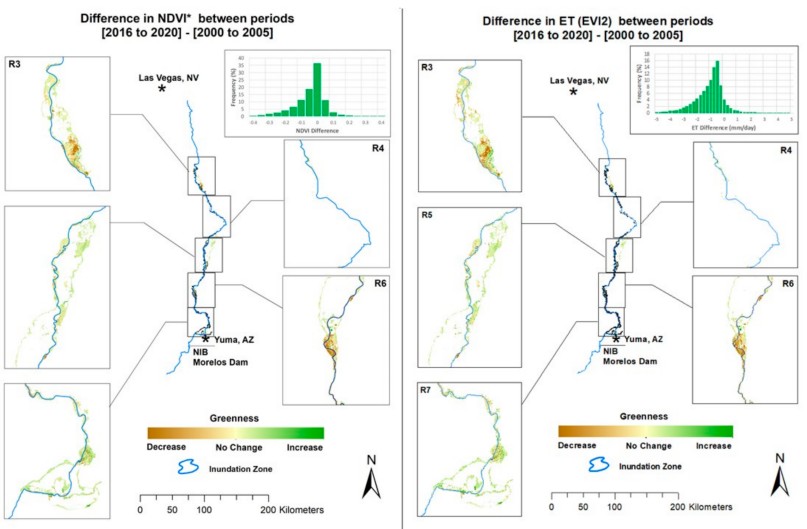

**Figure 13.** Maps of five reaches along the Lower Colorado River showing change in greenness using scaled NDVI (NDVI*) (13a, left) and ET (13b, right) for the compared periods of the first (2000 to 2005) and last decade (2016 to 2020). The histogram (upper right) shows the frequency of pixels. Values less than zero indicate a loss of vegetation greenness and water use. Furthermore, each map figure contains five enlarged inset areas, boxes from north to south showing riparian vegetation in R3, R4, R5, R6, and R7. The legend shows decreases (brown) and increases (green) depicting how these two metrics, NDVI* and ET, have changed over the respective time periods.

In these sets of maps showing change in NDVI* and ET, we compare the first and last decades of our study (Figure 12), as well as the first and last 5-year periods (Figure 13). In individual reaches, boxes that are enlarged show R3 to R7, allowing the reader to observe changes to the riparian cover in these areas. Historically, R3 and R6 have been heavily dominated by salt cedar plants [19,20,30,45] and have been validated with on-the-ground observations [15,28,73,120]. R5 and R7 have salt cedar stands but have not been dominated by vast, monotypic swaths of salt cedar like those in R3 and R6, which are a few kilometers across in some areas. All reaches show negative trends, declines in greenness, ET, and PAM ET. R5 and R7 are not showing these dramatic declines in greenness and water use seen in R3 and R6; however, these reaches do have significant losses in vegetation greenness cover as seen in the bar graph of daily ET (Figure 9). The percent change in PAM ET declines between the first and last 5-year periods and is smallest for R4, R7, and R5, but is still in the order of −20%. R3 shows declines of −34% and R6 declines of −29%, thus both reaches have considerably larger negative change than R5 and R7 (Figure 10). In the box plots, PAM ET percent change is negative for each reach, but less so in R5 and R7. Figure 12 depicts the first decade, 2000–2010, in comparison to the last, most recent decade, 2011–2020. The two reaches, R3 and R6, have displayed the greatest decline or negative change in greenness and water use. Furthermore, Reach 5 (Appendix A, Figure 1c) and Reach 7 (Figure 1e) show the smallest ranges of annual cycle data; Reach 3 (Figure 1a) and Reach 6 (Figure 1d) show the widest ranges, with decreases prominent in the last few years (see Appendix A Figure 1a-e). These results are in line with findings for other parts of the Colorado River basin where salt cedar cover across sites declined on average of −40–50% in response to the beetle, but then recovered [8,23,25,47,82]. However, we are now seeing that when we compare the initial five years and the last five years (Figure 13) with the last decade (Figure 12), the reaches dominated by salt cedar have undergone decreases in vegetation and show large declines in its water use due to the eradication of plant cover that has not been replaced by regrown salt cedar or native species. From the change maps that use NDVI* and ET from EVI2, for the two periods of change (Figures 12 and 13), we see much darker browns where salt cedar exists at Topock Marsh and Cibola NWR. In R3 and R6, we know these darker brown patches, indicating maximum browning or maximum negative change in greenness, are where salt cedar has been validated on the ground. In R5 and R7, the color indicates that there has been less browning change in comparison to the color scheme in the other reaches. Other areas near the international border and the city of Yuma, Arizona, such as R5 and R7, have experienced more subtle declines in riparian health. There has been active restoration in R7 and these small-scale and healthy areas may have positively impacted the this reach.

## 4. Discussion

Riparian systems in drylands occupy a sliver of land area yet provide rich and diverse ecosystems that are essential habitat for migrating and breeding birds, herpetofauna, and other wildlife, and thus their protection and restoration have been one of the top environmental management priorities [62,133]. These systems are also among the most vulnerable to climate change, particularly with increasing societal needs for water, its contested demand, and declining supply [51,52,59,78]. Most dryland rivers have been altered by dams and diversions [59], linkages between the natural flow regime, streamflow, blue water, and riparian health [57,58,64]; fire; salinity; non-native land clearing [60,61,63]; and receive considerable management action, resources, and concern [57,59,62,133–135]. Myriad management options are being considered and implemented to restore riparian ecosystems [120,136] despite pressures from lack of resources, environmental flows, drought, and increasing temperatures.

### 4.1. Remotely Sensed Findings and Their Implications

Even a decade ago, there was a lack of data and knowledge regarding trends in riparian habitats, despite riparian ecosystems being known to be one of the most utilized landscapes by a variety of biota and providing the widest range of conservation benefits and ecosystem services [62]. This study is among just a few in which information on riparian corridor greenness, ecosystem health, and its water use have been produced. This lack of information on reach-level water use of riparian woodlands is because this plant community has been neglected relative to water budgeting. This study was conducted as a follow on to a research chapter for the International Boundary Water Commission (IBWC) report on the Water Treaty Minutes 319 and 323 that assessed the riparian ecosystem health and water use in the delta of the Colorado [14]. In that study in the delta, we compared two scales (Landsat and MODIS); two indices for greenness, EVIs (EVI and EVI2); and finally, two estimates of ET (the daily ET with peak growing season dates and the annual PAM ET). We first developed and applied the annualized water use (PAM ET) due to riparian vegetation growing in this hot and arid region entirely south of Yuma, AZ. This study on the U.S. side of the LCR is an important study involving common methods and an overlap in some objectives. For this study, the methods were different in that we excluded MODIS as results were statistically comparable with Landsat [14], and thus we chose to use the higher resolution Landsat only. The methods here also rely entirely on EVI2, the simpler version to estimate greenness. Our data showed an $r^2 = 0.99$ between EVI and EVI2. Moreover, the methods in this study report both the daily ET and annual PAM ET for this region that allows for most of the riparian woodlands to grow year-round. Our results over the past two decades using Landsat time series data on the LCR riparian corridor provide the first comprehensive look at biocontrol impacts on the non-native species salt cedar health and its water use on the LCR. This study also shows timely new information in response to the pressures and concerns over available water and drought factors [136,137]. This important time series data regarding the health of this ecosystem and its water use is necessary for managers to understand the consequences of drought and biocontrol and to plan for disturbances in the future.

### 4.2. Strengths and Limitations

Remotely sensed measurements of ET at regional and global scales primarily capture agricultural water use, not ET from riparian woodlands. Given that the majority of southwestern U.S. drylands' in-stream water is allocated for agricultural use, the primary methods for estimating ET were developed, measured, and validated for crops, and not for natural riparian vegetation until approximately 1995. In 1995, the Lower Colorado River Accounting System (LCRAS) requested riparian plant species input for their water accounting system and riparian ET has now been reported by Reclamation every year since 2010 [134]. Because the contribution of water use from the riparian system has been neglected in water balance studies, our estimates of daily riparian ET are a critically important contribution. This study provides the first study of riparian health and water use for the riparian zone of the LCR between Hoover and Morelos Dams, expanding on small scale reach-level data over a few years, to now include four groups of ~5 years for comparisons to be made over periods of time over the last 21 years. The strength of our research is that is uses vegetation index-based ET to derive both daily and annual water use in dryland riparian river reaches and is only the second contribution to use PAM ET to assess the full year changes in riparian species and wide area composition.

The loss of healthy riparian vegetation in the last five years is likely due to large swaths of salt cedar being defoliated by the biocontrol beetle that began spreading south of Lake Mead in that same time period [42,97,105]. RiversEdge West has been tracking biocontrol using salt cedar beetle distribution maps [97]. Their maps show no beetles on the LCR south of Lake Mead in our study area in 2013, but show beetle presence in 2014 (https://riversedgewest.org/sites/default/files/images/Tamarisk_Coalition_2013_TLB_Distribution_Map_Final.pdf, accessed on 24 February 2021). Recent research also

shows the impact of defoliation from the salt cedar beetle on water use in nearby areas [136–139]. The years between 2014 and 2020 with active beetles and high levels of defoliation have contributed to our results of declining riparian greenness and water use.

The year-to-year and groups of years that were compared to determine changes with a greater number of observations demonstrate how the system has been undergoing negative changes and indicate a loss in riparian community greenness and its corresponding water use. The largest change was observed between the first and last 5-year periods and was also significant between the first and last decades. We believe the onset of the salt cedar beetles in 2014 is detectable and evident by the repeated cycles of defoliation and captured in the Landsat EVI2 time series data and summarized in the ET and PAM ET percent change tables. On a reach-by-reach basis, R3 and R6 have some large patches of monotypic salt cedar that in recent years have been defoliated. This appears to be the main reason these reaches have undergone the larger declines in our metrics, as is shown in figures and tabular data, as well as in our spatiotemporal change maps of greenness and water use. Primarily in 2005, there were observed shifts in green up start dates for which we cannot conclusively know the cause, but suspect it is due to a particular species or vegetation community that greened up earlier than the other species in the reach. We believe that if salt cedar were more widespread in the LCR reaches, then we would see more significant declines in greenness and water use data. Similarly, with larger patches of active restoration, greenness at the reach level would improve ecosystem health.

One of the strengths of this research is that we attempt to answer the cause-and-effect questions using our data findings. We had hoped that the relationship between our three ground meteorological stations, greenness, and ET findings would be stronger. From the 21 years shown in Figure 2 for precipitation, temperature, and ETo, there are no unusual data or trends. Importantly, 2020 had higher temperatures by 1–3 degrees C, higher ETo, and no rainfall, and that can be directly related to our estimates of water use which were higher in 2020. Despite the weather data significant declines in our metrics were measured. Greenness declines were large and widespread (less change in R3, R5, and R7, and more change in R3 and R6). We believe the loss of green vegetation in R3 and R6 is most likely due to biocontrol of salt cedar, particularly in the last five-year period. Daily ET percent change had means near zero or slightly negative, but the range for all reaches was large and, when grouped into periods of more data, such as the five-year bar plots, significant declines were observed on the order of −32.6%. PAM ET percent change had much lower means, on the order of decreases of −75%, with wide ranges. The PAM ET change from the first five-year period to the last fell from 1100 mmyr$^{-1}$ to 700 mmyr$^{-1}$. From studying five reaches, we observed some reach-level relationships between reaches dominated with native and dominated with non-native plant communities, and the trends in greenness and water use. The R3 and R6 reaches with areas where salt cedar exists show the largest declines in greenness and water use. The reach, R7, where active restoration is known to exist, and despite having small patches of native species in the landscape, has the least amount of change in the 21-year study. We believe that these data will be useful in making observations about greenness and water use in riparian reaches pre-beetle and post-beetle, and to other areas such as in the delta, where the beetle has not yet arrived.

There are possible shortcomings in our ET estimation methods. The data can be used for dryland riparian regions across the globe and is particularly accurate when using ground-validation such as ET data from flux towers. Analyses that require the use of ET estimates in other climates are outside the scope of this research, and it would be inappropriate to apply this optical-band ET algorithm to those studies. The VI-based ET methods should be applied primarily in regions where there are no illumination issues from haze, clouds, or shadows. We perform image correction, but this optical-based ET estimation does not perform well outside of arid and semi-arid regions. One of the limitations of this study is that it is not compared to other widely used ET products at USGS, such as the U.S. Operationalized Simplified Surface Energy Balance (SSEBop) ET and daily actual ET (ESPA), or to those many options available through Open ET (https://openetdata.org/). Another limitation of our

study is that the application of using the PAM ET in river basins outside of the arid/semi-arid landscapes with longer growing seasons for natural vegetation has not yet been tested. We expect there to be benefits of PAM ET in riparian ecosystems in other river basins with non-arid/semi-arid climate regions because using daily ET and multiplying by dates of interest does not capture the type of landscape change that unrestored, natural riparian areas undergo with sudden land clearing, re-sprouting of green leaves post-fire and post-defoliation, for example. We do recognize that in this region, the daily ET estimated from only the peak growing season when reported for the full year would produce overestimates based on existing literature [116,134,137,138]. For this reason, in this study, we report both the daily ET and the annualized PAM ET and do not use the peak growing season ET values for the full year. For future studies in which the growing season is reported, PAM ET is more accurate particularly for riparian woodlands in arid lands. In this region in the southwestern US, the growing season is much longer with temperatures in Yuma, AZ, having more than 175 days over 90 °F (https://247wallst.com/special-report/2015/08/11/americas-hottest-cities/4/). The last several years have been very hot and this excessive heat, longer number of days with high temperatures, and the extended drought could be responsible for these declines in riparian ecosystem health [85–87,135]. Thus, on the LCR, there is a clear advantage of applying the PAM ET method over the daily ET that is determined for these riparian communities from only the peak growing season.

*4.3. New Contributions and Next Steps*

This research on the LCR shows some unexpected findings and produces significant new data contributions. The initial study in the delta of the LCR was a region that declines in riparian health were expected because of diverted water at the international boundary [14], but finding similar declines in riparian ecosystem greenness on the US side on the LCR was surprising. This study provides different methods from the delta analyses. We grouped years together to compare to the year-to-year ups-and-downs to add more information by using 5-year groups. Furthermore, we did not see a logical reason for doing a statistical test between the ET averaged across four groups of years because the declines in ecosystem health are clear across these groups, but we did compare the trend of PAM ET in each of the five reaches and in the five-year periods and found no statistically significant differences because longer time periods are required to apply these methods. These trend analyses methods will be applied in future research.

To expand upon our riparian data knowledge within the LCR, we will test our ET results with SSEBop for a comparison of these two ET estimations and their means and ranges for riparian woodlands [7,36]. Furthermore, we expect to produce results using EVI2 and ET(EVI2) methods on the urban regions within the LCR to contribute new information on how these urban green spaces affect reach level ecosystem health and water use [31]. These VI-based ET findings have been applied in Australian riparian zones [22] and to urban gardens [8,9]. Our future work includes comparing the unrestored riparian corridor on the LCR to the restoration sites initiated by Reclamation's Multi-Species Conservation Plan, an important study outside the scope of this research. Comparisons of unrestored riparian areas to restoration sites in the Colorado River delta in Mexico are underway and supported by U.S. Interior resource managers and resource managers in Mexico and tribes, non-governmental organizations (NGOs), and other end users on both sides of the border.

## 5. Conclusions

We measured three metrics, EVI2, daily ET, and annual PAM ET. We found decreases in both riparian plant greenness and its water use for our study area between Hoover and Morelos Dams on the LCR over the years 2000–2020. First, EVI2 results show a decrease in greenness of −13.83% between the first and last decade (2011–2020) and a decrease of −10.46% between the first years (2000–2005) and the last years (2016–2020). Second, daily water use over the peak growing season (May 1 to October 30), as measured by ET, decreased over time. The comparison between the first and last five-year periods, 2000–

2005 and 2016–2020, showed the largest decrease in average daily water use, a decrease of 1.24 mmd$^{-1}$, a decline of $-32.61\%$. The comparison between the two decades, first and last, also yielded large declines in daily water use, an average decrease of over 1 mmd$^{-1}$, or a decline of $-27.30\%$ as measured by daily ET. Importantly, these declines come from a loss in healthy, green, riparian plant cover, and not a change in plant water use efficiency or efficient use of managed water resources. Third, annual water use as measured by PAM ET decreased over time. The comparison between the first and last periods, 2000–2005 and 2016–2020, show a decrease in ET of 256.35 mmyr$^{-1}$, a decline of $-25.80\%$. A comparison between the first and last decade also shows a decrease in ET of 170.91 mmyr$^{-1}$, a decline of $-17.95\%$. Our results indicate a shift toward an unhealthy riparian corridor with less green vegetation and corresponding water use. Our hypothesis is correct in that declines in riparian health and water use are observed on the LCR on the U.S. side of the border between Hoover and Morelos dams.

Riparian corridors in arid regions are critically important biodiversity hotspots serving myriad wildlife and providing ecosystem services. Yet over the last century and a half, these riparian areas have undergone significant transformations due to intensive land use, water mobilization for anthropogenic and environmental uses, non-native species introductions, as well as climate-related change processes. The largest declines in riparian health and water use correspond to the most recent 5-year period, 2016–2020, and are likely due in part to recent increases in maximum air temperatures, decreased precipitation, and post-beetle biocontrol which was observed in our study area beginning in 2014. Despite the riparian ecosystem showing some stability since 2000, the net result of these changes has been a reduction in native riparian vegetation, in both extent and diversity, particularly over the last 5 years.

**Author Contributions:** Conceptualization, P.L.N.; methodology, K.D., A.B.-M., C.J.J., S.C.B., and P.L.N.; software, K.D., A.B.-M., and S.C.B.; validation, P.L.N. and H.N.; formal analysis, P.L.N., A.B.-M., and S.C.B.; investigation, P.L.N.; resources, P.L.N.; data curation, A.B.-M.; writing—original draft preparation, P.L.N.; writing—review and editing, H.N.; visualization, A.B.-M., C.J.J., S.C.B., and P.L.N.; supervision, P.L.N. and K.D.; project administration, P.L.N. and K.D.; funding acquisition, P.L.N. All authors have read and agreed to the published version of the manuscript.

**Funding:** Funding was provided by the U.S. Geological Survey (USGS) under Ecosystems Invasive Species Program and the Desert Southwest Cooperative Ecosystem Studies Unit agreement #G18AC00321 (P.I. Didan) and National Aeronautics and Space Administration agreement #80NSSC18K0617 (P.I. Didan).

**Institutional Review Board Statement:** Not applicable.

**Informed Consent Statement:** Not applicable.

**Data Availability Statement:** Data generated during this study are available from the USGS ScienceBase-Catalog (Nagler et al., 2021 [140]).

**Acknowledgments:** This research is dedicated to Edward Glenn, who was my mentor for more than two decades (1998 to 2017). I was inspired to provide an analysis of the research and monitoring from 2000 to 2020 for the Lower Colorado River riparian zones because he provided the opportunity for me to collect data in this region for nearly the same years that he mentored me. We are grateful to Kul Khand for providing a USGS review and to our other reviewers for suggesting important changes to the organization of this manuscript that improved its readability. Any use of trade, firm, or product names is for descriptive purposes only and does not imply endorsement by the U.S. Government.

**Conflicts of Interest:** Pamela Nagler is an Associate Editor for Remote Sensing, but the authors declare no conflicts of interest. The funders had no role in the design of the study; in the collection, analyses, or interpretation of data; in the writing of the manuscript; or in the decision to publish the results.

## Appendix A

**Table A1.** Average peak growing season EVI2, the average and standard deviation from 2000 to 2020 for the Lower Colorado River.

| YEAR | R3 | R4 | R5 | R6 | R7 | ALL |
|---|---|---|---|---|---|---|
| 2000 | 0.1962 | 0.1820 | 0.1689 | 0.2484 | 0.2069 | 0.1993 |
| 2001 | 0.2090 | 0.1888 | 0.1775 | 0.2528 | 0.2069 | 0.2066 |
| 2002 | 0.2065 | 0.1886 | 0.1683 | 0.2447 | 0.2009 | 0.1999 |
| 2003 | 0.2045 | 0.1845 | 0.1627 | 0.2395 | 0.1961 | 0.1955 |
| 2004 | 0.1937 | 0.1741 | 0.1542 | 0.2400 | 0.1911 | 0.1889 |
| 2005 | 0.2140 | 0.2045 | 0.1795 | 0.2544 | 0.1955 | 0.2070 |
| 2006 | 0.2053 | 0.1937 | 0.1650 | 0.2367 | 0.2007 | 0.1967 |
| 2007 | 0.2013 | 0.1868 | 0.1591 | 0.2233 | 0.1958 | 0.1899 |
| 2008 | 0.2000 | 0.1849 | 0.1605 | 0.2254 | 0.1902 | 0.1895 |
| 2009 | 0.2123 | 0.2033 | 0.1626 | 0.2249 | 0.2029 | 0.1956 |
| 2010 | 0.2221 | 0.2123 | 0.1825 | 0.2363 | 0.2116 | 0.2091 |
| 2011 | 0.2006 | 0.1978 | 0.1659 | 0.2153 | 0.1901 | 0.1895 |
| 2012 | 0.1857 | 0.1813 | 0.1567 | 0.1986 | 0.1828 | 0.1776 |
| 2013 | 0.1869 | 0.1895 | 0.1654 | 0.1960 | 0.1859 | 0.1810 |
| 2014 | 0.1928 | 0.1840 | 0.1616 | 0.2062 | 0.1880 | 0.1837 |
| 2015 | 0.1793 | 0.1858 | 0.1657 | 0.2087 | 0.1845 | 0.1815 |
| 2016 | 0.1627 | 0.1759 | 0.1587 | 0.2026 | 0.1906 | 0.1745 |
| 2017 | 0.1490 | 0.1766 | 0.1707 | 0.2040 | 0.1902 | 0.1755 |
| 2018 | 0.1584 | 0.1830 | 0.1629 | 0.2064 | 0.1992 | 0.1772 |
| 2019 | 0.1607 | 0.1819 | 0.1578 | 0.1766 | 0.1855 | 0.1674 |
| 2020 | 0.1610 | 0.1787 | 0.1494 | 0.1604 | 0.1753 | 0.1650 |
| Average | 0.1906 | 0.1875 | 0.1646 | 0.2191 | 0.1939 | 0.1881 |
| STDEV | 0.0210 | 0.0099 | 0.0082 | 0.0251 | 0.0091 | 0.0127 |

**Table A2.** Differences in EVI2 demonstrating positive or negative (−) change in VI measured from one year to the next showing the average peak growing season, 1 May and October 30, values for each reach and the total weighted average in All reaches between 2000 and 2020 for the Lower Colorado River.

| YEARS | R3 | R4 | R5 | R6 | R7 | ALL |
|---|---|---|---|---|---|---|
| 2001–2000 | 0.0128 | 0.0068 | 0.0086 | 0.0044 | −0.0001 | 0.0073 |
| 2002–2001 | −0.0025 | −0.0002 | −0.0092 | −0.0081 | −0.0060 | −0.0066 |
| 2003–2002 | −0.0021 | −0.0040 | −0.0056 | −0.0052 | −0.0049 | −0.0045 |
| 2004–2003 | −0.0108 | −0.0105 | −0.0085 | 0.0005 | −0.0049 | −0.0066 |
| 2005–2004 | 0.0203 | 0.0304 | 0.0252 | 0.0144 | 0.0044 | 0.0181 |
| 2006–2005 | −0.0086 | −0.0108 | −0.0144 | −0.0177 | 0.0052 | −0.0102 |
| 2007–2006 | −0.0041 | −0.0069 | −0.0060 | −0.0134 | −0.0048 | −0.0068 |
| 2008–2007 | −0.0013 | −0.0019 | 0.0015 | 0.0021 | −0.0056 | −0.0004 |
| 2009–2008 | 0.0124 | 0.0185 | 0.0021 | −0.0005 | 0.0127 | 0.0061 |
| 2010–2009 | 0.0097 | 0.0090 | 0.0199 | 0.0115 | 0.0087 | 0.0135 |
| 2011–2010 | −0.0214 | −0.0145 | −0.0166 | −0.0210 | −0.0215 | −0.0196 |
| 2012–2011 | −0.0149 | −0.0165 | −0.0092 | −0.0167 | −0.0072 | −0.0119 |
| 2013–2012 | 0.0012 | 0.0082 | 0.0087 | −0.0027 | 0.0031 | 0.0034 |
| 2014–2013 | 0.0059 | −0.0055 | −0.0038 | 0.0102 | 0.0021 | 0.0027 |
| 2015–2014 | −0.0136 | 0.0018 | 0.0041 | 0.0024 | −0.0035 | −0.0021 |
| 2016–2015 | −0.0166 | −0.0099 | −0.0070 | −0.0061 | 0.0061 | −0.0070 |
| 2017–2016 | −0.0137 | 0.0008 | 0.0120 | 0.0014 | −0.0004 | 0.0010 |
| 2018–2017 | 0.0093 | 0.0063 | −0.0079 | 0.0024 | 0.0090 | 0.0017 |
| 2019–2018 | 0.0023 | −0.0011 | −0.0051 | −0.0298 | −0.0137 | −0.0098 |
| 2020–2019 | 0.0004 | −0.0032 | −0.0084 | −0.0162 | −0.0102 | −0.0024 |
| AVERAGE | −0.0018 | −0.0002 | −0.0010 | −0.0044 | −0.0016 | −0.0017 |
| STDEV | 0.0113 | 0.0113 | 0.0111 | 0.0117 | 0.0083 | 0.0089 |

**Table A3.** Landsat EVI2 percent change in the riparian vegetation greenness in the Lower Colorado River from year-to-year using average growing season dates from 1 May to October 30.

| Percent Change in EVI2 | R3 | R4 | R5 | R6 | R7 | ALL |
|---|---|---|---|---|---|---|
| 2001–2000 | 6.50 | 3.71 | 5.10 | 1.78 | −0.03 | 3.66 |
| 2002–2001 | −1.18 | −0.11 | −5.17 | −3.20 | −2.88 | −3.21 |
| 2003–2002 | −1.01 | −2.15 | −3.34 | −2.13 | −2.42 | −2.24 |
| 2004–2003 | −5.28 | −5.68 | −5.20 | 0.22 | −2.52 | −3.36 |
| 2005–2004 | 10.48 | 17.49 | 16.35 | 6.00 | 2.29 | 9.57 |
| 2006–2005 | −4.03 | −5.27 | −8.04 | −6.96 | 2.64 | −4.94 |
| 2007–2006 | −1.99 | −3.57 | −3.61 | −5.66 | −2.42 | −3.48 |
| 2008–2007 | −0.65 | −1.04 | 0.91 | 0.92 | −2.87 | −0.19 |
| 2009–2008 | 6.19 | 9.99 | 1.28 | −0.22 | 6.68 | 3.20 |
| 2010–2009 | 4.58 | 4.41 | 12.26 | 5.10 | 4.30 | 6.93 |
| 2011–2010 | −9.65 | −6.85 | −9.09 | −8.88 | −10.18 | −9.38 |
| 2012–2011 | −7.43 | −8.33 | −5.57 | −7.75 | −3.80 | −6.30 |
| 2013–2012 | 0.64 | 4.55 | 5.54 | −1.33 | 1.69 | 1.91 |
| 2014–2013 | 3.16 | −2.91 | −2.30 | 5.23 | 1.11 | 1.47 |
| 2015–2014 | −7.03 | 0.95 | 2.55 | 1.19 | −1.84 | −1.15 |
| 2016–2015 | −9.25 | −5.32 | −4.20 | −2.92 | 3.30 | −3.86 |
| 2017–2016 | −8.39 | 0.43 | 7.55 | 0.71 | −0.21 | 0.56 |
| 2018–2017 | 6.26 | 3.58 | −4.61 | 1.16 | 4.72 | 0.96 |
| 2019–2018 | 1.46 | −0.61 | −3.11 | −14.43 | −6.88 | −5.55 |
| 2020–2019 | 0.22 | −1.74 | −5.31 | −9.18 | −5.49 | −1.42 |

**Table A4.** The EVI2 difference between groups of years for the peak growing season, averaged between May 1 and October 30, is provided for each reach (R3 to R7) and the weighted average as All reaches. The groups include the entire 21-year period from 2000 to 2020; the ten-year periods from 2000–2010 and 2011–2020; the near-five years periods from 2000–2005 (six years), 2006–2010, 2011–2015, and 2016–2020.

| EVI2 AVG Period | R3 | R4 | R5 | R6 | R7 | ALL |
|---|---|---|---|---|---|---|
| 2000–2020 | 0.1906 | 0.1875 | 0.1646 | 0.2191 | 0.1939 | 0.1881 |
| 2000–2010 | 0.2059 | 0.1912 | 0.1673 | 0.2388 | 0.1999 | 0.1980 |
| 2011–2020 | 0.1737 | 0.1834 | 0.1615 | 0.1975 | 0.1872 | 0.1773 |
| 2000–2005 | 0.2040 | 0.1871 | 0.1685 | 0.2466 | 0.1996 | 0.1995 |
| 2006–2010 | 0.2082 | 0.1962 | 0.1659 | 0.2293 | 0.2002 | 0.1962 |
| 2011–2015 | 0.1891 | 0.1877 | 0.1630 | 0.2050 | 0.1863 | 0.1827 |
| 2016–2020 | 0.1584 | 0.1792 | 0.1599 | 0.1900 | 0.1882 | 0.1719 |

**Table A5.** ET (mmd$^{-1}$) from 2000–2020 shows the peak growing season (1 May to October 30) water use for each reach and the total weighted average in All reaches for each of the 21 years in the study and the average and standard deviation over the study period is provided in the last lines of the table.

| ET (mmd$^{-1}$) YEAR | R3 | R4 | R5 | R6 | R7 | ALL |
|---|---|---|---|---|---|---|
| 2000 | 4.59 | 4.11 | 3.82 | 5.09 | 4.46 | 4.34 |
| 2001 | 4.33 | 3.86 | 3.88 | 4.76 | 4.11 | 4.11 |
| 2002 | 3.38 | 3.50 | 2.99 | 3.95 | 3.04 | 3.05 |
| 2003 | 3.62 | 3.13 | 2.86 | 4.05 | 3.27 | 3.38 |
| 2004 | 3.59 | 3.08 | 2.97 | 4.20 | 3.55 | 3.49 |
| 2005 | 4.46 | 4.19 | 3.92 | 5.04 | 3.91 | 4.29 |
| 2006 | 3.64 | 3.17 | 3.02 | 3.97 | 3.33 | 3.43 |

**Table A5.** *Cont.*

| ET (mmd$^{-1}$) YEAR | R3 | R4 | R5 | R6 | R7 | ALL |
|---|---|---|---|---|---|---|
| 2007 | 4.30 | 3.96 | 3.64 | 4.61 | 4.19 | 4.11 |
| 2008 | 3.66 | 3.53 | 3.15 | 4.12 | 3.46 | 3.57 |
| 2009 | 4.20 | 3.92 | 3.37 | 4.30 | 3.91 | 3.83 |
| 2010 | 3.53 | 3.28 | 3.08 | 3.71 | 3.36 | 3.38 |
| 2011 | 3.34 | 3.19 | 2.91 | 3.53 | 3.12 | 3.19 |
| 2012 | 2.95 | 2.76 | 2.59 | 3.16 | 2.90 | 2.86 |
| 2013 | 2.20 | 2.06 | 1.84 | 2.26 | 2.10 | 2.07 |
| 2014 | 3.29 | 3.08 | 2.68 | 3.49 | 3.08 | 3.08 |
| 2015 | 3.15 | 3.37 | 2.97 | 3.59 | 3.19 | 3.17 |
| 2016 | 1.40 | 1.52 | 1.40 | 1.82 | 1.75 | 1.55 |
| 2017 | 2.20 | 2.69 | 2.59 | 3.11 | 2.98 | 2.63 |
| 2018 | 2.17 | 2.48 | 2.41 | 2.87 | 2.72 | 2.45 |
| 2019 | 2.75 | 2.95 | 2.71 | 2.97 | 3.17 | 2.86 |
| 2020 | 3.35 | 3.61 | 3.06 | 3.21 | 3.52 | 3.25 |
| AVERAGE | 3.34 | 3.21 | 2.94 | 3.71 | 3.29 | 3.24 |
| STDEV | 0.84 | 0.67 | 0.62 | 0.85 | 0.64 | 0.71 |

In Appendix A, Table A5, the peak growing season water use for each reach and the weighted average for the full area (All) is provided and, from this information, we show that daily ET over all reaches (All) dropped from 4.34 mmd$^{-1}$ in 2000 to 3.25 mmd$^{-1}$ in 2020. The year 2016 had the lowest daily ET values, ranging between 1.40 mmd$^{-1}$ and 1.82 mmd$^{-1}$ across the five reaches. The maximum daily ET, 5.09 mmd$^{-1}$, was recorded in R6 in 2000, and a value close to that (5.04 mmd$^{-1}$) was recorded in R6 again in 2005.

**Table A6.** Differences in the average daily water use using dates for the peak growing season (May 1 to October 30) for each of the 21 years in the study by subtracting the following year from the initial year as demonstrated in the first column. ET (mmd$^{-1}$) change is either positive or negative (−), and the data are shown for each reach and the weighted average in All reaches, with the average and standard deviation over the study period provided in the last two lines of the table.

| Difference in ET (mmd$^{-1}$) between Years | R3 | R4 | R5 | R6 | R7 | ALL |
|---|---|---|---|---|---|---|
| 2001–2000 | −0.26 | −0.25 | 0.06 | −0.33 | −0.35 | −0.23 |
| 2002–2001 | −0.96 | −0.36 | −0.90 | −0.81 | −1.07 | −1.05 |
| 2003–2002 | 0.24 | −0.37 | −0.13 | 0.10 | 0.23 | 0.32 |
| 2004–2003 | −0.03 | −0.05 | 0.11 | 0.15 | 0.28 | 0.11 |
| 2005–2004 | 0.87 | 1.11 | 0.95 | 0.83 | 0.36 | 0.80 |
| 2006–2005 | −0.81 | −1.02 | −0.90 | −1.07 | −0.58 | −0.86 |
| 2007–2006 | 0.66 | 0.79 | 0.62 | 0.64 | 0.87 | 0.68 |
| 2008–2007 | −0.65 | −0.42 | −0.49 | −0.49 | −0.73 | −0.54 |
| 2009–2008 | 0.54 | 0.39 | 0.22 | 0.17 | 0.45 | 0.26 |
| 2010–2009 | −0.67 | −0.65 | −0.29 | −0.59 | −0.55 | −0.45 |
| 2011–2010 | −0.19 | −0.08 | −0.17 | −0.18 | −0.24 | −0.19 |
| 2012–2011 | −0.39 | −0.44 | −0.32 | −0.37 | −0.22 | −0.33 |
| 2013–2012 | −0.76 | −0.70 | −0.75 | −0.89 | −0.80 | −0.79 |
| 2014–2013 | 1.09 | 1.02 | 0.85 | 1.23 | 0.98 | 1.01 |
| 2015–2014 | −0.14 | 0.30 | 0.29 | 0.10 | 0.11 | 0.09 |
| 2016–2015 | −1.75 | −1.86 | −1.58 | −1.77 | −1.44 | −1.62 |
| 2017–2016 | 0.80 | 1.17 | 1.20 | 1.29 | 1.23 | 1.08 |
| 2018–2017 | −0.03 | −0.21 | −0.18 | −0.24 | −0.26 | −0.18 |
| 2019–2018 | 0.58 | 0.47 | 0.30 | 0.10 | 0.45 | 0.41 |
| 2020–2019 | 0.60 | 0.66 | 0.35 | 0.24 | 0.35 | 0.39 |
| Average | −0.06 | −0.03 | −0.04 | −0.09 | −0.05 | −0.05 |
| STDEV | 0.74 | 0.77 | 0.69 | 0.76 | 0.70 | 0.71 |

Appendix A, Table A6 shows the difference in the average peak growing season water use for each reach and the total weighted average in All reaches. The greatest differences across all reaches were observed between 2015 and 2016, possibly due to the initial wave of biocontrol defoliation, where negative change persisted with average daily ET ranging between a decrease of 1.44 mmd$^{-1}$ and 1.86 mmd$^{-1}$, and All reaches was defined by a decrease in average daily ET of 1.62 mmd$^{-1}$.

**Table A7.** Percent change between years for the average daily water use using dates for the peak growing season (1 May to October 30) for each of the 21 years in the study. ET (mmd$^{-1}$) change is either positive or negative ($-$) and the data is shown for each reach and the weighted average in All reaches.

| Percent Change in ET (mmd$^{-1}$) between Years | | | | | | |
|---|---|---|---|---|---|---|
| YEARS | R3 | R4 | R5 | R6 | R7 | ALL |
| 2001–2000 | −5.6 | −6.1 | 1.7 | −6.6 | −7.9 | −5.3 |
| 2002–2001 | −22.1 | −9.3 | −23.1 | −17.0 | −26.0 | −25.6 |
| 2003–2002 | 7.2 | −10.6 | −4.4 | 2.5 | 7.5 | 10.5 |
| 2004–2003 | −0.8 | −1.6 | 3.9 | 3.8 | 8.4 | 3.3 |
| 2005–2004 | 24.1 | 35.8 | 32.1 | 19.8 | 10.3 | 23.0 |
| 2006–2005 | −18.3 | −24.3 | −22.9 | −21.2 | −14.9 | −20.0 |
| 2007–2006 | 18.1 | 24.8 | 20.5 | 16.2 | 26.0 | 19.7 |
| 2008–2007 | −15.0 | −10.7 | −13.4 | −10.7 | −17.5 | −13.2 |
| 2009–2008 | 14.8 | 11.0 | 7.1 | 4.2 | 13.1 | 7.3 |
| 2010–2009 | −15.9 | −16.5 | −8.6 | −13.7 | −14.1 | −11.8 |
| 2011–2010 | −5.4 | −2.5 | −5.5 | −4.8 | −7.1 | −5.6 |
| 2012–2011 | −11.6 | −13.7 | −11.1 | −10.6 | −7.2 | −10.5 |
| 2013–2012 | −25.6 | −25.3 | −29.0 | −28.3 | −27.5 | −27.7 |
| 2014–2013 | 49.6 | 49.5 | 46.0 | 54.5 | 46.5 | 49.1 |
| 2015–2014 | −4.1 | 9.7 | 10.7 | 2.8 | 3.6 | 2.9 |
| 2016–2015 | −55.7 | −55.0 | −53.0 | −49.3 | −45.1 | −51.1 |
| 2017–2016 | 57.4 | 77.3 | 86.0 | 71.1 | 70.3 | 69.5 |
| 2018–2017 | −1.4 | −7.9 | −7.1 | −7.8 | −8.8 | −6.8 |
| 2019–2018 | 26.9 | 19.0 | 12.4 | 3.4 | 16.7 | 16.8 |
| 2020–2019 | 21.7 | 22.2 | 12.8 | 8.2 | 11.1 | 13.7 |

Appendix A, Table A7 shows percent change in both positive and negative ($-$) directions for the year-to-year changes from 2000 to 2020 for the average, daily peak growing season ET (mmd$^{-1}$) and is presented for each reach and the total weighted average in All reaches.

**Table A8.** Difference in PAM ET (mmyr$^{-1}$) between years using the full year for each of the 21 years in the study. PAM ET (mmyr$^{-1}$) change is either positive or negative and the data is shown for each reach and the weighted average in All reaches.

| Difference in PAM ET (mmyr$^{-1}$) between Years | | | | | | |
|---|---|---|---|---|---|---|
| YEARS | R3 | R4 | R5 | R6 | R7 | ALL |
| 2001–2000 | −85.6 | −73.2 | −43.6 | −117.6 | −178.1 | −25.3 |
| 2002–2001 | 135.8 | 227.0 | 86.8 | 75.4 | 166.3 | 102.4 |
| 2003–2002 | −128.1 | −221.1 | −204.8 | −141.0 | −229.2 | −189.5 |
| 2004–2003 | −96.3 | −104.2 | −57.9 | −58.3 | −11.2 | −59.9 |
| 2005–2004 | 179.4 | 249.4 | 206.8 | 153.4 | 46.0 | 160.8 |
| 2006–2005 | −101.0 | −181.5 | −142.5 | −121.5 | −18.1 | −105.9 |
| 2007–2006 | −115.1 | −138.7 | −105.3 | −182.3 | −100.0 | −103.7 |

**Table A8.** *Cont.*

| Difference in PAM ET (mmyr$^{-1}$) between Years | | | | | | |
|---|---|---|---|---|---|---|
| 2008–2007 | 143.3 | 174.9 | 155.5 | 214.0 | 19.4 | 152.2 |
| 2009–2008 | 38.2 | 11.0 | −14.8 | −101.4 | 77.8 | −13.7 |
| 2010–2009 | −129.8 | −63.4 | −40.7 | −71.0 | −41.9 | −85.8 |
| 2011–2010 | −74.2 | −45.7 | −91.4 | −57.5 | −94.6 | −79.9 |
| 2012–2011 | −77.9 | −81.0 | −49.3 | −50.4 | −19.4 | −57.0 |
| 2013–2012 | 92.9 | 152.9 | 92.9 | 44.7 | 59.2 | 92.2 |
| 2014–2013 | 170.8 | 70.3 | 84.4 | 198.2 | 148.5 | 128.6 |
| 2015–2014 | −193.9 | −121.8 | −103.0 | −120.9 | −131.7 | −109.2 |
| 2016–2015 | −157.7 | −105.3 | −97.2 | −103.6 | −79.3 | −134.0 |
| 2017–2016 | 57.8 | 159.8 | 179.5 | 130.8 | 113.8 | 121.3 |
| 2018–2017 | −50.4 | −96.1 | −122.5 | −116.6 | −35.1 | −76.2 |
| 2019–2018 | 0.2 | −38.9 | −21.8 | −110.7 | −61.0 | −51.8 |
| 2020–2019 | 182.9 | 184.8 | 90.4 | 65.0 | 89.6 | 88.14 |

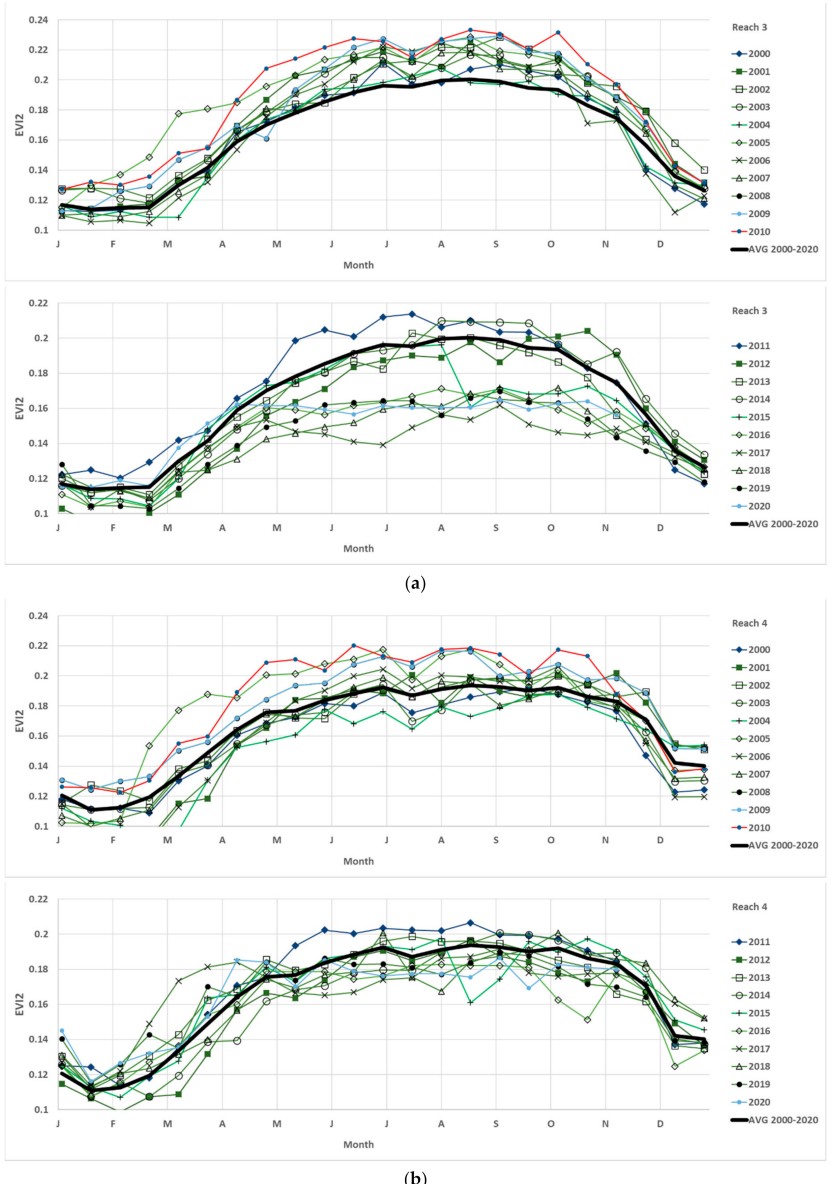

**Figure 1.** *Cont.*

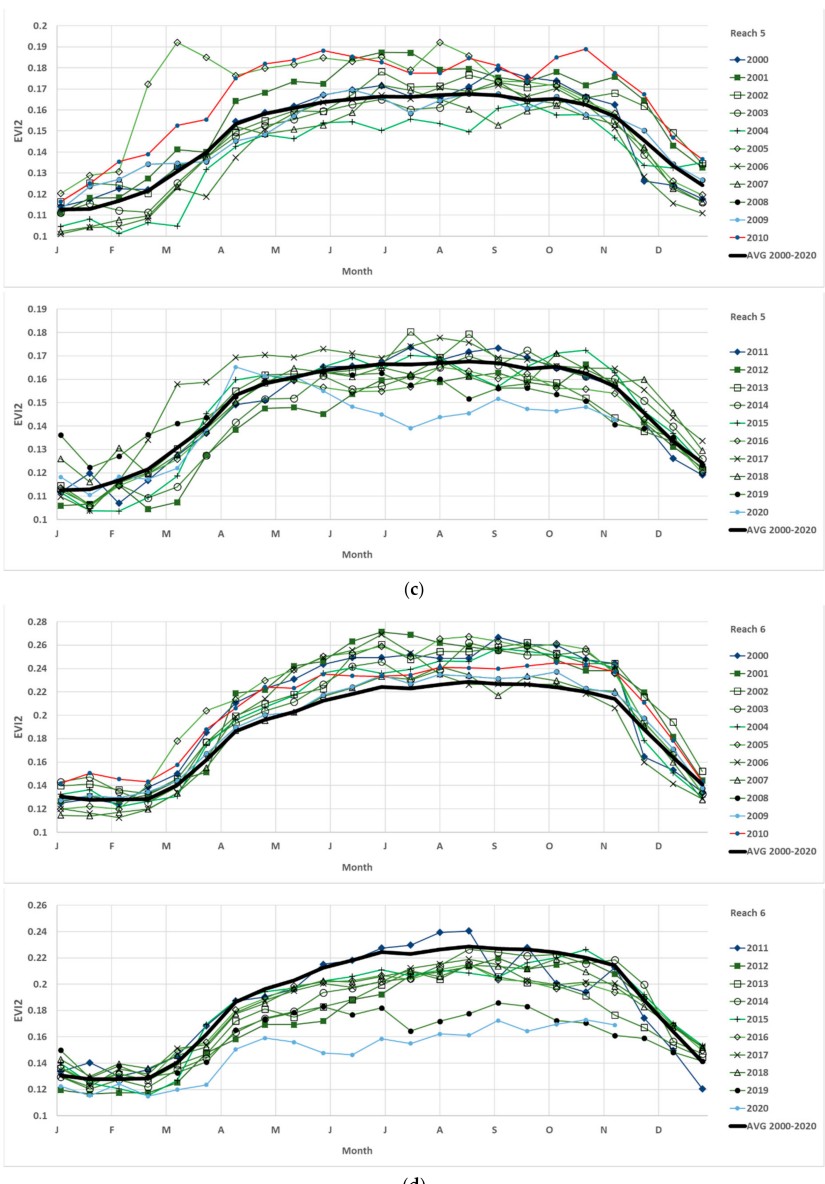

(**c**)

(**d**)

**Figure 1.** *Cont.*

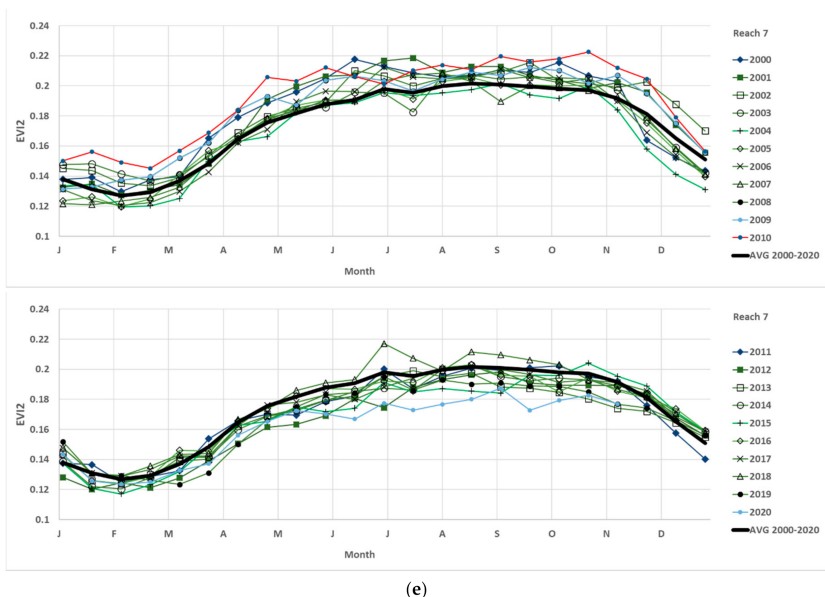

(e)

**Figure 1.** (**a–e**). EVI2 time-series information for the full 21-year period, 2000–2020, for the five reaches, R3 to R7 (Figure 1a–e),
for the full year with months on the *x*-axis (J-D), and for individual years 2000–2010 (**top**) and 2011–2020 (**bottom**).

Appendix A Figure 1a–e shows each individual reach with R3 in Figure 1a, R4 in
Figure 1b, R5 in Figure 1c, R6 in Figure 1d, and R7 in Figure 1e. In these five critically
important, reach-level plots of EVI2 over the last two decades (top is 2000–2010, bottom is
2011–2020), we demonstrate where and when positive or negative (−) changes occurred,
and we show not only the magnitude of the greenness change, but also any temporal shifts
(e.g., green-up dates and senescence onset) which exist compared to not only the year prior,
but also the 21-year average (bold black). For example, these plots generally show that in
2011 the riparian vegetation remained healthy. Following this, the year 2017 was not only
the worst year in terms of vegetation health as measured by EVI2 in R3 and R4, but also
the first to green up in R4 and also in R5. Green up occurred as early as March, showing a
departure from the normal phenology curve relative to the other years, which reached the
same VI two months later in May. For R4, 2017 not only peaked two months earlier than in
other years, but also depicts the least amount of green vegetation through July compared
to all other years. The most recent year of the study, 2020, shows very low EVI2 in R3 and
R4, and the least amount in R5–R7, actually falling far below the two ten-year periods and
the overall average of the 21 years of the study in R5 and R6 especially. The data from 2019
show declines in vegetation health across all five reaches.

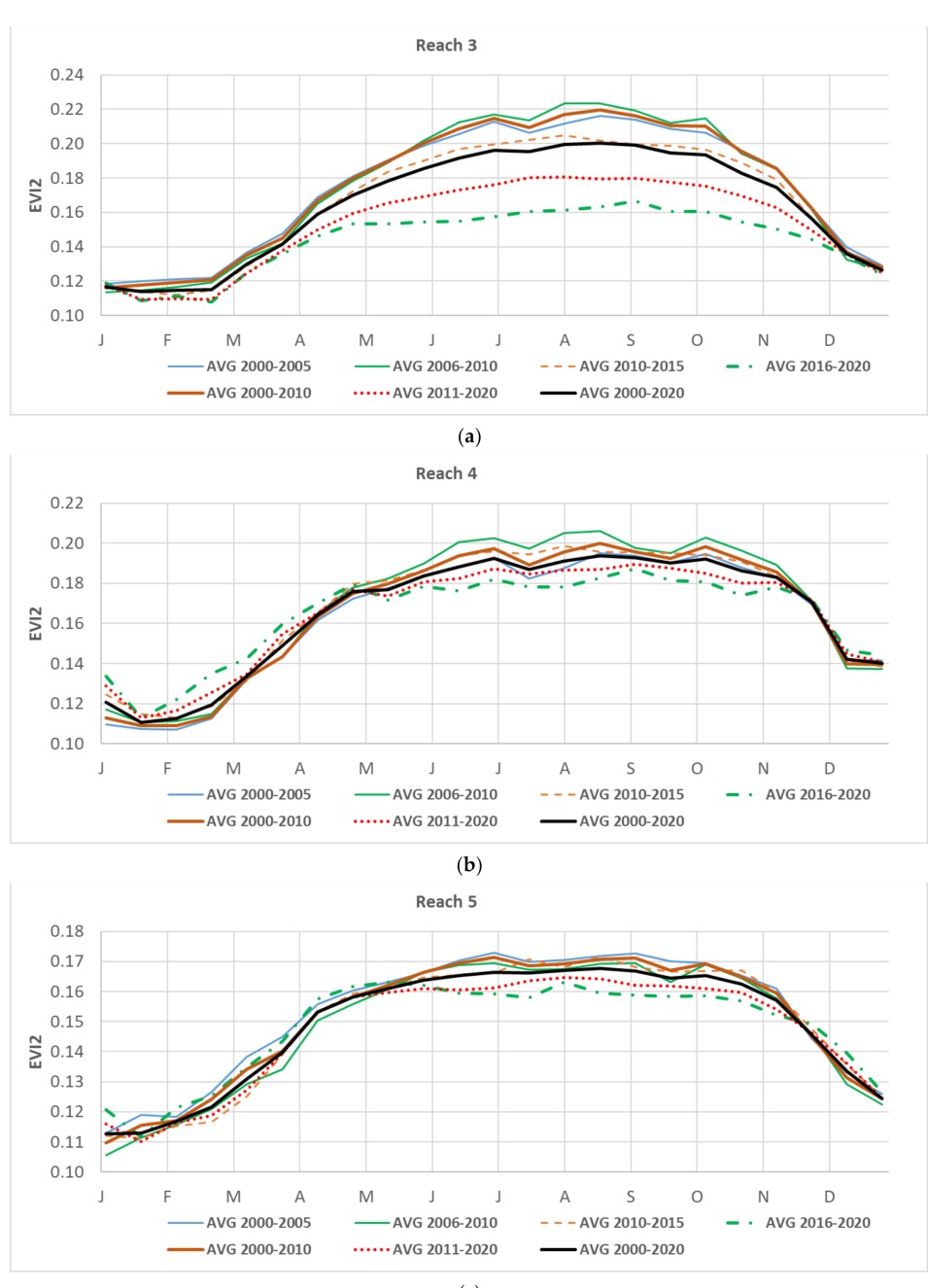

**Figure 2.** *Cont.*

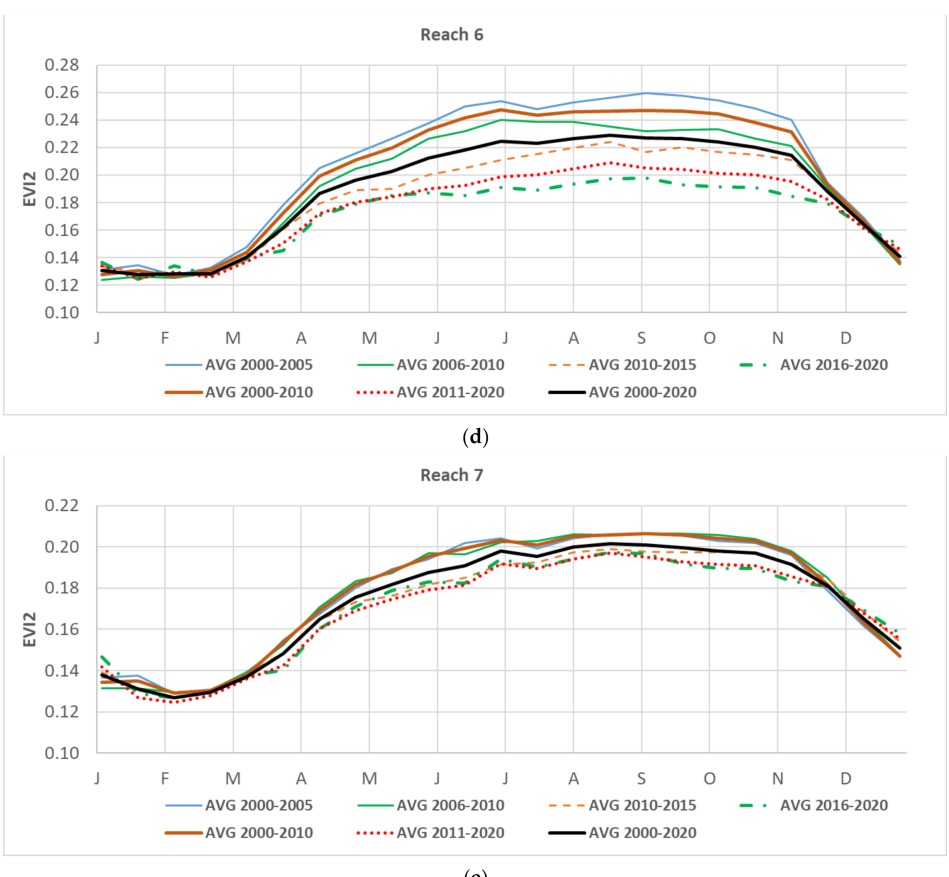

(**d**)

(**e**)

**Figure 2.** (**a**–**e**). EVI2 time-series information for the five reaches, R3–R7 (Figure 2a–e) for the full year with months on the *x*-axis (J–D), for years 2011 to 2020 individually and averaged by two decades, 2000–2010 and 2011–2020, and the full 21-year period, 2000–2020 as well as the two recent five-year periods, 2010–2015 and 2016–2020.

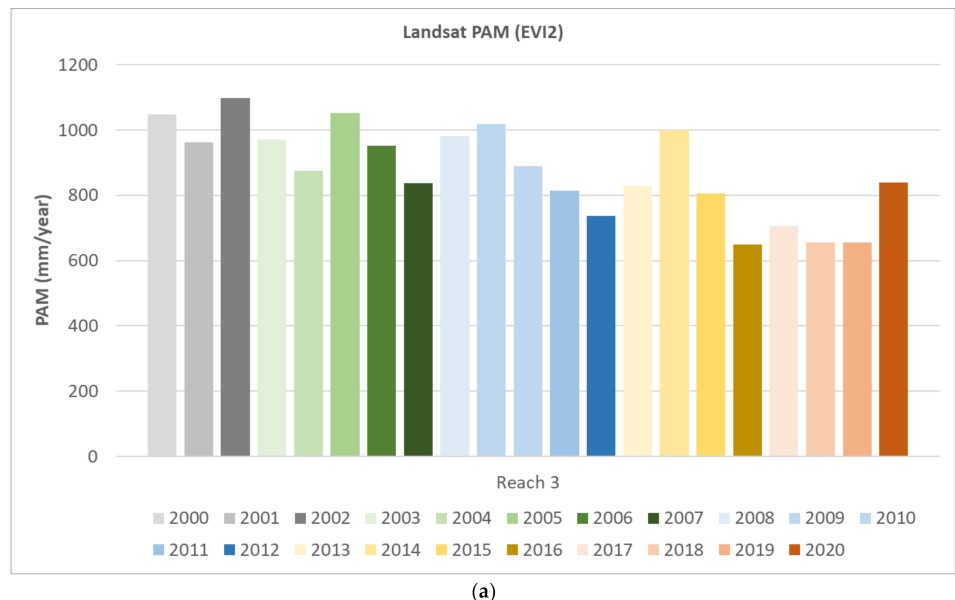

(**a**)

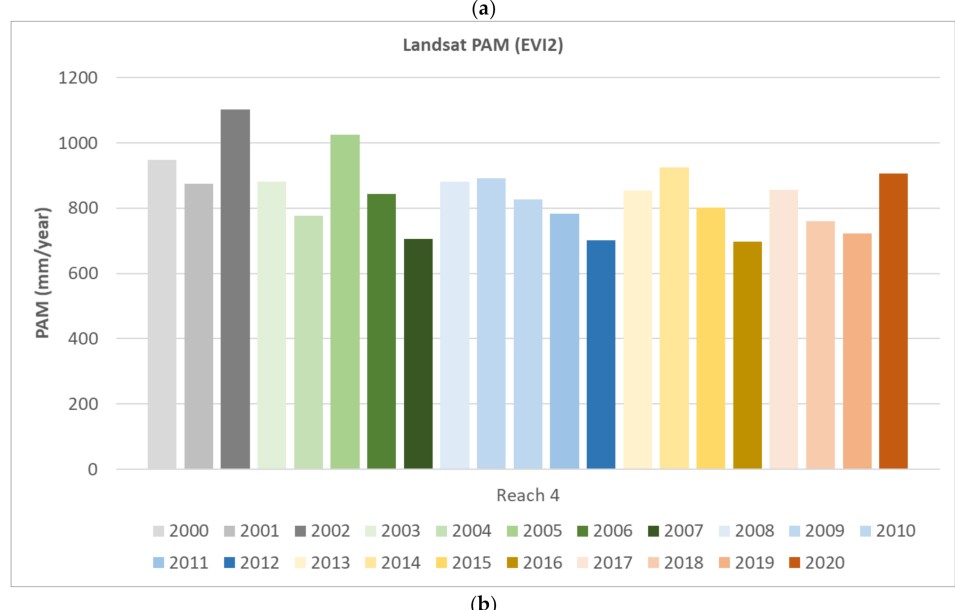

(**b**)

**Figure A3.** *Cont.*

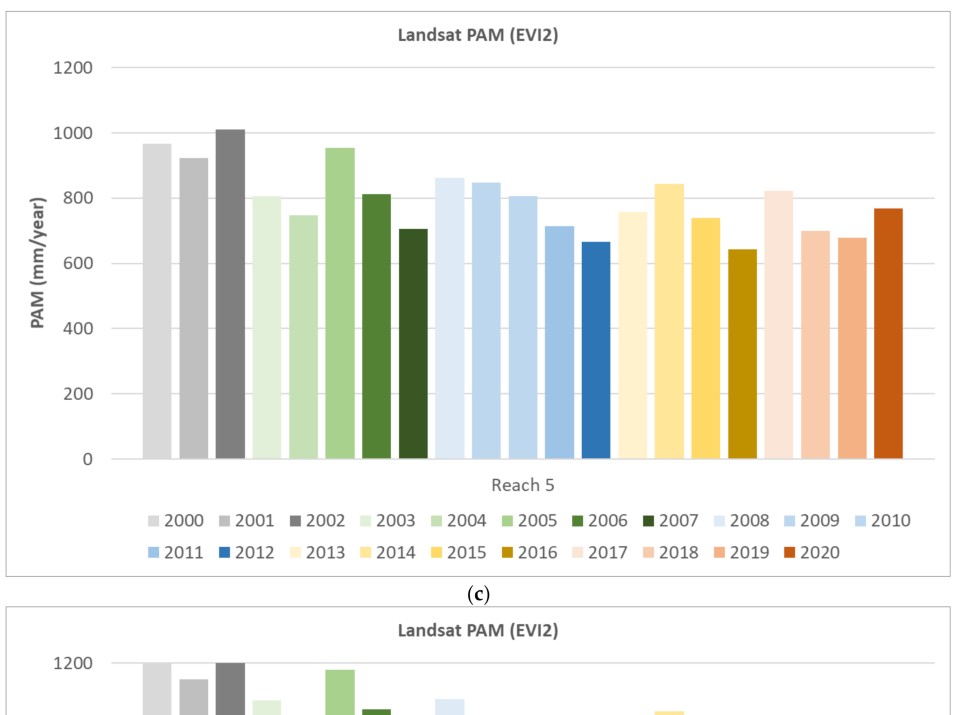

(**c**)

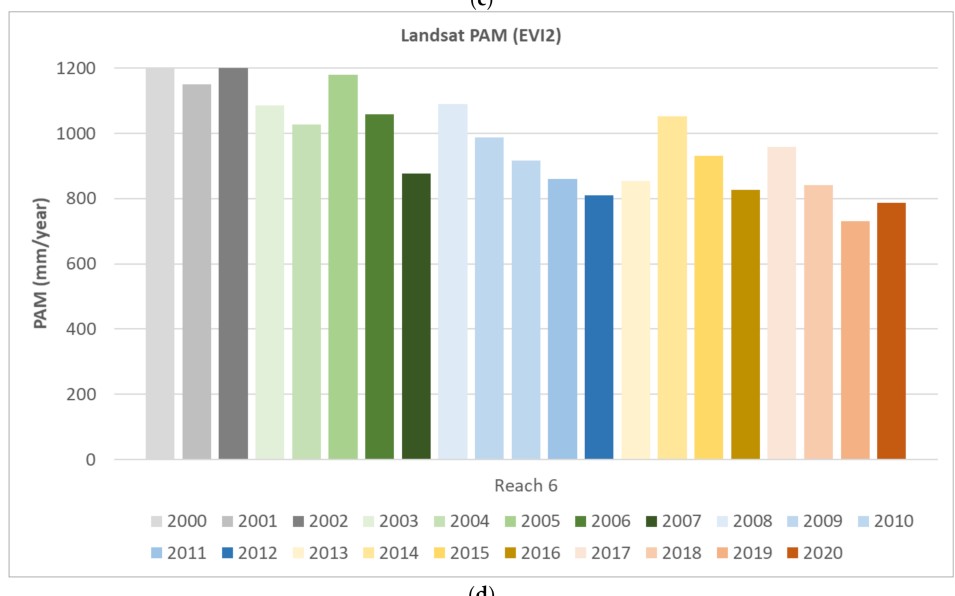

(**d**)

**Figure A3.** *Cont.*

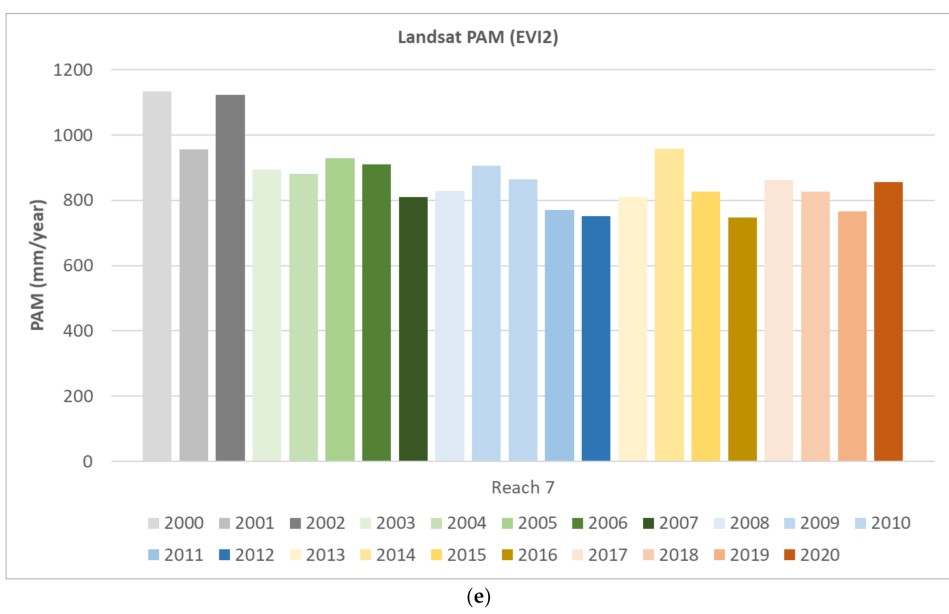

(e)

**Figure A3.** (**a**–**e**). Bars of PAM ET (mmyr$^{-1}$) based on EVI2 time-series information for the Lower Colorado River riparian reaches R3 to R7 shown in Figure A3a–e for each of 21 years between 2000 and 2020 on the *x*-axis.

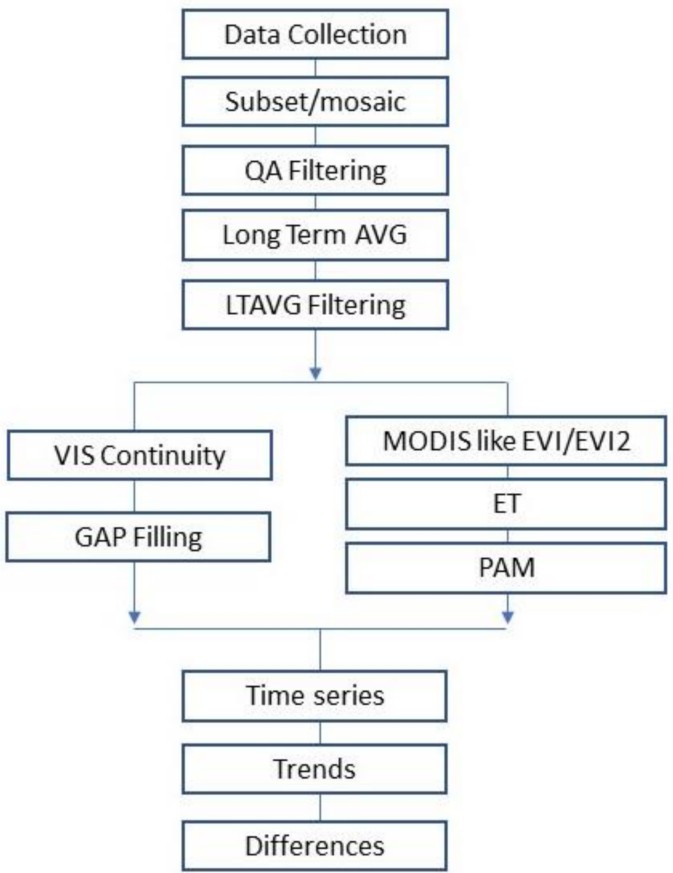

**Figure A4.** Flow chart of methodology.

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
