# Peer review of "Riparian Area Changes in Greenness and Water Use on the Lower Colorado River in the USA from 2000 to 2020"

_remotesensing, doi:10.3390/rs13071332_

Round 1

Reviewer 1 Report

I felt that this was a very strong and timely study and I provided numerous, but minor recommendations for revisions. These would shorten and streamline the paper and there were a few clerical corrections.

The authors have undertaken a thorough revision and adopted almost all of my recommendations. With the relocation of some results to supplemental files, the published paper will be suitably shortened - I think that this will improve the accessibility to the readers. Those interested in some of the more detailed aspects are still able to access the linked results.

I thus recommend acceptance and it will provide an important contribution. It will be cited and will hopefully influence management.

Author Response

“An excellent revision - all of my minor concerns were addressed.”

Response:  We very much appreciate that you have read the revision that we improved with your guidance. The audience that we have hoped to respond to are both academics and land managers and our goal is to provide to them with our insights on this part of the Lower Colorado River.

I felt that this was a very strong and timely study and I provided numerous, but minor recommendations for revisions. These would shorten and streamline the paper and there were a few clerical corrections.

Response:  We have continued to shorten and streamline the submitted revision.

The authors have undertaken a thorough revision and adopted almost all of my recommendations. With the relocation of some results to supplemental files, the published paper will be suitably shortened - I think that this will improve the accessibility to the readers. Those interested in some of the more detailed aspects are still able to access the linked results.

Response:  We have continued to relocate some of the text to results and shorten the submitted revision.

I thus recommend acceptance and it will provide an important contribution. It will be cited and will hopefully influence management.

Response:  We are grateful and hope this does become an important contribution.

Reviewer 2 Report

The paper presents an evaluation of the change of a riparian area according to its greenness and water use with the use of remote sensing. The current version of the manuscript seems to be a draft, the reading is hard to follow and the content is too much. It needs to be reduced in the number of pages., In addition, different colors can be found in the text and it does not use the format requested by the journal. Thus, it does not seem to be ready for publication unless some major revisions get addressed.

Revisions are listed below:

General comments

What are the new contributions this study provides compared to previous research? Authors could improve the readers’ interest by highlighting what knowledge gap they are filling with their paper

What are the advantages and disadvantages of the methodology used in this study?

Are there any studies similar to this one? What methodologies have been used? How are they different from this study?

For the most part, the Discussion seems to be a continuation of the Results Section instead of focusing on the implications of the findings from this reasearch.

What are the limitations of this study, what needs to be done?

Authors could also emphasize particular strengths and limitations of the study for potential applications of their method in other regions, contexts and scales.

For readers involved in the management of riparian ecosystems, what can be learned from this study?

The answer to these questions should be reflected in the manuscript.

There are many references used to support an idea; one or two are sufficient for that purpose. Edit throughout the manuscript. That would reduce the number of references used; the number now is too high.

The manuscript can be improved by including a figure summarizing the methodology of the work.

The manuscript does not follow the instructions for authors that the journal requires, consider using the Template.

Specific comments

Line 41: Check journal format for in-text citations.

Line 44: Department of the Interior, from where?

Line 45: Check journal format for in-text citations.

Line 80: This is the first time USGS is mentioned, write its meaning.

Line 139: This is not the correct way to cite.

Line 220-223: Why in 7, and not in 9, 10?

Line 230: Add the reference system grid.

Line 241: In the sentence: "the same methods used recently in the riparian ecosystem...". Which methods?

Line 246-247: What was the standard radiometric correction method employed?

Line 463: Table 1 does not follow the Journal format.

Line 734-840: It seems you are explaining results and not the implications of what was obtained in this study.

Line 760: If NDVI was not included in the Discussion Section, either delete these results or discuss them accordingly in the Discussion Section.

Author Response

Thank you for the excellent review that helped us to improve our submission. The response file is attached and copied and pasted below, except for the flow chart.

Comments and Suggestions for Authors

The paper presents an evaluation of the change of a riparian area according to its greenness and water use with the use of remote sensing. The current version of the manuscript seems to be a draft, the reading is hard to follow and the content is too much. It needs to be reduced in the number of pages. In addition, different colors can be found in the text and it does not use the format requested by the journal. Thus, it does not seem to be ready for publication unless some major revisions get addressed.

Response:  Thank you very much for your review of our study that follows on the study further downstream in the delta (Nagler et al. 2020, Hydrological Processes). The audience that we have hoped to respond to are both academics and land managers. Our goal for this companion study is to provide to them with our insights on this part of the Lower Colorado River (LCR). Although this was not a draft, we agree this ms needed to be further streamlined and shortened. In this revision, we have removed more figures from the reach-level results to the Supplemental Information in Appendix B, Figures 1a-1e that were Figures 5a-5e. We have continued to relocate much of the Discussion text to Results and some to the supplemental information and this shortened the Discussion text to allow focus on the main findings for this submitted revision. The Discussion now focuses on (a) remotely sensed metric findings, implications, strengths and limitations and (b) new contributions and next steps. We appreciate that you have suggested guidance that is followed now in this revision.

Revisions are listed below:

General comments

What are the new contributions this study provides compared to previous research? Authors could improve the readers’ interest by highlighting what knowledge gap they are filling with their paper

Response: The primary contribution is that riparian vegetation health in the LCR has been declining since 2000, and this has had an impact on riparian woodlands water use which has a significant but often incorrectly measured contribution to water balance.

What are the advantages and disadvantages of the methodology used in this study?

Are there any studies similar to this one? What methodologies have been used? How are they different from this study?

Response: This study was conducted as a follow on to a research chapter for the International Boundary Water Commission (IBWC) report on the Water Treaty Minutes 319 and 323 that look at the riparian ecosystem health and water use in the delta of the Colorado River (see also Nagler et al. 2020, Hydrological Processes). We compared two scales (Landsat and MODIS) and two indices for greenness, EVIs (EVI and EVI2), and finally two estimates of ET (the daily ET with peak growing season dates and the annual PAM ET). In that study in the delta, we first developed and applied the annualized water use (PAM ET) due to riparian vegetation growing in this hot and arid region entirely south of Yuma, AZ. This study on the U.S. side of the LCR is an important study involving common methods and an overlap in some objectives. For this study, the methods were different in that we excluded MODIS since our finding in the HYP ms was that it was statistically comparable with Landsat, and thus we chose to use the higher resolution Landsat only. The methods here also rely entirely on EVI2, the simpler version to estimate greenness. Our data showed an r2 = 0.99 between EVI and EVI2. And the methods in this study report both the daily and annual ET for this region that allows for most of the riparian woodlands to ‘grow’ year-round. This research on the LCR showed some unexpected and significant findings. The initial study in the delta was a region that declines in riparian health were expected because of diverted water at the international boundary, but these significant declines were surprising to find for the LCR on the US side. This study also provided added methods. We grouped years together to compare to the year-to-year ups-and-downs to add more information by using groups. Also, we did not see a logical reason for doing a statistical test between the ETs averaged across four groups of years because the declines in ecosystem health are clear, but we did compare the trend of the total annualized water use in each five reaches and in the five-year periods and found no statistically significant differences because longer time periods are required to apply these methods. These analyses methods were new to this revision, but unnecessary to include in this study and will be applied in future research.

For the most part, the Discussion seems to be a continuation of the Results Section instead of focusing on the implications of the findings from this research. What are the limitations of this study, what needs to be done?

Response: We have continued to relocate some of the text from discussion to results, shorten text where there is overlap, and added the limitations of the study to the discussion. For example, VI-based ET methods should be applied primarily in regions where there are no illumination issues from haze, clouds, shadows, etc. We perform image correction, but this optical-based ET estimation does not perform well outside of arid and semi-arid regions. One of the limitations of this study is that it is not compared to other ET estimates such as those many options available from USGS (SSEBop) and Open ET. We have the data for these comparisons for the LCR riparian corridor, but it was outside the scope of this ms and will be performed ‘next.’ Furthermore, we do expect to produce results using our EVI2 and ET(EVI2) methods on the urban regions within the LCR as a next step to much of the research led by Nouri et al. in urban green spaces. And our future work in 2021 includes comparing the unrestored riparian corridor on the LCR to the Multi-Species Conservation Plan restoration sites, an important study (outside the scope of this ms) to the end-users, NGOs and resource managers of the LCR lands.

Authors could also emphasize particular strengths and limitations of the study for potential applications of their method in other regions, contexts and scales.

Response: Thank you, we have added a portion of the text above to the ms to highlight the strengths and limitations and discuss future work that we expect to produce this year. Furthermore, we add a section on how these VI-based ET findings have been applied in Australia riparian zones, how riparian regions have declined elsewhere, and we added to the discussion the application of this method to urban gardens in Australia and how we expect these methods to work on the LCR for both the adjacent urban regions and the restored plots on the US portion as well as the restoration zones in Mexico. See new text section under 4.3. New contributions and next steps

For readers involved in the management of riparian ecosystems, what can be learned from this study? The answer to these questions should be reflected in the manuscript.

Response: We have added a portion of the text describing what can be learned in a new section at the end of the shortened discussion. See text under 4.2. Strengths and limitations as well as new text under 4.3. New contributions and next steps

There are many references used to support an idea; one or two are sufficient for that purpose. Edit throughout the manuscript. That would reduce the number of references used; the number now is too high.

Response: We are aware that there are too many references, but we have not reduced their number; however, in a few places, we have reduced the extra citations per topic.

The manuscript can be improved by including a figure summarizing the methodology of the work.

Response: A flow chart has been produced, but we are not keen on including it in the main text because of the length of the manuscript and so we include it in Appendix B as Figure A4.

(see attached file, the image would not paste here)

The manuscript does not follow the instructions for authors that the journal requires, consider the Template.

Response:  Upon contacting the submission coordinator, I learned that a new version of the template was updated and this occurred after our first submission. I was assured that our format is okay to use here, so we have not changed the formatting as requested. The following text was sent by the managing editor regarding the journal’s layout.

> The template has updated since December 2020. However,
> this will not be a problem. We'll do the layout work for you later.
> As for the in-text citations, for example Line 41: [1,2,3,4,5] should
> be [1–5]. Also, this will not be a problem.

Specific comments

Line 41: Check journal format for in-text citations.

Response: see above from managing editor.

Line 44: Department of the Interior, from where?

Response: We have added the U.S.

Line 45: Check journal format for in-text citations.

Response: see above from managing editor.

Line 80: This is the first time USGS is mentioned, write its meaning.

Response: We have added the U.S. Geological Survey

Line 139: This is not the correct way to cite.

Response: see above from managing editor.

Line 220-223: Why in 7, and not in 9, 10?

Response:  I am unsure what is being asked. We chose to number with 7 reaches because we had 7 reaches in the delta and the templates for maps, figs and tables already existed.

Line 230: Add the reference system grid.

Response:  This has been added to the figure (Fig. 1)

Line 241: In the sentence: "the same methods used recently in the riparian ecosystem...". Which methods?

Response: we add [14] to reflect the companion study in the delta (Nagler et al., 2020)

Line 246-247: What was the standard radiometric correction method employed?

Response:  We did not do any further radiometric correction because the data we obtained from the USGS site already was corrected for all reflectance in all bands and the VIs were already calculated.

Line 463: Table 1 does not follow the Journal format.

Response: see above from managing editor.

Line 734-840: It seems you are explaining results and not the implications of what was obtained in this study.

Response: We moved the text describing the explanations of results, where necessary, to the results section and we now discuss the implications of the study in discussion in better detail.

Line 760: If NDVI was not included in the Discussion Section, either delete these results or discuss them accordingly in the Discussion Section.

Response: The NDVI time-series is shown in Figure 3a, the NDVI* (scaled version of NDVI) is used in Figure 12, 13 for change maps. We add a sentence to describe how NDVI (Figure 3a) is shown for comparison to EVI2 (Figure 3b) and for reference for other studies that use NDVI as the primary standard NASA product.

Submission Date

25 February 2021

Date of this review

10 Mar 2021 22:47:25

Reviewer 3 Report

 A substantial revision is required to enhance this manuscript's quality before it is considered for potential publication.

  1. The abstract needs to be carefully rephrased to highlight the new findings with innovation for this study rather than addressing too much on what they did. 
  2. The introduction is somewhat too long. Instead, the authors should condense the scientific importance of this study and underline the researching gaps between the existing studies.
  3. The statistical analysis for the assumed cause-effect relationships between overall declines in riparian health and water use is missing. I mean, at least the multiple regression approach should be employed to quantify the assumed cause-effect relationships.
  4. The authors presented the full visions of past trends in metrics indicating riparian health and water use; however, further analysis for decomposing the time series of the datasets should be performed to explore the useful information underlying the curves. Frank to say, an insightful analysis for the results (from Fig.3.1 to Fig.8) should be refined.
  5. For Figure 9, why not perform the significance test between the ETs averaged across four groups of years? 
  6. The discussion section is too long. What are the new findings and limitations of this study? What are the key issues that deserve further investigation in future research?

Author Response

Thank you for your excellent reviewer comments. We have substantially revised the ms accordingly. Please see the attached response document which has been copied and pasted below. The statistics table may not have copied below very well, but is in the Word Document attached.

Comments and Suggestions for Authors

A substantial revision is required to enhance this manuscript's quality before it is considered for potential publication.

Response:  Thank you very much for your review and helpful comments for our revision. We agree this ms needed to be further streamlined and shortened. We have continued to relocate some of the figures and corresponding text to the Supplemental Information found in Appendix A (now Figures A1a-e). We also shortened the main text for this submitted revision and moved text in discussion to results so that we could enhance our discussion to focus on findings and limitations of our study. Most of the Discussion is entirely new. We appreciate that you have suggested guidance that is followed now in this revision.

The abstract needs to be carefully rephrased to highlight the new findings with innovation for this study rather than addressing too much on what they did. 

Response:  We have rewritten the abstract to include key findings. We are unsure if a word limit applies to the abstract post-reviewer requests, but we wrote this concisely.

  1. The introduction is somewhat too long. Instead, the authors should condense the scientific importance of this study and underline the researching gaps between the existing studies.

Response:  We have slightly shortened the introduction, but primarily discussed the research gaps and information from existing studies in the new Discussion. We deleted some unnecessary sentences and condensed the importance of the existing issues for the end user community of NGOs, academics and managers who are particularly interested in the state of these riparian ecosystems undergoing impacts from biocontrol and drought. Not in the introduction, but in the discussion, we have added a bit on the existing gaps in recent research on riparian areas and included some text on studies where riparian greenness and water use has been done in other arid land regions.

  1. The statistical analysis for the assumed cause-effect relationships between overall declines in riparian health and water use is missing. I mean, at least the multiple regression approach should be employed to quantify the assumed cause-effect relationships.

Response:  Based on your suggestion, we compared trends of the total annualized water use in each of the five reaches and in the five-year periods (see below). However, since the main message of our research is demonstrating decreases/increases of greenness and water use of riparian vegetationover time, and investigating their differences and percent change between periods we do not see any necessity for further statistical analysis to arrive to the same conclusion. Our current analyses clearly state that these ecosystems are undergoing losses in greenness and water use.

  1. The authors presented the full visions of past trends in metrics indicating riparian health and water use; however, further analysis for decomposing the time series of the datasets should be performed to explore the useful information underlying the curves. Frank to say, an insightful analysis for the results (from Fig.3.1 to Fig.8) should be refined.

Response:  We revised the manuscript extensively to explain the purpose and main message of this research better, highlight its strengths and limitations, discuss the application of our method to other regions and landscapes, and also describe what can be learned from this study. We believe that our revised manuscript with current analyses is adequate to serve our purpose. We appreciate your suggestion for a more detailed exploration of our datasets; we will indeed consider it in our next studies.

  1. For Figure 9, why not perform the significance test between the ETs averaged across four groups of years? 

Response:  We assumed that we have not been clear enough in our original manuscript about the purpose and main message of our study; this might be the reason to ask for extra data analysis (comments 3 to 5). We hope that with our comprehensive revision, we could successfully communicate our research’s purpose. For the previous comment and this one, we (coauthors) discussed these comments among ourselves. Since the purpose of this particular analysis is unclear to us and sounds outsitdeof our study scope, we decided not to add extra analysis to the revised manuscript. Trend analyses is another topic for a future manuscript, but we performed it on our dataset. Below you can see a new table that was created by our coauthor; yet, we do not feel the evaluation of five-year periods adds to this ms. so we do not include it here, but are considering this type of analysis for future work.

Table for Stats: Modified Mann Kendall test for the total annualized water use in each five reaches and in the five-year periods.

Reach

Period

Z-Value

Sen's slope

S

Var(S)

P-value

Tau

3

2000-2005

-0.73

-36.53

-4.00

16.66

0.46

-0.40

2005-2010

0.24

2.75

0.00

16.67

0.81

0.00

2010-2015

0.24

17.68

2.00

16.67

0.81

0.20

2015-2020

0.38

2.53

3.00

28.33

0.71

0.20

4

2000-2005

-0.73

-37.79

-4.00

16.66

0.46

-0.40

2005-2010

0.24

-11.29

0.00

16.67

0.81

0.00

2010-2015

0.73

30.05

4.00

16.67

0.46

0.40

2015-2020

0.38

16.60

3.00

28.33

0.71

0.20

5

2000-2005

-1.22

-56.37

-6.00

16.66

0.22

-0.60

2005-2010

-0.24

-20.80

-2.00

16.67

0.81

-0.20

2010-2015

0.24

15.45

2.00

16.67

0.81

0.20

2015-2020

0.00

5.70

1.00

28.33

1.00

0.07

6

2000-2005

-1.71

-59.35

-8.00

16.66

0.08

-0.80

2005-2010

-0.73

-38.90

-4.00

16.67

0.46

-0.40

2010-2015

0.24

15.45

0.00

16.67

0.81

0.00

2015-2020

-1.13

-29.82

-7.00

28.33

0.26

-0.47

7

2000-2005

-1.71

-47.22

-8.00

16.66

0.08

-0.80

2005-2010

-0.73

-11.69

-4.00

16.67

0.46

-0.40

2010-2015

0.24

21.67

2.00

16.67

0.81

0.20

2015-2020

0.00

5.59

1.00

28.33

1.00

0.07

In order to compare the trend of total annualized water use of five reaches within five- year periods, the modified Mann–Kendall trend test was computed using “modifiedmk” package in RStudio (above Table).

Mann-Kendall test is a non-parametric test, used to recognize the possible monotonic upward or downward trend in a series of hydrological or climate data. The null hypothesis shows no trend in the series while the alternative hypothesis indicates that the data follow a monotonic trend. If -1.96 ≤ Z-value ≤ 1.96, the null hypothesis is accepted, which means that the data series has no trend. The data series were initially tested for autocorrelation. The most important statistics of a Mann-Kendall test include Z-value, a measure of significance of trend, Kendall's tau, a measure of correlation, and Sen's slope that shows the magnitude of the increase or decrease in the trend. According to the Z-values results (see Table) no trend can be found in all of the five reaches in 5-year periods. Observing trends over longer time periods will likely yield different results.

References

Kendall, M. (1975). Rank Correlation Methods. Griffin, London, 202 pp.

Mann, H. B. (1945). Nonparametric Tests Against Trend. Econometrica, 13(3): 245-259.

...............................

  1. The discussion section is too long. What are the new findings and limitations of this study? What are the key issues that deserve further investigation in future research?

Response:  

We removed the Discussion points that belonged in Results, which shortened this section substantially. We then completely wrote a new Discussion based on review comments. For example, we now include a section in Discussion that emphasizes the innovative methods applied to these semi-arid riparian woodland communities and the limitations of our techniques. We also discuss the limitations of this study with regards to what is still lacking, e.g., future work. We specifically highlight our new findings and a section discussing the limitations of the study. For example, VI-based ET methods should be applied primarily in regions where there are no illumination issues from haze, clouds, shadows, etc. We perform image correction, but this optical-based ET estimation does not perform well outside of arid and semi-arid regions. We added a section on how these VI-based ET findings have been applied in other arid/semi-arid regions such as some findings from Australia riparian zones, how riparian regions have declined elsewhere. We also added to the discussion the application of this method to urban gardens in Australia and how we expect these methods to work on the LCR for both the adjacent urban regions and the restored plots on the US portion as well as the restoration zones in Mexico.

One of the limitations of this study is that it is not compared to other ET estimates such as those many options available from USGS (SSEBop) and Open ET. We have the data for these comparisons for the LCR riparian corridor, but it was outside the scope of this ms and will be performed ‘next.’ Furthermore, we do expect to produce results using our EVI2 and ET(EVI2) methods on the urban regions within the LCR as a next step to much of the research led by Nouri et al. in urban spaces. And our future work in 2021 includes comparing the unrestored riparian corridor on the LCR to the Multi-Species Conservation Plan restoration sites, an important study (outside the scope of this ms) to the end-users, NGOs and resource managers of the LCR lands.

Please see the three new sections in the Discussion that capture our responses to your comment.

4.1. Remotely sensed findings and their implications

4.2. Strengths and limitations

4.3. New contributions and next steps

Submission Date

25 February 2021

Date of this review

08 Mar 2021 16:07:08

Round 2

Reviewer 2 Report

Thank you very much for addressing the revisions on the manuscript. Just an additional minor revision: Would you please improve the quality of the figures of the Appendi B. It is hard to read their content.

Author Response

Would you please improve the quality of the figures of the Appendi B. It is hard to read their content.

Response: We were very happy that you made suggestions to improve our contribution. The high-resolution images in Appendix B have been sent via email to the RS Editor. Thank you.

Reviewer 3 Report

The authors did a good job! I recommend this manuscript for publication in its present form.

Author Response

The authors did a good job! I recommend this manuscript for publication in its present form.

Response: Thank you very much. The comments you provided were indeed helpful in making our revision much better.

This manuscript is a resubmission of an earlier submission. The following is a list of the peer review reports and author responses from that submission.

Round 1

Reviewer 1 Report

General comment

The author studied the riparian ecosystem and its ET over 20 years, 10 years and 5 years by using three metrics (plant greenness, growing season ET, and full-year PAM-ET). The results are helpful for further deterioration of biodiversity, wildlife habitat and other key ecosystem services. While just from abstract, I can’t find the research background and objective of the study, and the whole structure and figures need to be adjusted. Also, the authors need to revise the manuscript carefully.

Major comments

  1. L17: in Abstract, it would be better to add the background and purpose of the study.
  2. L37: The title and abstract mainly focused on health and water use, but a lot of contents in the Introduction was used to introduce the content about ET (although water use in the study area was introduced in section 1.1), so it is suggested to modify it.
  3. L141-150: The relationship between climate and ET was not showed in the Result section.
  4. L243: in section 1.4 (Research Objective), the author proposed eight questions, please explain the main and specific problems to be solved in the research.
  5. L359-362: Is there a literature basis for such data processing method?
  6. L496: in Result, the author just simply introduced the graph, but did not describe the characteristics of data, figures, and tables. Besides, the author calculated the index and its changes of every year, every five years, every 10 years and 20 years. There are too many time scales to choose, so they need to explain why choose these scales.
  7. L627-636: What's the meaning of calculating the value of each year in each river (Figure 4 and 5 (a~e))? It’s suggested to put in the appendix.
  8. L746: It is difficult to judge the color, so the specific range can be marked on the figure.
  9. L799-802: the author just showed the value of 2010-2020, why did not show the data during 2000-2009?
  10. L849: please check the section number, it was 3.4.
  11. L863-877: The spatial distribution in Figure 11 and 12 was not clear. It is better to adjust the resolution and size. Besides, why choose the [2016 to 2020] – [2000 to 2005]?
  12. L886: in Discussion, many contents actually belong to Result section, so it’s suggested to adjust them.
  13. L1029-1032: please explain more details about this result.
  14. L1210-1223: The author said that tamarisk had a great influence, in fact, three influencing factors proposed in section 1.3 Objective should be analyzed in detail in Discussion. However, I only saw the description of a large number of data, it’s better to adjust the structure and the length of the article.

Author Response

The author studied the riparian ecosystem and its ET over 20 years, 10 years and 5 years by using three metrics (plant greenness, growing season ET, and full-year PAM-ET). The results are helpful for further deterioration of biodiversity, wildlife habitat and other key ecosystem services. While just from abstract, I can’t find the research background and objective of the study, and the whole structure and figures need to be adjusted. Also, the authors need to revise the manuscript carefully.

Response: Thank you for your review. I have added a sentence to describe the background and the objective in the abstract, but cannot stay under 200 words. I have decided to put several tables and figures into the appendices to make the manuscript shorter and easier to read the revised version.

These Supplemental Information files are now included in Appendix A (Tables) and Appendix B (Figures):

Table S1. Average peak growing season EVI2, the average and standard deviation from 2000-2020 for the Lower Colorado River.

Table S2. Differences in EVI2 demonstrating positive or negative (-) change in VI measured from one year to the next showing the average peak growing season, May 1 and Oct. 30, values for each reach and the total weighted average in All reaches between 2000-2020 for the Lower Colorado River.

Table S3. Landsat EVI2 percent change in the riparian vegetation greenness in the Lower Colorado River from year-to-year using average growing season dates from May 1 to Oct 30.

Table S4. The EVI2 peak growing season averaged per period. The EVI2 difference between groups of years for the peak growing season, averaged between May 1 and Oct. 30, is provided for each reach (R3..R7) and the weighted average as All Reaches. The groups include the entire 21-year period from 2000 to 2020, the ten-year periods from 2000-2010 and 2011-2020, the near-five years periods from 2000-2005 (six years), 2006-2010, 2011-2015, and 2016-2020.

Table S5. ET (mmd-1) from 2000-2020 shows the peak growing season water use for each reach and the total weighted average in All reaches for each of the 21 years in the study and the average and standard deviation over the study period is provided in the last lines of the table.

Table S6. Differences in the average daily water use using dates for the peak growing season for each of the 21 years in the study by subtracting the following year from the initial year as demonstrated in the first column. ET (mmd-1) change is either positive or negative (-) and the data is shown for each reach and the weighted average in All reaches, with the average and standard deviation over the study period provided in the last two lines of the table.

Figure A1 (a-e). EVI2 time-series information for the five reaches, R3..R7 (Figure A1a, A1b, A1c, A1d, A1e) for the full year with months on the x-axis (J-D), for years 2011 to 2020 individually and averaged by two decades, 2000-2010 and 2011-2020, and the full 21-year period, 2000-2020 as well as the two recent five-year periods, 2010-2015 and 2016-2020.

Major comments

  1. L17: in Abstract, it would be better to add the background and purpose of the study.

Response: This has been corrected in the abstract by including these first sentences and shortening other text in the abstract.

Declines in riparian ecosystem health and water use were observed in the delta of the Lower Colorado River (LCR) since 2000. The purpose of our case study was to measure these metrics on the U.S. side of the border between Hoover and Morelos dams to see if declining greenness was unique to the portion of the river in Mexico. In this case study, five riparian reaches of the LCR from Hoover to Morelos Dam since 2000 were studied to evaluate trends in riparian ecosystem health. We measure riparian water use using remotely sensed measurements of vegetation index (VI), a proxy for greenness, daily evapotranspiration (ET, mm/day) and the Phenology Assessment Metric (PAM), an annualized ET. We use 30 m Landsat time-series for NDVI and EVI2, the two-band EVI. We use weather station potential ET with EVI2 to parameterize the ET algorithm. A key finding is that riparian health and its water use has been in decline since 2000 on the U.S. portion of the LCR, depicting a loss of green vegetation over the last two decades. EVI2 results show a loss in greenness of 13.83% (decrease), while average daily water use between the first and last decades had a decrease in peak growing season (May 1 to October 30) ET of over 1 mmd-1 (27.30% decrease) and the average annual PAM ET losses were 170.91 mmyr-1 (17.95% decrease). We provide spatial change maps. Our results suggest further deterioration of biodiversity, wildlife habitat and other key ecosystem services on the U.S. portion of the LCR.

  1. L37: The title and abstract mainly focused on health and water use, but a lot of contents in the Introduction was used to introduce the content about ET (although water use in the study area was introduced in section 1.1), so it is suggested to modify it.

Response:  The content about the need for ET on lines 37-43 sets the background importance and will remain in the draft submitted, however, I have substantially revised other parts of the introduction.

  1. L141-150: The relationship between climate and ET was not showed in the Result section.

Response: The Results now contain a couple of plots to show the weather station data from each of the three AZMET stations. These include three stations from AZMET and the variables precipitation, temperature and potential ET (as well as discharge above and below Parker Dam) so that my added discussion points may include the impact of drought, in particular, on our ET findings. The AZMET locations are shown on the revised map.

  1. L243: in section 1.4 (Research Objective), the author proposed eight questions, please explain the main and specific problems to be solved in the research.

Response:  The Science Questions in the introduction are revised and arranged now to be answered in the Discussion. The primary research questions have been stated and the specific problems are now described.

  1. L359-362: Is there a literature basis for such data processing method?

Response:  This method was derived in the VIP lab for MODIS and VIIRS and was applied for the first time in the companion study in the delta (Nagler et al., 2020 Hydrological Processes) and the citation #14 has been added.

  1. Didan, K.; Barreto-Muñoz, A., Solano, R., & Huete, A. MODIS vegetation index user's guide (MOD13 series). Vegetation Index and Phenology Lab, The University of Arizona (pp. 1–38). 2015. Retrieved from https://vip.arizona.edu/MODIS_UsersGuide.php
  2. Didan, K.; Barreto-Muñoz, A., Tucker, C., & Pinzon, J. Suomi National Polar-orbiting Partnership, Visible Infrared Imaging Radiometer Suite, Vegetation Index Product Suite, User Guide & Abridged Algorithm Theoretical Basis Document. Vegetation Index and Phenology Lab, The University of Arizona (pp. 1–108). 2018. Retrieved from https://vip.arizona.edu/VIIRS_UsersGuide.php

  1. L496: in Result, the author just simply introduced the graph, but did not describe the characteristics of data, figures, and tables. Besides, the author calculated the index and its changes of every year, every five years, every 10 years and 20 years. There are too many time scales to choose, so they need to explain why choose these scales.

Response:  We have now moved the table and figure descriptions from Discussion to the Results and freed up room to better discuss the implications of drought and/or biocontrol on our ET findings. Furthermore, we have moved some tables and reach-level figures to Supplemental Information as described at the beginning of these response comments.

  1. L627-636: What's the meaning of calculating the value of each year in each river (Figure 4 and 5 (a~e))? It’s suggested to put in the appendix.

Response:  We initially thought that presenting each river reach would be of interest to the reader because of the species composition and land cover in each were very different; however, upon receiving the reviewer comments, we have decided to present the average of the reaches (All) and move the reach-level figures to Supplemental Information.

  1. L746: It is difficult to judge the color, so the specific range can be marked on the figure.

Response:  Thank you for pointing this out. We have provided the range using box and whisker plots and we just used color to demonstrate the range visually as some of the points showing the largest range may be overlooked.

  1. L799-802: the author just showed the value of 2010-2020, why did not show the data during 2000-2009?

Response: We originally could not fit all the years for all the reaches into one figure and have it still be legible or convey the meaning. Taking your comment into consideration, we now will show the average of all the reaches (All) for years 2000-2020 (we completed the dataset for 2020) and will move the individual reach data to the Supplemental Information.

  1. L849: please check the section number, it was 3.4.

Response:  Thanks for noting this and we changed it.

  1. L863-877: The spatial distribution in Figure 11 and 12 was not clear. It is better to adjust the resolution and size. Besides, why choose the [2016 to 2020] – [2000 to 2005]?

Response:  We can increase the spatial resolution if we use a full page for each reach, but that takes up too much space. The idea of these maps is not to show the pixel information in detail; instead, we wanted to show the general areas where increases (greening) or decreases (browning) in ecosystem health occurred. For users interested in the details of the maps, we will be distributing the TIFF files for these in our Data Release through the USGS Science Base, see the section at the end of the ms.

We wanted to describe the change in these 21 years. Instead of using just the first and last year, we used an average of several years to remove outliers that could be seen in using just one year. Thus, we used the beginning and end years, averaging a group of years for the first (6 years) and last (5 years) periods.

We also changed to NDVI heading to be NDVI* as we used the scaled NDVI for the maps.

  1. L886: in Discussion, many contents actually belong to Result section, so it’s suggested to adjust them.

Response: We agree; the initial draft submitted contained further description about the results in the Discussion section and we have now moved much of this to the Results and expanded on the Discussion points, hopefully making it a stronger research paper.

  1. L1029-1032: please explain more details about this result.

Response: We noticed that when showing results year on top of year that there were shifts in some years by as much as two months earlier, such as below.

In the discussion text, we now can discuss anomalies in all years because despite the crowded figure appearance, we plotted the first decade years above in a separate plot to compare with the last (plotted already, as in above). We now relate the anomalies in these to spring temperatures, precipitation, biocontrol defoliation and salt cedar dieback or other variables.

  1. L1210-1223: The author said that tamarisk had a great influence, in fact, three influencing factors proposed in section 1.3 Objective should be analyzed in detail in Discussion. However, I only saw the description of a large number of data, it’s better to adjust the structure and the length of the article.

Response:  We agree and moved much of the text in the Results to the Discussion section so that we can expand our discussion to include the influencing factors, including the impact of weather and defoliation from biocontrol.

Submission Date

25 December 2020

Date of this review

18 Jan 2021 04:13:41

Reviewer 2 Report

This is an impressive and important study that follows from the prior paper for the Colorado R. delta (paper #8). This follow-up study involved common methods and an overlapping focus, and this could lead to some streamlining of this paper.

It also seems to be a case study rather than a review and consequently the broader content could be shortened and the reference listing reduced. Three references may be sufficient to support most statements.

There is some overlap and duplication within and across the sections, such as relating to the objectives and methods. By merging these, the text could be collapsed.

The Results are extensive and I think that these could also be streamlined. The same outcomes are presented as full interval figures followed by tables with values, yearly changes and then groupings, and further figures. It is difficult for the reader to sort through and the theme may be diluted. I would recommend substantially cutting back the Results presentation. Some tables or figures could be provided as supplemental information or deleted, since the content overlaps. (Fig. 8 provides a clear summary, confirming the progressive decline.)

Some suggestions for the authors’ consideration:

Reference numbering is lost part-way through the listing (after 22).

Reference citation in the text includes a mixing of #’s and names.

There is inconsistency in presentation of 20 vs. 21 years.

The values presented throughout should be rounded off to reflect the precision of these measures.

Probably ‘USA’ throughout, rather than ‘US’.

Provide metric equivalents (acre ft, etc).

l. 129 – no need to indicate what’s coming up – it will follow

l. 156+ - this is an interesting and important section – recognizing the relatively recent inclusion of riparian woodland water use for water budgeting. Some of the prior content could be shortened to get the readers more directly to these key points that contribute to the motivation for this study:

  1. Water use by riparian woodlands has been somewhat neglected relative to water budgeting,
  2. Riparian woodlands provide rich and diverse ecosystems that are stressed by river water withdrawal and
  3. It is important to understand the consequences from actions such as tamarisk biocontrol.

l. 243 – yes!, extend the strategy upstream from the prior study along the delta zone

l. 254 – a few questions could be merged

l. 279 – ‘fewest’

l. 285 – delete final sentence

l. 293+ - more background content that could be collapsed

l. 322 – change in referencing - #’s vs. names

l. 326 – ‘2.2’ – this is another objectives section and overlaps with 1.3 and 1.4 – this content could be merged

l. 358 – restatement of prior content – could be shortened or deleted

l. 369 – collapse this description

l. 388 – specification could be with the prior methods

l. 418 – this seems to relate to the data source, already introduced – could be combined

l. 426 – overlaps with prior content – merge and collapse

l. 474 – collapse content, less explanation, especially here in Methods

l. 497 – this seems to be a restatement of Methods and then ‘forewarning’ of the Results sequence – may be unnecessary

l. 884 – delete ‘of’

Section 4.1 – seems to be more Results rather than Discussion - this content could be considerably reduced.

Fig. 1 – Define ‘R’s’. Indicate USA, and the locations for the two dams.

I’ve provided more specific suggestions for the earlier sections and there could be similar shortening of some of the latter sections.

This will provide a very strong paper. The science is solid and I think that streamlining the presentation could increase its accessibility to a range of readers.

Author Response

This is an impressive and important study that follows from the prior paper for the Colorado R. delta (paper #8). This follow-up study involved common methods and an overlapping focus, and this could lead to some streamlining of this paper.

Response:  Thank you very much for your review of this study that follows on the study in the delta (now #14). We agree this ms needed to be streamlined and shortened. We have decided to place many of the tables (now six tables in Appendix A) and figures (now includes in Appendix B five figures showing reach-level information) into Supplemental Information to make the ms easier to read.

It also seems to be a case study rather than a review and consequently the broader content could be shortened and the reference listing reduced. Three references may be sufficient to support most statements.

Response:  For this case study (not review), the references citing the statements made have been reduced throughout the manuscript. However, in the revision there are still many references overall which contribute importantly to the comprehensive background which is threefold and includes ET, riparian ecosystems and biocontrol.

There is some overlap and duplication within and across the sections, such as relating to the objectives and methods. By merging these, the text could be collapsed.

Response:  The overlap has been removed and there should be no further duplication.

The Results are extensive and I think that these could also be streamlined. The same outcomes are presented as full interval figures followed by tables with values, yearly changes and then groupings, and further figures. It is difficult for the reader to sort through and the theme may be diluted. I would recommend substantially cutting back the Results presentation. Some tables or figures could be provided as supplemental information or deleted, since the content overlaps. (Fig. 8 provides a clear summary, confirming the progressive decline.)

Response:  We have moved six tables and five figures into the Appendices. The time period tables included absolute values, differences and percent changes, many of which were duplicated in figures and thus these were moved to the Supplemental Information section. We now present three VIs in time series, daily ET for the full year (used for PAM ET) and the growing season, its yearly difference, and percent change between years, followed by annual PAM ET, its yearly difference, and percent change between years, and lastly change between groups of years to show significance by using more data per period.

Some suggestions for the authors’ consideration:

Reference numbering is lost part-way through the listing (after 22).

Response:  This has been corrected.

Reference citation in the text includes a mixing of #’s and names.

Response:  This has been corrected.

There is inconsistency in presentation of 20 vs. 21 years.

Response:  This has been corrected to be 21 years.

The values presented throughout should be rounded off to reflect the precision of these measures.

Response:  This has been corrected by rounding the values.

Probably ‘USA’ throughout, rather than ‘US’.

Response:  We are following USGS rules and using U.S. throughout

Provide metric equivalents (acre ft, etc).

Response:  This has been corrected by adding the metric equivalents to each place where sensible.

  1. 129 – no need to indicate what’s coming up – it will follow

Response:  This has been corrected by eliminating the text.

  1. 156+ - this is an interesting and important section – recognizing the relatively recent inclusion of riparian woodland water use for water budgeting. Some of the prior content could be shortened to get the readers more directly to these key points that contribute to the motivation for this study:
  1. Water use by riparian woodlands has been somewhat neglected relative to water budgeting,
  2. Riparian woodlands provide rich and diverse ecosystems that are stressed by river water withdrawal and
  3. It is important to understand the consequences from actions such as tamarisk biocontrol.

Response:  This has been corrected by revising and shortening the text and making the key points more clear and concise at the beginning of the Discussion section.

  1. 243 – yes!, extend the strategy upstream from the prior study along the delta zone

Reviewer 3 Report

This paper quantifies changes in Landsat based vegetation indices and derived estimates of evapotranspiration for 5 riparian reaches along the lower Colorado River over a 21-year period. While the underlying analytical methods are well established and sound, I do not find this paper to be suitable for publication in its current form as there appears to be erroneous information in the methods, and possibly in the results, and all sections need work in terms of clarity of writing and maintaining a focused storyline. I offer the following comments and suggestions for improving the paper:

  • The paper requires a thorough proofreading and edit. There are many incomplete sentences, grammatical errors, misplaced commas, missing words, inappropriate punctuation and mixes of verb tense.
  • The introduction, in particular, needs to be proofread and revised to be more succinct and linear. There is information in the introduction that is repeated in the methods and should be removed from one of those places.
  • The methods need to be clarified – it seems there are relict/recycled text in the method descriptions that are from another iteration of this study and/or a previous paper that don’t appear to be relevant to this study. For example, there are contrasting statements of what metrics are used and there are statements within the methods that appear to be redundant that should be tightened up. In particular, please clarify what NDVI and NDVI* were used for in this study and why NDVI is used for timeseries while NDVI* is used for change maps. It appears that NDVI* was not used to parameterize the ET model but there are sections in the methods that say it is.
  • All tables and figures should be checked to confirm that they are presenting the intended information
  • There is a high ratio of tables and figures to text in the results while much of the discussion appears to be describing the results with a very high level of detail. Text describing results in detail should be moved to the results section, then use the discussion to reiterate key points and interpret them. As it stands, the discussion leaves the reader mired in the minute reach-by-reach and metric-by-metric details and somewhat lost as to what the key findings are. In addition, the authors should consider whether any of the results can be culled or distilled to better elucidate the main findings. I think I gathered from the discussion that two of the reaches are tamarisk dominated and others aren’t. Instead of reporting for all reaches, could you report an aggregated result for tamarisk dominated and non-tamarisk dominated reaches if this is one of the key points you are trying to convey?
  • As a follow on to the above, I think a more directed and well-developed discussion is needed that elucidates the potential roles of biocontrol and climate. It doesn’t have to be conclusive necessarily but the motivation for looking at all of the different time periods on a reach-by-reach basis was presumably to try to sort this out? I think some of this is already in the paper but it is deeply buried amidst the detailed results descriptions and there needs to be more interpretation. It would be helpful to have a part of the discussion that focuses on biocontrol specifically to lay out what those impacts are to EVI2 and corresponding ET metrics and when/where they are observed.
  • In addition, given your conclusion of the role of climate impacts, these needs to be more discussion and substantiation of this. At a minimum, citations are needed to say that year x,y,z were dry/hot. Better yet, perhaps a plot of annual temp/precipitation or streamflow to show how these differ during the later years of the study would help to better substantiate your conclusions that this is what is driving the decline in the last years of the study. Or perhaps even evaporative demands/ETo – this would help emphasize the argument that reductions in ET are to be unexpected given that presumably ETo is above average during those years.

Minor comments:

Line 18 – shouldn’t this be 21 years? Please be consistent throughout paper

Lines 19-22 – This sentence is confusing and reads as a list of five different things, which I don’t think is what you intend. Perhaps using the sentence from lines 240-242 would help make this clearer.

Line 26 – show a decrease in greenness? (then parenthetical decrease could be removed)

Lines 43-44 ‘also used as inputs for’ is not consistent with rest of sentence consider using ‘determining’ instead

Sentence going from line 46-52 needs to be proofread and edited for clarity and consistency – too many ands make it read as a run-on sentence that is confusing.

Line 80 – missing ‘and’ between use and supply

Line 91 – describes shoud be plural

Line 99 – 100 proofread this sentence

Line 122-129. Not clear whether natural vegetation or narrow riparian ecosystems of the LCR are the focus do you mean narrow riparian systems composed of natural or restored vegetation? This would be a clearer statement, if so. Also – this is repeated in the study area section so you might consider removing this paragraph altogether.

Pargraph starting on line 151 – consider moving this to follow paragraph ending on line 121 –this section goes from talking about estimating ET in that paragraph, then to what the study area is, then to threats, then back to measurement of ET with remote sensing and it makes for a rambling read. You could even fold that threats paragraph in with the tamarisk info and give it a broader heading since it seems that you are wanting to suggest that climate, water extraction, and defoliation are the major driving variables of change so it would be helpful to discuss these together.

Line 160 – the sentence about phreatophyte water use is confusing. Is that part of the riparian ET that was reported since 2010 ? How does this relate to the rest of the information in the paragraph?

Line 170 – ‘with the mangement of water resources’ seems out of place in this sentence, Consider rewriting this sentence to improve clarity.

Line 173 – would this be more correctly described as a land cover change, rather than land use?

Lines 177-180 – sentence unclear - what is the need of particular importance?

Paragraph starting on line 172 and throughout paper – As I am sure the lead author knows - when referring to the species’ latin name it should be Tamarix spp (with italics, lowercase s) Tamarisk is a common name and doesn’t need to be italicized. It would be helpful to the reader to be consistent in use of term Tamarisk or salt cedar and not interchange them as is currently done. Similarly, be consistent in use of tamarisk defoliation beetle, tamarisk leaf beetle or Diorhabda sp. The scientific name is introduced twice in this paragraph and really only needs to be once unless that is how you refer to it in the rest of the paper. This comment applies to the entire paper

Lines 202 – consider revising to ‘Regardless of ‘ whether changes are due to biocontrol...and splitting this into two sentences

Line 230 – longer what?

Line 235 – maybe elaborate on why they were exempted –because of shadows? – actually this information is repeated in the study area section. Consider removing from here

Line 242 – PAM is not defined yet. Define it here.

246-247 – on the other side of the border?

Line 325 – Why not just call these metrics EVI2, ET and PAM ET throughout since it is already established that you used Landsat and EVI2 is the only VI you use to parameterize ET (I think – see comments below). The subscripts are unnecessary for this paper it would be much easier on the reader if this was simplified.

Line 347 temporal gaps resulting from what?

Line 347-57 – this paragraph seems almost entirely redundant to the paragraph that follows and possibly even the one after that. Consider removing this paragraph and adding any necessary additional detail contained here into the next paragraph(s).

Line 380 – how is this different from the continuity corrections mentioned in the paragraph beginning on line 347?

Line 405 – if EVI and EVI2 are the same, why not use EVI? Please explain why this index is preferred

Line 409? Should this be NDVI*? If not, I don’t understand how you conclude that NDVI* is a competent predictor

Paragraph beginning on line 407 – please rewrite for clarity. It isn’t clear whether you are saying you did this for this study or summarizing conclusions from another study.

Paragraph starting on lines 418 and 426 – can these be moved up to before the paragraph starting on line 407? It would be helpful to know what NDVI* is ahead of reading that paragraph

Paragraph starting on line 450 through end of section– I thought only EVI2 was used to parameterize the ET models (lines 312-315) but I see now that you refer to ETNDVI in line 334-335 and intermittently throughout the rest of the section. Is this an error or are you using both EVI2 and NDVI* to parameterize ET models? If so, why do you say only EVI2 was used in lines 312-315 and this is all that is mentioned in the abstract and this is all I see in the results. This needs to be clarified. The methods read as if they contain relict text from another study/paper or represent a previous iteration of this study that is not consistent with the results that are presented.

Line 479 – what is primarily used with agricultural crops? PAM? Please clarify this sentence

Line 489 – reference inconsistent with journal style

Lines 499-500 – metric names should have subscripts to be consistent with rest of paper (or better yet, no subscripts). Also, what about NDVI*? Isn’t this assessed too?

Line 504 – not clear what is meant by relative information

Line 507 – and NDVI*?

Line 522 – should this be NDVI*?

Line 520 - It would be helpful to preface the results with some explanation as to why all of these single, five, and ten-year comparisons are needed. Along those lines – are the 10-year comparisons really necessary? I see more value in the single and five year comparisons and there are a lot of results to sort through – less may be more in terms of getting the key points across.

Figure 2 – is this raw NDVI or NDVI*?

Figure 3a – It wasn’t immediately clear what the purpose of the colors are as it doesn’t look like R4 has the largest range at first glance – but I see now that it is when the outlier is included. Consider helping the reader out a little more by adding parenthetically what the ranges for R4 and R5 are in the text (i.e.,~ -8 – 17 for R4)

Figure 4  - consider removing the 2011-2020 and 2000-2020 average lines from the graph. If the point is to show that EVI2 is lower later in the time series, the 2000-2010 average is enough to demonstrate that and it would help to remove 2 lines from the already very busy graphic.

Figure 5 – why not just plot lines for each 5 year period in the timeseries in a similar spirit as to what was done in Figure 8 for ET? This would do a better job of conveying change over time. I don’t find the longer term averages to be helpful in telling a story and instead find them distracting.

Figure 9 – What does EVI2 mod mean? Is this a Modis result?

Table 10 – are these really differences? Shouldn’t they be negative?

Lines 881-885 – qualify this that you observed health in the delta on the Mexico side as declining. Also, only one type of imagery was used for this study. Please rewrite this for clarity

928-930 – can you elaborate more on how NDVI saturation leads to the declining pattern that is not seen in EVI2.

Line 1019 –unnecessary to list all of the subfigures and which reaches they correspond to here and elsewhere where this is done.

Lines 1206-1209 – how is the conclusion that R5 and R7 are somewhat unchanged drawn? Looking at raw and percent changes in ET/EVI – it appears there have been changes of 20% between the start and end of the time series based on table 11

1223: hanging The

Line 1260-1263 – what is meant by stability since 2000 – didn’t this paper demonstrate that the system is not stable, rather it is declining? Or do you mean it was stable until the last 5 years or so?

Line 1279 – remove ‘please add:’

References list is not formatted correctly after #22

Author Response

This paper quantifies changes in Landsat based vegetation indices and derived estimates of evapotranspiration for 5 riparian reaches along the lower Colorado River over a 21-year period. While the underlying analytical methods are well established and sound, I do not find this paper to be suitable for publication in its current form as there appears to be erroneous information in the methods, and possibly in the results, and all sections need work in terms of clarity of writing and maintaining a focused storyline.

Response:  Thank you for reviewing this manuscript that was assembled with haste to meet the SI deadline. We are in agreement with your recommendations; this version needed major editing and rearranging to provide clear objectives, simplify the results, use the discussion to link to answers related to the main research questions and to address drought and biocontrol, and to present our major findings in a more focused way.

We agree that it appeared that there was erroneous information in the methods in the original version because we at first included the one per year ET annual estimate based on NDVI* in a table, but decided not to use this. This language and the algorithm is removed. There was also missing symbolism (we left out the indicator (*) when discussing NDVI versus scaled NDVI (NDVI*)) and jargon (we should have eliminated “mod” meaning modified from the MODIS ET equation to be used now with Landsat). These details have now been cleared up. We also recognize that the reference numbering not only had some errors now corrected with the revision and rewritten text, but that under the section References, after #22 the numbers disappeared in the draft submitted for some unknown reason. Now, the references are all correctly numbered and follow the format for the journal without any being dropped.

I offer the following comments and suggestions for improving the paper:

  • The paper requires a thorough proofreading and edit. There are many incomplete sentences, grammatical errors, misplaced commas, missing words, inappropriate punctuation and mixes of verb tense.

Response:  Thank you for the comments on the ms writing; after re-organizing and pulling some of the information out to place into the Supplemental Information section, this ms is improved. The revision was thoroughly reviewed by an editor.

  • The introduction, in particular, needs to be proofread and revised to be more succinct and linear. There is information in the introduction that is repeated in the methods and should be removed from one of those places.

Response:  We have moved information from the methods ‘that were already in the introduction so it only appears one time.

  • The methods need to be clarified – it seems there are relict/recycled text in the method descriptions that are from another iteration of this study and/or a previous paper that don’t appear to be relevant to this study. For example, there are contrasting statements of what metrics are used and there are statements within the methods that appear to be redundant that should be tightened up. In particular, please clarify what NDVI and NDVI* were used for in this study and why NDVI is used for timeseries while NDVI* is used for change maps. It appears that NDVI* was not used to parameterize the ET model but there are sections in the methods that say it is.

Response:  The information in the methods regarding ET from the scaled NDVI (“NDVI*”) has been removed. We originally produced an annual ET estimate using NDVI* but did not include this table in the submitted ms. The Methods text section has been revised and now we refer to only these two uses: (i) the time-series information NDVI for use in doing a comparison of VI trends and because NDVI is a standard product that everyone knows/uses, and (ii) NDVI* information for use in the change maps in this ms and not for ET maps. We accidentally left off the “*” to indicate in the change maps that we were using the NDVI* and not NDVI. For clarity, NDVI* is an additional correction to the NDVI to account for the illumination conditions; in other words, it is a processing correction step to improve the usefulness of NDVI. The primary reason for using the NDVI* scaling method is that it takes into account changing illumination conditions scene-to-scene. Because our ET formula accounts for using EVI or EVI2, and not NDVI or NDVI*, the scaled NDVI version is not used to parameterize our ET model at all. For our change maps, we do use NDVI* as it is improved over the NDVI. We do not plot the time-series data as NDVI* because we not find it necessary. We have removed any text that says we used NDVI* to parameterize the ET model which would have produced an annual value for ET.

  • All tables and figures should be checked to confirm that they are presenting the intended information

Response:  Thank you, we have moved most of the tables into Supplemental Information, Appendix A, and one of the figures’ reach-level plots into Appendix B. All the other information in the plots and tables that was erroneous was removed.

  • There is a high ratio of tables and figures to text in the results while much of the discussion appears to be describing the results with a very high level of detail. Text describing results in detail should be moved to the results section, then use the discussion to reiterate key points and interpret them. As it stands, the discussion leaves the reader mired in the minute reach-by-reach and metric-by-metric details and somewhat lost as to what the key findings are. In addition, the authors should consider whether any of the results can be culled or distilled to better elucidate the main findings. I think I gathered from the discussion that two of the reaches are tamarisk dominated and others aren’t. Instead of reporting for all reaches, could you report an aggregated result for tamarisk dominated and non-tamarisk dominated reaches if this is one of the key points you are trying to convey?

Response:  This has been corrected by moving the text previously in the Discussion section to the Results. We also used the initial section of the Discussion to focus on the key points. We used the break out sections, VI, daily ET and annual PAM ET, to discuss the interpretation of the findings and answer the research questions. We cannot for this study go back and redefine the native and salt cedar dominated areas, however we are doing this for a future manuscript and comparing those results to restoration plots. For this study, we add discussion to better describe the reach plant composition.

  • As a follow on to the above, I think a more directed and well-developed discussion is needed that elucidates the potential roles of biocontrol and climate. It doesn’t have to be conclusive necessarily but the motivation for looking at all of the different time periods on a reach-by-reach basis was presumably to try to sort this out? I think some of this is already in the paper but it is deeply buried amidst the detailed results descriptions and there needs to be more interpretation. It would be helpful to have a part of the discussion that focuses on biocontrol specifically to lay out what those impacts are to EVI2 and corresponding ET metrics and when/where they are observed.

Response:  Thank you for this advice and in response we have added in two figures to show the weather station data we used from the three nearest AZMET stations and revised the Discussion to talk about the relationship between these variables (precipitation, temperature and potential ET) to the full-year EVI2 results (now expanded to include 2000-2010 in detail as with the existing 2011-2020) which show in 16 day periods how the EVI2 responded in part to the weather data. In other places where we see dips in the EVI2 curve, we attempt to correlate it with information we have about biocontrol during the time period.

  • In addition, given your conclusion of the role of climate impacts, these needs to be more discussion and substantiation of this. At a minimum, citations are needed to say that year x,y,z were dry/hot. Better yet, perhaps a plot of annual temp/precipitation or streamflow to show how these differ during the later years of the study would help to better substantiate your conclusions that this is what is driving the decline in the last years of the study. Or perhaps even evaporative demands/ETo – this would help emphasize the argument that reductions in ET are to be unexpected given that presumably ETo is above average during those years.

Response:  The plots of weather station information have now been added. We used the three nearest AZMET stations and revised the Discussion to talk about the relationship between these variables (precipitation, temperature and potential ET) and how they corresponded to the full-year EVI2 results (now expanded to include 2000-2010 in detail as with the existing 2011-2020) which we show in 16 day periods.

Minor comments:

Line 18 – shouldn’t this be 21 years? Please be consistent throughout paper

Response:  Yes, it is 21 years. We have now made this consistent throughout the ms

Lines 19-22 – This sentence is confusing and reads as a list of five different things, which I don’t think is what you intend. Perhaps using the sentence from lines 240-242 would help make this clearer.

Response:  This has been revised considerably and no longer appears in the same place, as it has moved to objectives and has been written more concisely.

Line 26 – show a decrease in greenness? (then parenthetical decrease could be removed)

Response:  This has been corrected by using decrease and showing percentages as (-) for decreasing in the text, but are symbolized by “–“ in tables.

Lines 43-44 ‘also used as inputs for’ is not consistent with rest of sentence consider using ‘determining’ instead

Response:  This has been corrected by eliminating ‘used as inputs for’ and replaced with “also determining.”

Sentence going from line 46-52 needs to be proofread and edited for clarity and consistency – too many ands make it read as a run-on sentence that is confusing.

Response:  This has been corrected by breaking this into two sentences and revising.

Line 80 – missing ‘and’ between use and supply

Response:  We moved the quotation mark and added ‘and’ between use and supply.

Line 91 – describes shoud be plural

Response:  This has been corrected by adding an ‘s’ to describe.

Line 99 – 100 proofread this sentence

Response:  This has been corrected by adding missing words and removing ‘restore damaged’ and reads like this now: “Water shortage in the Colorado River Basin is one of the major threats to riparian ecosystems.”

Line 122-129. Not clear whether natural vegetation or narrow riparian ecosystems of the LCR are the focus do you mean narrow riparian systems composed of natural or restored vegetation? This would be a clearer statement, if so. Also – this is repeated in the study area section so you might consider removing this paragraph altogether.

Response:  This has been corrected by stating this only one time. We mean “narrow riparian systems composed of natural or restored vegetation”

Pargraph starting on line 151 – consider moving this to follow paragraph ending on line 121 –this section goes from talking about estimating ET in that paragraph, then to what the study area is, then to threats, then back to measurement of ET with remote sensing and it makes for a rambling read. You could even fold that threats paragraph in with the tamarisk info and give it a broader heading since it seems that you are wanting to suggest that climate, water extraction, and defoliation are the major driving variables of change so it would be helpful to discuss these together.

Response:  This entire paragraph was removed. Later, now in the Discussion section, we come back to the main point of it (more clearly) and tie in our weather station trends and the timing of biocontrol in the LCR as we know of to date.

Line 160 – the sentence about phreatophyte water use is confusing. Is that part of the riparian ET that was reported since 2010 ? How does this relate to the rest of the information in the paragraph?

Response:  This was removed, but our ET results do capture phreatophyte water use, but for the background information, we now separate out the studies that were specifically applied for phreatophytes.

Line 170 – ‘with the mangement of water resources’ seems out of place in this sentence, Consider rewriting this sentence to improve clarity.

Response:  This has been corrected by eliminating this sentence.

Line 173 – would this be more correctly described as a land cover change, rather than land use?

Response:  This has been corrected by replacing use with cover

Lines 177-180 – sentence unclear - what is the need of particular importance?

Response:  This has been revised for clarity: “Our goal of quantifying not only riparian woodlands water use but also discriminating non-native plant area dominated by salt cedar and quantifying its wide-area, reach-level water use, is of particular importance to land managers in the southwestern U.S.”

Paragraph starting on line 172 and throughout paper – As I am sure the lead author knows - when referring to the species’ latin name it should be Tamarix spp (with italics, lowercase s) Tamarisk is a common name and doesn’t need to be italicized. It would be helpful to the reader to be consistent in use of term Tamarisk or salt cedar and not interchange them as is currently done. Similarly, be consistent in use of tamarisk defoliation beetle, tamarisk leaf beetle or Diorhabda sp. The scientific name is introduced twice in this paragraph and really only needs to be once unless that is how you refer to it in the rest of the paper. This comment applies to the entire paper

Response:  This has been corrected by using only salt cedar and salt cedar leaf beetle and introducing the latin name the initial time. There are four Diorhabda species, two discussed in the ms. the northern and the subtropical ones, so we used spp.

For example: “…by salt cedar (Tamarix spp.), and more recently by the Tamarix defoliation beetles (Diorhabda spp.)…”

Lines 202 – consider revising to ‘Regardless of ‘ whether changes are due to biocontrol...and splitting this into two sentences

Response:  Thank you for the suggestion, we removed the sentence altogether.

Line 230 – longer what?

Response:  This has been removed (what was meant by longer was more days of maximum high temperatures)

Line 235 – maybe elaborate on why they were exempted –because of shadows? – actually this information is repeated in the study area section. Consider removing from here

Response:  The objectives were revised and occur in only one place. This section has some added description as to why: (the first two reaches were in canyons with little riparian vegetation and exempted from this study)

Line 242 – PAM is not defined yet. Define it here.

Response:  Thank you, this has been corrected by defining PAM even earlier in the document

246-247 – on the other side of the border?

Response:  This has been corrected by describing the geographic location relative to the dams:

U.S. side of the border between Hoover and Morelos Dams … in the delta of the Colorado River south of Morelos Dam and into Mexico.”

Line 325 – Why not just call these metrics EVI2, ET and PAM ET throughout since it is already established that you used Landsat and EVI2 is the only VI you use to parameterize ET (I think – see comments below). The subscripts are unnecessary for this paper it would be much easier on the reader if this was simplified.

Response:  I really like this suggestion and have changed all the names to be EVI, ET and PAM ET. I was trying to be consistent with the other study in the delta which used both Landsat and MODIS, but in this work, I am only using Landsat as you mention and so the label Landsat is not needed.

Line 347 temporal gaps resulting from what?

Response:  This has been corrected by revising the entire section and now reads more clearly. The temporal gaps are due to ETM+ which developed a scan line issue back in 2003 and there are holes in its temporal sequence, but we filled in the 2012 period between TM5 and TM8 OLI using this data.

Line 347-57 – this paragraph seems almost entirely redundant to the paragraph that follows and possibly even the one after that. Consider removing this paragraph and adding any necessary additional detail contained here into the next paragraph(s).

Response:  This is true, the information was duplicative and has been revised.

Line 380 – how is this different from the continuity corrections mentioned in the paragraph beginning on line 347?

Response:  This has been corrected in the revision.

Line 405 – if EVI and EVI2 are the same, why not use EVI? Please explain why this index is preferred

Response:  Thank you, and the reason is now stated more clearly in the text. The EVI and EVI2 have an r2 = 0.99 and either could be used; however, the two-band version, EVI2, is a simpler VI because it does not require the blue band. For this reason, we are using EVI2.

Line 409? Should this be NDVI*? If not, I don’t understand how you conclude that NDVI* is a competent predictor

Response:  We originally calculated ET using annual ETo x NDVI*, but given that we produce one ET estimate per year, we eliminated this table from our study and I failed to remove the text in this section.

Paragraph beginning on line 407 – please rewrite for clarity. It isn’t clear whether you are saying you did this for this study or summarizing conclusions from another study.

Response:  This is now eliminated. There is no text talking about ET derived from NDVI* and annual ETo. We do provide background on NDVI*

Paragraph starting on lines 418 and 426 – can these be moved up to before the paragraph starting on line 407? It would be helpful to know what NDVI* is ahead of reading that paragraph

Response:  This has been corrected by moving the information up in the text so that NDVI* is defined prior.

Paragraph starting on line 450 through end of section– I thought only EVI2 was used to parameterize the ET models (lines 312-315) but I see now that you refer to ETNDVI in line 334-335 and intermittently throughout the rest of the section. Is this an error or are you using both EVI2 and NDVI* to parameterize ET models? If so, why do you say only EVI2 was used in lines 312-315 and this is all that is mentioned in the abstract and this is all I see in the results. This needs to be clarified. The methods read as if they contain relict text from another study/paper or represent a previous iteration of this study that is not consistent with the results that are presented.

Response:  This information is removed and the reason is that we did produce it for the annual ET for 21 years and then removed it as it did not add much to the study and we wished to shorten the ms.

Line 479 – what is primarily used with agricultural crops? PAM? Please clarify this sentence

Response:  The growing season between May and October is often used to define peak water use, particular in agricultural water use studies. We wanted to calculate the full year water use for these riparian species. By using the area under the VI curve in PAM ET, we capture defoliation or other plant stress that lowers water use during the peak growing season as well as the full year.

Line 489 – reference inconsistent with journal style

Response:  Thank you, yes this is an error that needed to be corrected. It is now correct.

Lines 499-500 – metric names should have subscripts to be consistent with rest of paper (or better yet, no subscripts). Also, what about NDVI*? Isn’t this assessed too?

Response:  We removed this section. We show NDVI and provide change maps using NDVI* because much of the readership will want to see this index and for scene-to-scene comparisons of change the scaled version is required (hence NDVI* for maps). We changed the text to say we assessed NDVI and NDVI* but our assessment is really confined to EVI2, ET and PAM ET.

Line 504 – not clear what is meant by relative information

Response:  This has been eliminated, revised or shortened, but what was meant is the relevant years for each the absolute value, the difference and percent change.

Line 507 – and NDVI*?

Response:  Not needed here, we only used EVI2 for our main results, ET and PAM ET. See next response.

Line 522 – should this be NDVI*?

Response:  No, we produced NDVI times series for comparison with EVI2 to see the amplitude change, but NDVI* (an improved, corrected for illumination version) was used to do the scene-to-scene comparison of change depicted in the maps.

Line 520 - It would be helpful to preface the results with some explanation as to why all of these single, five, and ten-year comparisons are needed. Along those lines – are the 10-year comparisons really necessary? I see more value in the single and five year comparisons and there are a lot of results to sort through – less may be more in terms of getting the key points across.

Response:  The data is shown for these time periods because to really determine percent change, the more years, the more robust the results. Using a group of years decreases error. We decided to use only the 21 years averaged as a bold line from comparison, however we eliminated the other two average lines as suggested.

We originally used the two decades because from ca. 2000-2010 the peaks in EVI2 were approximately the same level, but from 2010 they began to decline, falling below an EVI2 = 0.20 which lasted through 2020. Also, the data from a 10-year period is the maximum amount to plot to show detail, and it is easier to show two plots of 10-years.

We moved some of the data in the tables to the Supplementary Information to focus more on the graphic results in the main text.

Figure 2 – is this raw NDVI or NDVI*?

Response:  This is plain NDVI, a standard NASA product, widely used for comparison with EVI2. We only use the enhanced, corrected NDVI* for scene-to-scene comparisons, e.g., maps.  We could show NDVI* in time-series if requested.

Figure 3a – It wasn’t immediately clear what the purpose of the colors are as it doesn’t look like R4 has the largest range at first glance – but I see now that it is when the outlier is included. Consider helping the reader out a little more by adding parenthetically what the ranges for R4 and R5 are in the text (i.e.,~ -8 – 17 for R4)

Response:  This is another good point and has been added to the Figure text. We are unsure how to represent negative percent changes so we use (-) as described.  “R4 shows the largest range of change in EVI2 ((-)8.33% to 17.49%, followed by R5 ((-)9.09% to 16.35%).”

Figure 4  - consider removing the 2011-2020 and 2000-2020 average lines from the graph. If the point is to show that EVI2 is lower later in the time series, the 2000-2010 average is enough to demonstrate that and it would help to remove 2 lines from the already very busy graphic.

Response:  This has been considered. In responding to another reviewer about the first ten individual years, and for help in writing the discussion, we plotted each year and are only showing the average of one of the lines on each of the two plots (first decade and second decade) and decided to use the last 5 years, 2016-2020, for comparison and based on your previous comment.

Figure 5 – why not just plot lines for each 5 year period in the timeseries in a similar spirit as to what was done in Figure 8 for ET? This would do a better job of conveying change over time. I don’t find the longer term averages to be helpful in telling a story and instead find them distracting.

Response:  The reason we chose not to use a bar chart but rather a time-series by year for these plots is that we wish to see the months, the seasonal behavior of the data. From this we can tell if the onset of greening was earlier in one period over another and whether late-spring or mid-summer decreases show up due to biocontrol. Since this could not be determined using the multi-year averages, we moved this information to the Supplementary Information section for the individual reaches.

Figure 9 – What does EVI2 mod mean? Is this a Modis result?

Response:  This was an error to have it on the figure template. This means modified from the ET equation using MODIS EVI for using ET from Landsat EVI2.

Table 10 – are these really differences? Shouldn’t they be negative?

Response:  This has been corrected to state the values are the averages (not differences).

Lines 881-885 – qualify this that you observed health in the delta on the Mexico side as declining. Also, only one type of imagery was used for this study. Please rewrite this for clarity

Response:  This has been removed because another review did not think we should foreshadow the coming section, so we just removed the text. Also, it was already described previously.

928-930 – can you elaborate more on how NDVI saturation leads to the declining pattern that is not seen in EVI2.

Response:  The best way to explain this is to show the “Theoretical Basis for EVI” and the following graphic: see Didan, K.; Barreto-Muñoz, A., Solano, R., & Huete, A. MODIS vegetation index user's guide (MOD13 series). Vegetation Index and Phenology Lab, The University of Arizona (pp. 1–38). 2015. Retrieved from https://vip.arizona.edu/MODIS_UsersGuide.php

Liine 1019 –unnecessary to list all of the subfigures and which reaches they correspond to here and elsewhere where this is done.

Response:  These reach level plots have been moved to the Supplemental Info (Appendix B)

Lines 1206-1209 – how is the conclusion that R5 and R7 are somewhat unchanged drawn? Looking at raw and percent changes in ET/EVI – it appears there have been changes of 20% between the start and end of the time series based on table 11

Response:  This information about observations in each reach, especially between R3 & R6 and R5 & R7, has been further explained. The reaches with more salt cedar dominance are R3 & R6. All reaches show negative trends, declines in greenness, ET, and PAM ET. I have changed the wording in the text to reflect the change values we recorded in all reaches and removed the ‘somewhat unchanged’ description, which is not scientific anyhow.

The percent change in PAM ET between the first and last 5-year periods are smallest for R4, R7 and R5, but are still on the order of 20%, but R3 is 34% and R6 is 29%, both reaches have considerably larger negative change (see table). PAM ET percent change is negative for each reach, but less so in R5 and R7 (see figure).

Period 1

Period 2

R3

R4

R5

R6

R7

2000-2005

2016-2020

-33.57

-19.52

-21.88

-29.20

-19.53

The decline in daily ET (negative change) is also seen across each reach, but the drop is less drastic in R5 and R7. 

From the change maps that use NDVI* and ET from EVI2, for the two periods of change (Figure 11 and 12), we see much darker browns where salt cedar exists at Topock Marsh and Cibola NWR. In R3 and R6, we know these darker brown patches, indicating maximum browning or maximum negative change in greenness, are where salt cedar has been validated on the ground (studies Nagler et al., 2005, 2008). In R5 and R7, the color indicates that not much as changed in comparison to the color scheme in the other reaches. Also Reach 5 (Figure 4c) and Reach 7 (Figure 4e) show the least range of annual cycle data; Reach 3 and Reach 6 show the largest ranges being from decreases primarily in the last few years.

1223: hanging The

Response:  oops I must have had a thought (ADHD).  This has been removed

Line 1260-1263 – what is meant by stability since 2000 – didn’t this paper demonstrate that the system is not stable, rather it is declining? Or do you mean it was stable until the last 5 years or so?

Response:  I intended that the system was stable the first decade or so, then started to decline ca. 2010-2020, and was very unstable the last five years.

Line 1279 – remove ‘please add:’

Response:  This has been removed.

References list is not formatted correctly after #22

Response:  The references are now cited correctly for the revision and are all showing up in the REFERENCE list (for some reason the numbering went AWOL in the original submitted copy).

Submission Date

25 December 2020

Date of this review

09 Jan 2021 20:33:37
